# Continental-scale controls on soil organic carbon across sub-Saharan Africa

Sophie F. von Fromm[1,2], Alison M. Hoyt[1,3], Markus Lange[1], Gifty E. Acquah[4], Ermias Aynekulu[5], Asmeret Asefaw Berhe[6], Stephan M. Haefele[4], Steve P. McGrath[4], Keith D. Shepherd[5], Andrew M. Sila[5], Johan Six[2], Erick K. Towett[5], Susan E. Trumbore[1], Tor-G. Vågen[5], Elvis Weullow[5], Leigh A. Winowiecki[5], Sebastian Doetterl[2]

[1]Department of Biogeochemical Processes, Max-Planck Institute for Biogeochemistry, Jena, Germany

[2]Department of Environmental Systems Science, ETH Zurich, Zurich, Switzerland

[3]Climate and Ecosystem Sciences Division, Lawrence Berkeley National Laboratory, Berkeley, CA, USA

[4]Department of Sustainable Agriculture Sciences, Rothamsted Research, Harpenden, UK

[5]World Agroforestry Centre (ICRAF), Nairobi, Kenya

[6]Department of Live and Environmental Sciences, University of California Merced, Merced, CA, USA

Correspondence to: Sophie F. von Fromm, Max-Planck Institute for Biogeochemistry, Hans-Knoell-Street 10, 07745 Jena, Germany, sfromm@bgc-jena.mpg.de, phone: +49 3641-576184

**Abstract**

Soil organic carbon (SOC) stabilization and destabilization has been studied intensively. Yet, the factors which control SOC content across scales remain unclear. Earlier studies demonstrated that soil texture and geochemistry strongly affect SOC content. However, those findings primarily rely on data from temperate regions where soil mineralogy, weathering status and climatic conditions generally differ from tropical and sub-tropical regions. We investigated soil properties and climate variables influencing SOC concentrations across sub-Saharan Africa. A total of 1,601 samples were analyzed, collected from

two depths (0–20 cm and 20–50 cm) from 17 countries as part of the Africa Soil Information Service project (AfSIS). The data set spans from arid to humid climates and includes soils with a wide range of pH values, weathering status, soil texture, exchangeable cations, extractable metals and land cover types. The most important SOC predictors were identified by linear mixed-effects models, regression trees and random forest models. Our results indicate that geochemical properties, mainly oxalate-extractable metals (Al and Fe) and exchangeable Ca, are equally important compared to climatic variables (mean

annual temperature and aridity index). Together, they explain approximately two thirds of SOC variation across sub-Saharan Africa. Oxalate-extractable metals were most important in wet regions with acidic and highly weathered soils, whereas exchangeable Ca was more important in alkaline and less weathered soils in drier regions. In contrast, land cover and soil texture were not significant SOC predictors on this large scale. Our findings indicate that key factors controlling SOC across sub-Saharan Africa are broadly similar to those in temperate regions, despite differences in soil development history.

**Keywords**: biogeochemistry, land-use, soil organic matter, clay mineralogy, subtropical soils

# 1. Introduction

Soil conservation and sustainable management are crucial to address some of the main challenges humanity is facing, such as climate change, food security, environmental degradation, and loss of soil biodiversity. Assessing the state of soils and their potential responses to climate and land-use change requires carefully designed sampling strategies, combined with systematic analytical and statistical analyses across locations and scale (IPCC, 2019). One key component is soil organic carbon (SOC). Due to its variety of sources, transformations and stabilization mechanisms, SOC is chemically very complex and spatially heterogeneous. This complexity causes significant uncertainties in global climate models (Friedlingstein et al., 2014). It also complicates the extrapolation of SOC to a global scale using statistical relationships to build robust global SOC products, such as SoilGrids and the Harmonized World Soil Database (Tifafi et al., 2018). To improve our understanding of global C dynamics, it is important to better understand the factors that control SOC stabilization and destabilization in soils from regional to global scales (Blankinship et al., 2018; Heimann and Reichstein, 2008).

SOC-stabilizing drivers and processes have been intensively studied over the past several decades. Dokuchaev (1883) and Jenny (1941) shaped the understanding that soil properties are correlated with (independent) variables – the so-called soil-forming factors (eq. 1):

$$s = f'(cl, o, r, p, t) \qquad (1)$$

where $s$ stands for any type of soil property, such as pH, carbon content, mineralogy, etc., and is determined by the function $f'$ of soil-forming factors: $cl$ – climate, $o$ – organisms, $r$ – topography, $p$ – parent material, and $t$ – time. This concept is still relevant and forms the basis for many experiments and research attempting to understand SOC storage. However, the importance of the individual factors of equation (1) at different spatiotemporal scales remains unclear (Doetterl et al., 2015; Rasmussen et al. 2018; Wiesmeier et al., 2019). This uncertainty hinders implementation of equation (1) in Earth System models, resulting in a gap between the theoretical understanding of SOM dynamics and our ability to improve terrestrial biogeochemical projections that rely on existing models (Blankinship et al., 2018; Rasmussen et al., 2018; Schmidt et al., 2011). Despite the long history of studying SOC stabilization (Greenland, 1965; Oades, 1988), there still is increasing demand for data on SOC dynamics at landscape to global scales (Blankinship et al., 2018), especially from sub-tropical and tropical ecosystems.

SOC stabilization is commonly conceptualized as competition between accessibility for microorganisms versus chemical associations with minerals (Oades, 1988; Schmidt et al., 2011). These processes are often only considered implicitly by models (Blankinship et al., 2018; Schmidt et al., 2011). Instead, models commonly rely on broader variables such as clay content, which is used as a proxy for sorption and other organo-mineral interactions (Rasmussen et al., 2018; Schmidt et al., 2011). These more generic variables integrate a variety of stabilization processes which can be difficult to disentangle. They can differ in their relative importance and may not adequately capture soil mineralogy and chemistry across different ecosystems and climate zones. Hence, improving the predictive capacity of such models requires not only a better understanding of the factors

that control SOC dynamics, but also verification (or falsification) of those new findings in regions that are underrepresented in field studies and models.

For example, Rasmussen et al. (2018) found that exchangeable Ca was correlated with the quantity of SOC in water-limited soils, while $Al_{ox}$ was a better predictor of SOC in wet, acidic soils. However, those findings may not be directly transferable to sub-tropical and tropical soils, since they differ greatly in climate, parent material and vegetation (Six et al., 2002b), which usually results in more weathered and older soils compared to those in temperate regions (Feller and Beare, 1997). This was illustrated recently in Quesada et al. (2020), where SOC variation in highly weathered forest soils from across the Amazon basin was best explained by clay content, whereas the best explanatory variables for less-weathered soils were Al species, pH and litter quality. Feller and Beare (1997) also found that tropical soils, dominated by low-activity clays (i.e. 1:1 clays), show a strong relationship between SOC and clay + silt content. In addition, Barthès et al. (2008) found that sesquioxides (Al and Fe) play an important role in SOC stabilization for various tropical soils. However, the relationship for high activity clays (i.e. 2:1 clays) is less clear and contrasting trends between SOC and clay + silt content have been reported (Feller and Beare, 1997; Six et al., 2002a). In terms of SOC distribution across sub-Saharan Africa, Vågen et al. (2016) showed, by using a data set similar to this paper, that SOC content was highest in equatorial and warm temperate climates, where sand content, sum of base concentrations and pH values were low. With regard to land cover, it has been shown for several sites across Africa that forests usually contained the highest amount of SOC, whereas differences between cropland, grassland and shrubland were less distinct (Abegaz et al., 2016; Olorunfemi et al., 2020; Winowiecki et al., 2016a). Cropland cultivation decreased carbon content by 50% compared to forested and semi-natural plots for sites in Tanzania, regardless of sand content and topographic position (Winowiecki et al., 2016b). However, land degradation (i.e. erosion) resulted in SOC concentration decreases in those ecosystems; independent of vegetation cover (Winowiecki et al., 2016a).

To address these diverging explanations of SOC variations on regional scales, we analyzed a comprehensive soil data set collected across the African continent using the Land Degradation Surveillance Framework (Vågen et al., 2010). This data set covers a wide range of climatic and mineralogical conditions – from very arid to humid regions, with different $pH_{H2O}$ values, soil texture, weathering status, exchangeable cations and extractable metals – allowing us to test different parameters to explain the variation in SOC content in subtropical and tropical soils across sub-Saharan Africa for two distinctive depth layers (topsoil: 0–20 cm and subsoil: 20–50 cm). Here, we use this continental-scale data set to address the following research questions:

1. Which soil properties and climate parameters best explain SOC content variation across sub-Saharan Africa?

    We explored the importance of soil texture, exchangeable Ca, oxalate-extractable Al and Fe, soil $pH_{H2O}$, mean annual temperature, aridity index (PET/MAP), land cover and weathering status to explain variation in SOC content on a continental scale. We expect that oxalate-extractable metals, soil texture and climate will be among the most important predictors of SOC concentration.

2. How do geochemical SOC-controlling factors vary between environmentally distinct sub-regions?

Due to the heterogeneity of climate and soil conditions across sub-Saharan Africa, we expect to see different geochemical controls explaining variations in SOC content between regions. For example, we expect exchangeable Ca will be most important in regions that are drier with less weathered and alkaline soils, while oxalate-extractable Al and Fe will mainly be important in humid regions with highly weathered and acidic soils.

## 2. Methods

### 2.1. Study area and data collection

Soil data used in this study were collected during the AfSIS (Africa Soil Information Service) project. In total, 18,257 soil samples were taken from 60 sentinel sites and from two different depths (topsoil: 0–20 cm and subsoil: 20–50 cm). Samples stem from 19 countries across sub-Saharan Africa and were collected between 2009 and 2012, following the well-established Land Degradation Surveillance Framework (Vågen et al., 2010). The sixty sentinel sites (each 100 km²) were stratified across sub-Saharan Africa according to Koeppen-Geiger zones (Vågen et al., 2016). Ten 1000 m² plots were randomized within sixteen spatially stratified 1 km² clusters per site (Figure 1). This hierarchical sampling design allows process identification at a continental scale without losing the ability to understand and quantify local heterogeneity (Nave et al., 2021; Vågen et al., 2010). For more details about sampling design and field survey, see Towett et al. (2015);Vågen et al. (2013a), and Winowiecki et al. (2016a).

Our analyses built upon a subset of samples (11% of total, n = 2,002) which were originally selected as reference samples for laboratory measurements. These samples were used to calibrate mid-infrared spectroscopy models (Terhoeven-Urselmans et al., 2010) and to predict properties in the remaining 16,255 soil samples (Vågen et al., 2016; Winowiecki et al., 2017). The calibration subset was chosen to maximize the variation of the spectral data using the Kennard-Stone algorithm (Kennard and Stone, 1969). More information about this approach can be found in Terhoeven-Urselmans et al. (2010). This selection strategy results in unequally distributed samples across 51 of the 60 sentinel sites, yet captures the variation of the original data set.

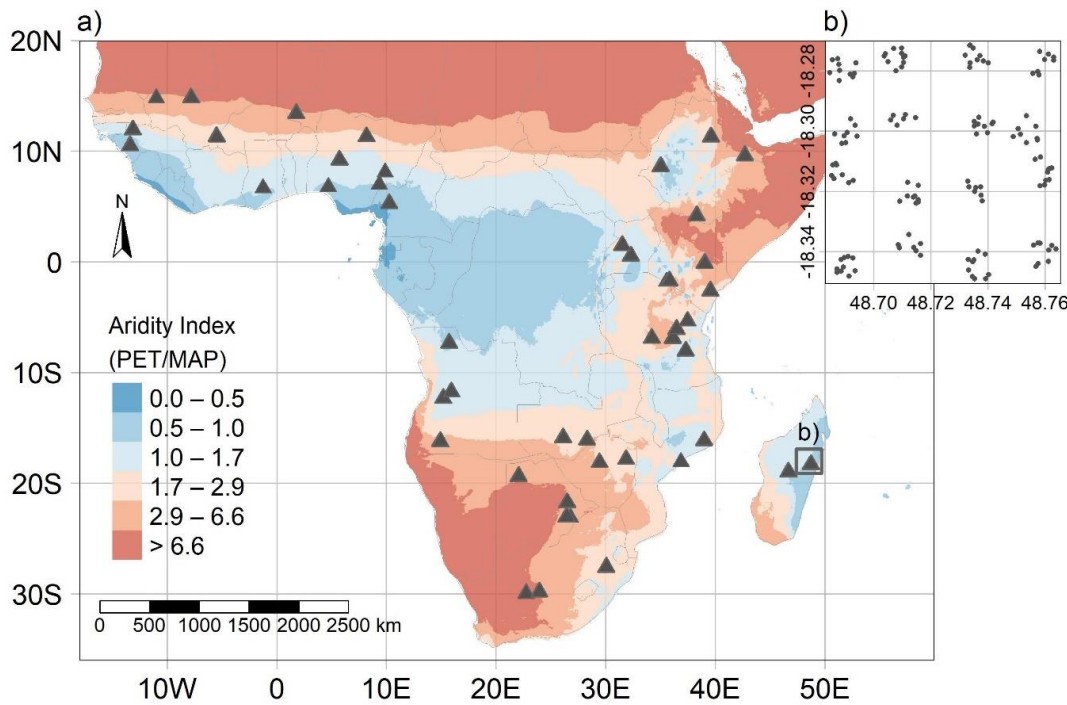

Figure 1: a) Aridity Index map and sampling scheme ($n_{total}$ = 1,601). Grey triangles represent individual sentinel sites where sample clusters were collected. The top-right inset (b) shows the exact sampling points within one of the sentinel sites (Didy, Madagascar) as an example.

## 2.2.    Sample and data processing

Soil material was air-dried and sieved to a particle size <2 mm in the Soil-Plant Spectroscopy Laboratory at the World Agroforestry Centre (ICRAF) in Nairobi, Kenya. All soil properties (except for soil texture, which was measured at ICRAF), were analyzed at Rothamsted Research in Harpenden, U.K.

Data for soil organic carbon (SOC; wt-%), $pH_{H20}$, amorphous oxalate-extractable aluminum ($Al_{ox}$, wt-%) and iron ($Fe_{ox}$, wt-%), exchangeable calcium ($Ca_{ex}$, cmol$^+$ kg$^{-1}$), clay + fine silt content (<8 µm, %), and total element concentrations (in wt-%) of Al, Ca, K, and Na, were selected in order to cover a wide range of soil properties that have been identified to relate to SOC stabilization mechanisms (Oades, 1988; Rasmussen et al., 2018), while maximizing the number of samples and minimizing the correlation among variables included in our analysis.

SOC was calculated from the difference of total C and inorganic C. The latter was directly measured with a Primacs AIC100 analyzer (Skalar Analytical B.V., Breda, Netherlands) by treating the sample with phosphoric acid and heating it to 135 °C in a closed system. Inorganic C in the sample was converted to $CO_2$ and then measured by Non-Dispersive Infrared Detection (NDIR). Total C was determined with the TruMac Total N and C combustion analyzer (Leco, St. Joseph, Michigan, USA). Soil $pH_{H20}$ was performed in a 1:2.5 soil:water suspension. The extraction of Al and Fe with oxalic acid and ammonium oxalate solution was done by shaking the solution for 4 h at 25 °C in the dark. Carbonate-rich samples were pre-treated with ammonium

acetate at pH 5.5 to remove any $CaCO_3$. Acid-oxalate extraction in particular dissolves short range-order minerals such as ferrihydrite (Fe), allophane and imogolite (Al), as well as other amorphous and organic Fe and Al minerals (Parfitt and Childs, 1988). Hexamine-cobalt trichloride solution was used as extractant to determine $Ca_{ex}$. Aqua regia acid digestion was applied for major and trace elements, including Al, Ca, K and Na. Although this method does not give absolute total contents, it does give results sufficiently close to accepted values for different soils (McGrath and Cunliffe, 1985). Samples were digested in tubes in time and temperature-controlled heating blocks. All elements were measured with ICP-OES (Optima 7300 DV, Perkin Elmer, Waltham, Massachusetts, USA). Particle size distribution was measured using a Laser Diffraction Particle Size Analyzer (LDPSA) Model LA 950 (Horiba, Kyoto, Japan). Each sample was shaken for 4 min in a 1% sodium hexametaphosphate (calgon) solution with ultrasonic energy before measuring to disperse aggregates. We used 8 µm as cut-off to capture all clay and fine silt particles. Results were comparable to <20 µm (see SI material Figure A1), but <8 µm was selected because it is more relevant to our interest in studying the influence of smaller particles with large surface area on SOC concentration. In addition, particles <8 µm resulted in a reproducible fraction across soil types, unlike using only clay particles <2 µm (Figure A1). Aluminum, Ca, K and Na concentrations were used to calculate the chemical index of alteration (CIA) after Nesbit and Young (1982), using the following equation:

$$CIA = Al_2O_3 / (Al_2O_3 + CaO + K_2O + Na_2O) * 100 \quad (2)$$

where CaO is the amount incorporated in the silicate fraction. Correction is necessary for samples that contain carbonates and apatite (Nesbit and Young, 1982). We adopted an approach introduced by McLennan (1993): The correction assumes that Ca is typically lost more rapidly than Na during weathering. If a soil sample contained inorganic C ($C_{total} - C_{org}$; used as a proxy for carbonates and apatite) and the CaO content was greater than that of $Na_2O$ in the same sample (n = 476), then the CaO concentration was set to that of $Na_2O$ from the same sample (Malick and Ishiga, 2016). After applying the correction, no obvious correlation remained between CIA and inorganic C (Figure A2). The index increases (i.e. more highly-weathered soil) with the loss of $Ca^{2+}$, $K^+$, and $Na^+$.

Samples were removed that contained missing or negative values for one or more of the above-mentioned parameters. In addition, a single sample with extraordinarily high SOC content (>22 wt-%) was excluded. This resulted in a total of 1,601 soil samples (out of the original 2,002 samples) at 45 sentinel sites across 17 countries. Note that due to the sample selection, not all profiles had data from both topsoil and subsoil layers (Table B1).

The remaining soil samples (n = 1,601) were paired (based on longitude and latitude at the profile level) with mean annual temperature (MAT, °C) and mean annual precipitation (MAP, mm) from the *WorldClim* data set at 30 sec resolution (Fick and Hijmans, 2017). Potential annual evapotranspiration (PET, mm) was added from Trabucco and Zomer (2019), who calculated it after the Penman-Monteith method, based on the WorldClim data. Mean annual precipitation and PET were used to calculate an annual aridity index, defined as PET/MAP (Budyko, 1974). Values >1 indicate water-limited (dry) regions and ratios <1

point to energy-limited (wet) regions. For the monthly aridity index, we used monthly climate data at the same spatial resolution and from the same data sources.

Land-cover data was used from the collected field data. The land-cover groups were re-classified into four major groups: a) Cropland (including all cultivated plots), b) Forest, c) Grassland and d) Other (including mainly woodland, shrubland, and bushland, but also samples classified as other). Ten missing values were gap-filled from a prototype high resolution Africa land-cover map at 20 m resolution based on one-year of Sentinel-2A observations from December 2015 to December 2016 (http://2016africalandcover20m.esrin.esa.int/).

Due to the lack of precise data products for lithology and soil types in sub-Saharan Africa, we did not include these variables in our analyses. Soils at AfSIS sites (Figure 1) developed mainly from two parent material types: i) metamorphic and ii) volcanic rocks (Hartmann and Moosdorf, 2012; Jones et al., 2013; Schlüter, 2008), likely modified throughout the Quaternary. i) Metamorphic rocks are most commonly found in West Africa, Southern Africa and Madagascar. These regions are characterized by old cratons, except for Madagascar, which is influenced by Mesozoic volcanism (Schlüter, 2008). Most of these soils are classified as Ferralsols (WRB soil classification system; Jones et al., 2013). Related AfSIS soils from those regions are usually highly weathered with low $pH_{H2O}$ values. In contrast, soils derived from ii) volcanic rocks are mainly found in the East African Rift System. They are usually younger and less weathered (Buringh, 1970). Beyond the influence of volcanic rocks, $Ca^{2+}$ rich soils are frequent in East Africa.

## 2.3.    Statistical analyses

We used three different statistical approaches, including linear mixed-effects models, regression trees and random forests to determine geochemical and climatic parameters that best explain SOC variation across sub-Saharan Africa. In brief, we used linear mixed-effects models to handle the hierarchal sampling design of the AfSIS data set, whereas regression trees and random forests enabled us to account for non-linearities within the data. More precisely, we used regression trees as a qualitative tool to explore and understand the structure of the data, whereas random forests offered more generalizable models. All statistical analyses were performed within the R computing environment (Version 4.0.0, R Core Team, 2020). The R Markdown file in the SI provides the code to reproduce all our analyses.

Linear mixed-effects modeling was performed by using the *nlme* R package (Pinheiro et al., 2020) to account for the nested sampling scheme (clusters within sites and two sampling depths within one profile). This allows the intercept of the regression to vary for each site, for each cluster within the same site, and for each sample within the same profile (Harrison et al., 2018). The variance inflation factor was used to check for multi-collinearity among predictor variables with a threshold of <3.0 (Zuur et al., 2010). To meet linear mixed-effects model assumptions and to standardize variation among variables, all continuous parameters were transformed to a normal distribution using Box-Cox transformation, followed by standardization to a mean of 0 and standard deviation of 1 by using the R package *bestNormalize* (Peterson and Cavanaugh, 2019). The relationship between SOC and the predictors of the original data may not be linear.

To answer our 1st research question, which soil properties and climate parameters best explain SOC content, we started from a constant null model with siteID/clusterID/plotID as random effects and then extended the model step-wise by fitting the following sequence of fixed effects: MAT, PET/MAP, depth, land cover, clay and fine silt, $pH_{H20}$, CIA, $M_{ox}$ ($Al_{ox}$ + ½$Fe_{ox}$), $Ca_{ex}$, $pH_{H2O}$*$M_{ox}$. The order and selection of fixed effects was pre-defined based on *a-priori* knowledge out of a larger set of variables (Burnham and Anderson, 2002), starting with large-scale climate variables and ending with fine scale physiochemical soil properties. The oxalate-extractable metals $Al_{ox}$ and $Fe_{ox}$ were summed to $M_{ox}$ ($Al_{ox}$ + ½$Fe_{ox}$) to normalize the atomic mass difference between Al and Fe (Wagai et al., 2020) and to account for their similar behavior over their concentration range (Figure 5b). The maximum likelihood method and likelihood ratio tests (L.ratio) were applied to evaluate model performance and the statistical significance of the added fixed effects (Table B4-B9). The variation explained by each fixed effect was obtained by calculating the marginal $R^2$ (excluding the variation explained by the random effects siteID/clusterID/plotID) for each model and subtracting the $R^2$ from the previous fitted model using the function *r.squaredGLMM* from the *MuMIn* R package (Barton, 2020; Nakagawa and Schielzeth, 2013). To identify how much SOC variation is explained by climate and geochemistry only (Legendre and Legendre, 2012), we built one model with climate parameters (MAT, PET/MAP) only, and one model with geochemistry variables (clay and fine silt, $pH_{H20}$, CIA, $M_{ox}$, $Ca_{ex}$, $pH_{H2O}$*$M_{ox}$) only. In addition, we analyzed the two sampling depths (0–20 and 30–50 cm) separately to determine whether the same factors are important for topsoil versus the deeper soil layer (Table 1). For this model, we did not include plotID as a random effect since each profile only contained one sample in each depth model.

For the 2nd research question, how geochemical controls on SOC content vary between environmentally distinct sub-regions, we grouped the data based on a) $pH_{H2O}$, b) wetness, c) weathering, and d) land cover (Table 1). Soil $pH_{H2O}$ and weathering data were grouped with the number of categories chosen to maximize and equalize the number of samples in each category and to correspond with common $pH_{H2O}$ and weathering groups (Nesbit and Young, 1982). In order to take seasonality of the sites into account separately, the data were divided into three categories based on the number of wet months (i.e. months with P/PET > 1). Land cover was grouped based on the four pre-defined categories. For each category within each sub-group, we built a linear mixed-effects model as previously described, yet only included the geochemical properties (clay + fine silt, $pH_{H2O}$, CIA, $M_{ox}$, $Ca_{ex}$, $pH_{H2O}$*$M_{ox}$) as fixed effects, since we intended to test if the importance of these predictors changed between environmentally distinct sub-regions (Table 1). When CIA or $pH_{H2O}$ were used to create the categories, they were not included as a fixed effect in the corresponding sub-models.

**Table 1: Grouping variables, sub-groups, number of samples and fixed effects used for the linear mixed-effects models**

| Groups | Categories | n | Fixed effects |
|---|---|---|---|
| All samples | None | 1,601 | All, Climate, Geochemistry |
| Depth | Topsoil (0–20 cm) | 791 | Geochemistry |
| | Subsoil (30–50 cm) | 810 | |
| $pH_{H2O}$ | Strongly acidic (3.9–5.2 $pH_{H2O}$) | 404 | Geochemistry |
| | Moderately acidic (5.2–6.1 $pH_{H2O}$) | 399 | |
| | Neutral (6.1–7.5 $pH_{H2O}$) | 398 | |
| | Alkaline (7.5–9.9 $pH_{H2O}$) | 400 | |
| Wetness (Number of wet months (P/PET > 1)) | 0 wet months | 572 | Geochemistry |
| | 1–3 wet months | 367 | |
| | 4–7 wet months | 662 | |
| Weathering (CIA) | Moderate (10–88% CIA) | 801 | Geochemistry |
| | High (88–100% CIA) | 800 | |
| Land cover | Cropland | 429 | Geochemistry |
| | Forest | 228 | |
| | Grassland | 242 | |
| | Other | 702 | |

P: Monthly precipitation [mm]; PET: Monthly potential evapotranspiration [mm]; CIA: Chemical Index of Alteration [%]; Fixed effects: All (Mean annual precipitation (MAT), Aridity index (PET/MAP), depth, land cover, clay and fine silt, $pH_{H2O}$, CIA, oxalate-extractable metals ($M_{ox}$), exchangeable Ca ($Ca_{ex}$), $pH_{H2O}*M_{ox}$), Climate (MAT, PET/MAP), Geochemistry (clay and fine silt, $pH_{H2O}$, CIA, $M_{ox}$, $Ca_{ex}$, $pH_{H2O}*M_{ox}$)

Regression tree (R packages: *rpart* and *rpart.plot;* Milborrow, 2019; Therneau and Atkinson, 2019) and random forest analyses (R packages: *ranger;* Wright and Ziegler, 2017) were conducted to identify non-linear relationships between SOC and any explanatory variable. This also enabled the identification of pedogenic thresholds within the data. Each analysis was conducted with the same explanatory variables as for the linear mixed-effects models. However, no data transformation was needed due to the non-linearity of the models.

Regression tree analysis was applied to obtain an easily interpretable and non-linear model for the entire data set and for both depth layers (topsoil vs subsoil) that best describes the existing data (Breiman et al., 1984). Since regression trees are known to easily overfit data, we used a grid search to prune the model (Boehmke and Greenwell, 2020) according to the minimum number of data points required to attempt a split, and the maximum number of internal nodes between the root node and terminal nodes in order to minimize the cross-validation error (Breiman et al., 1984). The overall performance of the regression tree analysis was tested using five-fold spatial cross-validation (R package: *mlr;* Bischl et al., 2016). Spatial partitioning was used to split the data into five disjoint subsets, using the coordinates from each sample, and repeating the partitioning 100

times (Figure A3). This results in a bias-reduced assessment of model performance (Brenning, 2012; Lovelace et al., 2019).

Absolute values at the bottom of each node indicate the predicted SOC content [wt-%] and the percentage corresponds to the relative number of samples in this node (Figure A5).

Random forest was used to build more generalized models since it is an ensemble of multiple decorrelated trees. Tuning of the model hyperparameters was done based on spatial tuning (R package: *mlr;* Bischl et al., 2016; Lovelace et al., 2019). These hyperparameters included the number of predictors used at each split, the minimum number of observations in a terminal node

and the fraction of samples used in each tree (Probst et al., 2019). The best hyperparameter combination search was done for the complete data set via a five-fold spatial cross-validation with one repetition. In each of these five spatial partitions, we ran 50 models to find the optimal hyperparameter combination (Lovelace et al., 2019).

Partial dependence plots were used to further explore the relationship between the predicted SOC content and the explanatory variables of the tuned random forest models (R package: *pdp;* Greenwell, 2017). These plots were used to investigate the

255 marginal effect of individual explanatory variables (such as $Al_{ox}$, $Ca_{ex}$, etc) on the predicted SOC content (Friedman, 2001). This allowed us to identify thresholds within the data and provided an indication of how important each explanatory variable was to predict SOC concentration across specific value ranges.

# 3. Results

 ## 3.1. Data distribution across sub-Saharan Africa

All soil and climate variables spanned at least one order of magnitude (except MAT and PET), demonstrating the diversity of this continent-wide data set. Based on skewness, kurtosis, histograms, and Shapiro-Wilk-tests (data not shown for the latter two), no variable was normally distributed (Table 2).

**Table 2: Summary statistics of all numerical soil and climate variables for the entire data set ($n_{total} = 1,601$; $n_{Topsoil} = 791$; $n_{Subsoil} = 810$)**

| Variable | Mean | SD | P0 | P25 | P50 | P75 | P100 | Skewness | Kurtosis |
|---|---|---|---|---|---|---|---|---|---|
| SOC [wt-%] | 1.84 | 1.51 | 0.07 | 0.65 | 1.42 | 2.54 | 9.19 | 1.42 | 2.23 |
| MAT [°C] | 21.7 | 3.2 | 13.7 | 19.8 | 21.5 | 23.0 | 29.8 | 0.17 | -0.12 |
| MAP [mm] | 1070 | 487 | 255 | 648 | 1057 | 1432 | 2708 | 0.29 | -0.63 |
| PET [mm] | 1810 | 310 | 1350 | 1571 | 1759 | 1933 | 2949 | 1.19 | 1.96 |
| PET/MAP | 2.35 | 1.73 | 0.71 | 1.2 | 1.54 | 3.16 | 9.54 | 1.46 | 1.31 |
| Clay + fine silt [%] | 55.4 | 22.6 | 0.1 | 37.7 | 57.9 | 74.7 | 100.0 | -0.26 | -1.00 |
| $Al_{ox}$ [wt-%] | 0.28 | 0.36 | 0.01 | 0.12 | 0.20 | 0.29 | 3.71 | 4.52 | 25.29 |
| $Fe_{ox}$ [wt-%] | 0.38 | 0.56 | 0.01 | 0.10 | 0.21 | 0.40 | 4.46 | 3.60 | 14.96 |
| $Ca_{ex}$ [cmol$^+$ kg$^{-1}$] | 10.29 | 11.01 | 0.03 | 1.34 | 5.86 | 16.49 | 75.66 | 1.28 | 1.32 |
| $pH_{H2O}$ | 6.3 | 1.3 | 3.9 | 5.2 | 6.1 | 7.5 | 9.9 | 0.27 | -1.11 |
| CIA [%] | 87.7 | 9.3 | 10.3 | 81.7 | 88.1 | 96.0 | 99.9 | -1.04 | 3.88 |

 SD: Standard deviation; P: Percentile; SOC: Soil organic carbon; MAT: Mean annual temperature; MAP: Mean annual precipitation; PET: Potential evapotranspiration; $Al_{ox}$: Oxalate-extractable Al; $Fe_{ox}$: Oxalate-extractable Fe; $Ca_{ex}$: Exchangeable Ca; CIA: Chemical Index of Alteration

In total, 429 samples were classified as cropland, 228 as forest, 242 as grassland and 702 as other land covers, including mainly shrubland, bushland and woodland. The SOC content decreased among those groups in the following sequence: Forest (2.69 ± 1.15 wt-%) > Cropland (2.21 ± 1.68 wt-%) > Grassland (1.77 ± 1.55 wt-%) > Other (1.35 ± 1.28 wt-%; Figure 2a). Clay + fine silt content and SOC showed a positive relationship across the entire data set, yet with a large spread (Figure 2b). However, individual sites showed contrasting correlations between SOC and clay + fine silt content – including, none, positive, and negative values (Figure 2c; Figure A4 for all individual sites).

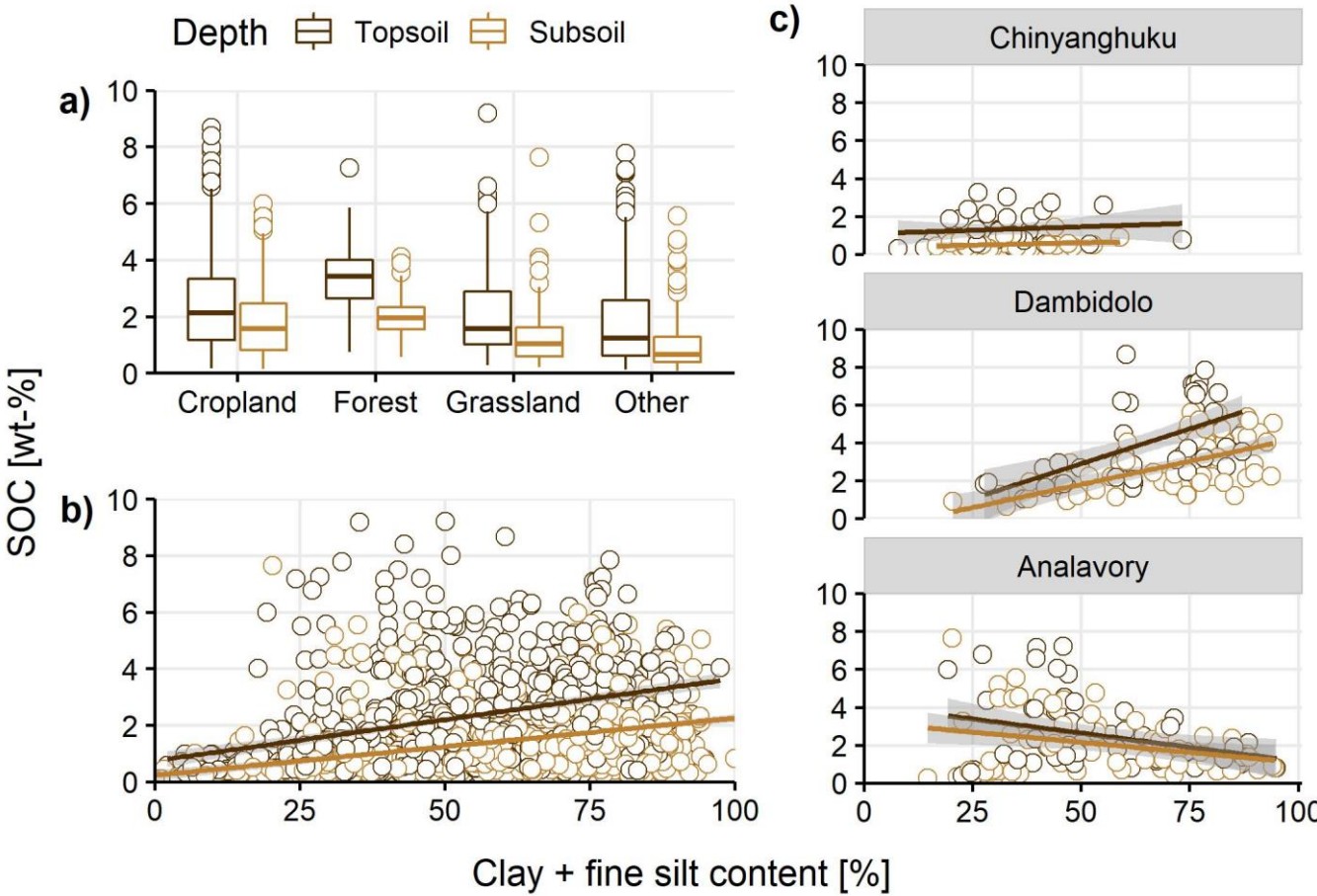

Figure 2: a) Soil organic carbon (SOC) content [wt-%] for the different land-covers (cropland, forest, grassland, other (bushland, shrubland, woodland) by depth (topsoil: 0–20 cm, subsoil: 20–50 cm); b) SOC [wt-%] and clay + fine silt content (<8 µm) [%] by depth; c) SOC [wt-%] and clay + fine silt content (<8 µm) [%] by depth for three example sites that show contrasting trends. Gray area around fitted linear regressions (y~x, for illustration only) in b) and c) show the 95% confidence interval. For the relationship between SOC [wt-%] and clay + fine silt content (<8 µm) [%] for all individual sites, see Figure A4.

## 3.2.    Predictors of soil organic carbon

*Linear mixed-effects modelling*

The full linear-mixed effects model for the entire data set had a marginal $R^2$ of 0.72. The two climate parameters (MAT, PET/MAP), depth, $M_{ox}$ and $Ca_{ex}$ were the most important predictors of SOC content, based on their marginal $R^2$. Land cover, clay + fine silt, $pH_{H2O}$, CIA and $pH_{H2O}$*$M_{ox}$ contributed little or not at all to the overall explanatory power of the model. Clay + fine silt, $M_{ox}$ and $Ca_{ex}$ were positively correlated with SOC, whereas all other fixed effects showed negative relationships with SOC concentration. The negative coefficient for depth indicates that the SOC content in the subsoil layers is on average lower as compared with the topsoil samples (Figure 3a).

| Model | MAT | PET/MAP | Depth | Land cover | Clay + fine silt | $pH_{H2O}$ | CIA | $M_{ox}$ | $Ca_{ex}$ | pH*$M_{ox}$ |
|---|---|---|---|---|---|---|---|---|---|---|
| Full (R² = 0.72) | **0.17 (-)** | **0.30 (-)** | **0.05 (-)** | 0.01 (+) | **0.04 (+)** | 0.00 (-) | **0.00 (-)** | **0.08 (+)** | **0.05 (+)** | **0.01 (-)** |
| Geochemistry (R² = 0.46) | – | – | – | – | **0.01 (-)** | 0.00 (-) | **0.04 (-)** | **0.26 (+)** | **0.11 (+)** | **0.04 (-)** |
| Climate (R² = 0.48) | **0.17 (-)** | **0.30 (-)** | – | – | – | – | – | – | – | – |

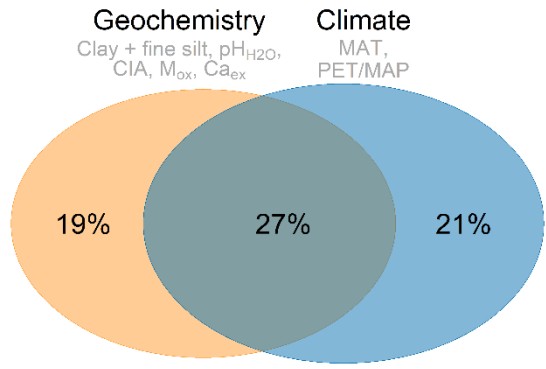

Geochemistry + Climate: 67%

Figure 3: a) Marginal R² for each predictor based on sequential fitting of the linear mixed-effects models of all samples ($n_{Total}$ = 1,601) for the full, geochemistry-only and climate-only model. Sign in parentheses refers to the correlation between the predictors and soil organic carbon. Bold values have a p-value < 0.05 based on likelihood-ratio test; b) Venn-Diagram illustrating the independent-explained and shared-explained variations by the geochemistry-only and the climate-only linear mixed-effects models.

The marginal R² for the geochemistry model was 0.46; almost the same as for the climate model (R² = 0.48). For the geochemistry model, the contribution of $M_{ox}$ and $Ca_{ex}$ to explain SOC content was much higher than in the full model (Figure 3a). Based on variation partitioning, 27% of the explained variation is shared between the geochemistry model and the climate model, whereas the variation explained by the geochemical or climate variables alone is 19% and 21%, respectively (Figure 3b).

Differences between the predictors were negligible for the two depth models (topsoil vs subsoil). However, the explained variation by clay + fine silt was larger in the subsoil layers compared with the topsoil layers. For $Ca_{ex}$, the opposite was true (Figure 4a).

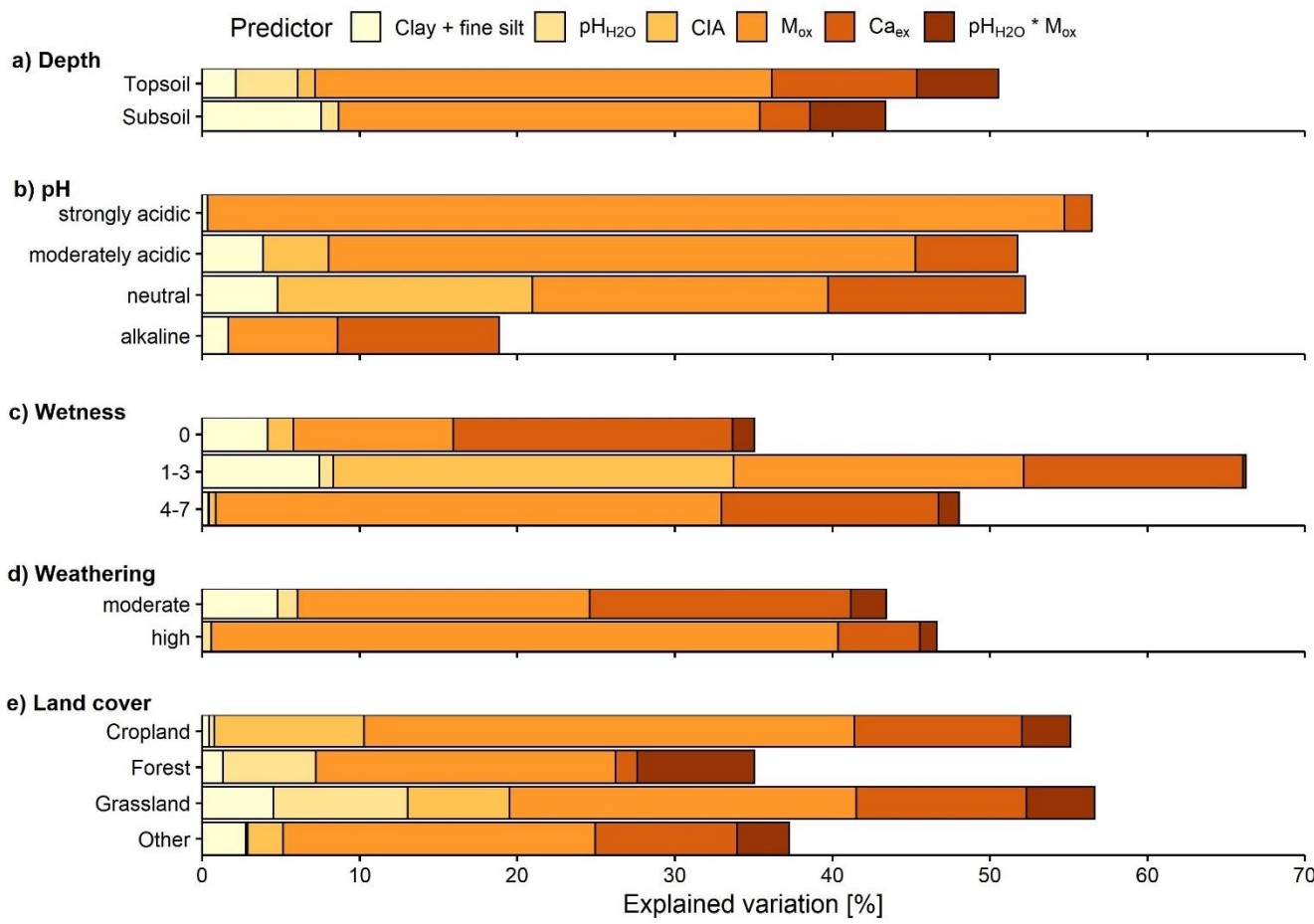

**Figure 4: Explained variation (based on marginal R²) for each fixed effect, based on sequential fitting of the linear mixed-effects models grouped by a) Depth (topsoil: 0–20 cm; subsoil: 20–50 cm); b) pH classes (strongly acidic: 3.9–5.2 pH, moderately acidic: 5.2–6.1, neutral: 6.1–7.5, alkaline: 7.5–9.9); c) Wetness (number of wet months (P/PET > 0); 0, 1–3, 4-7); d) Weathering (CIA: Chemical Index of alteration: moderate: 10–88% CIA, high: 88–100%) d) land cover.**

Within the $pH_{H2O}$ sub-models, $M_{ox}$ was most important in the strongly acidic model. The opposite was observed for $Ca_{ex}$ (Figure 4b), which corresponds to higher concentrations of $Ca_{ex}$ in neutral and alkaline soils compared with moderately and strongly acidic soils. However, $Ca_{ex}$ was also found to have a positive relationship with SOC in acidic soils (Figure 5; Table B2). The direction of the correlation between clay + fine silt and SOC concentration was not consistent across the four pH groups, in contrast to the other fixed effects (Table B2). The alkaline sub-model had the lowest marginal R² of all $pH_{H2O}$ sub-models, which suggests that important predictors were missing (Figure 4b).

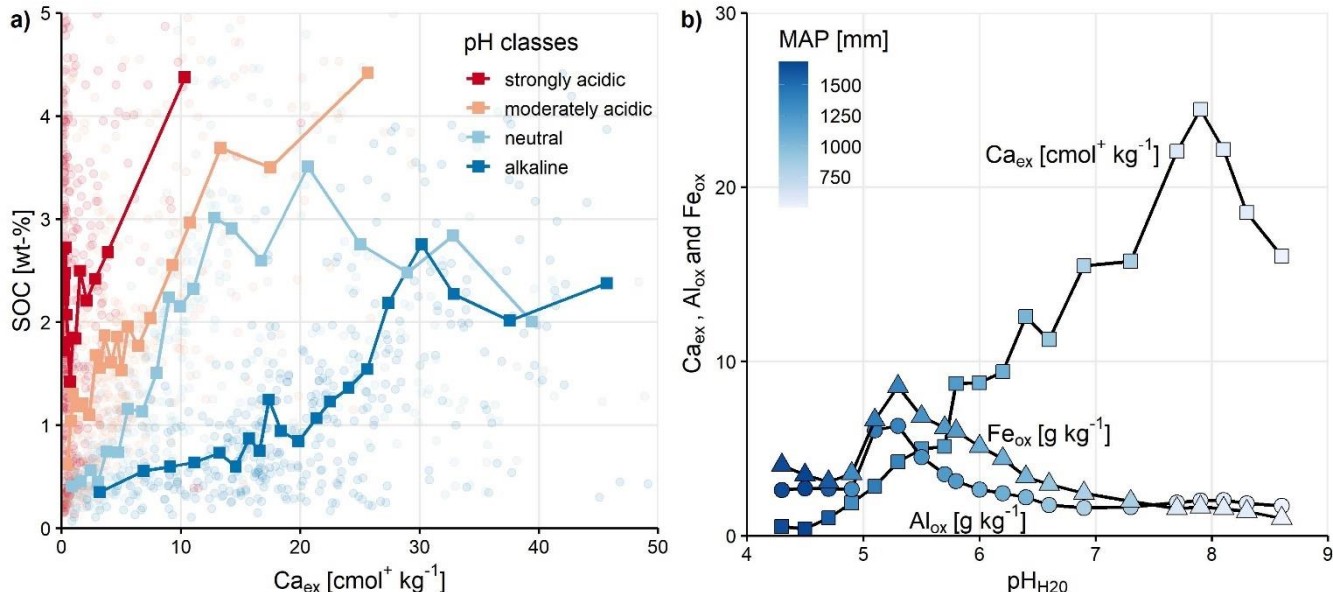

Figure 5: a) Soil organic carbon (SOC) [wt-%] and exchangeable Ca (Ca$_{ex}$) [cmol$^+$ kg$^{-1}$] content colored by pH classes (strongly acidic: 3.9–5.2 pH, moderately acidic: 5.2–6.1, neutral: 6.1–7.5, alkaline: 7.5–9.9) with a moving average (bold squares; n = 20). Note that x-axis is truncated for improved visualization, which removes 3 data points (Ca$_{ex}$ = 53.91, 54.58, and 75.66 cmol$^+$ kg$^{-1}$); b) Al$_{ox}$, Fe$_{ox}$ [g kg$^{-1}$] (which were combined to M$_{ox}$ (Al$_{ox}$ + ½Fe$_{ox}$) for the linear mixed effects models) and Ca$_{ex}$ [cmol$^+$ kg$^{-1}$] averaged content (n = 20) across pH$_{H2O}$ and mean annual precipitation (MAP) [mm].

Grouping by the number of wet months (wetness) showed that M$_{ox}$ explained most of the variation in wet regions, whereas Ca$_{ex}$ was most important in drier regions (Figure 4c). This corresponds to the overall distribution of M$_{ox}$ and Ca$_{ex}$ across MAP and pH$_{H2O}$ (Figure 5b). The chemical index of alteration (CIA) explained most of the variation in the intermediate wet regions (Figure 4c).

The high weathering model was dominated by M$_{ox}$, whereas the importance of M$_{ox}$ and Ca$_{ex}$ in the moderate weathering model was similar. The other fixed effects did not explain much of the variation of the two weathering models (Figure 4d).

Within the land cover models, the Cropland and grassland models had the highest marginal R² and were both dominated by M$_{ox}$. The variation explained by Ca$_{ex}$ was smallest for the forest model, whereas it did not change much for the other three models (Figure 4e).

In summary, in the linear mixed-effects models, M$_{ox}$ was more important in wetter regions, acidic and highly weathered soils, whereas Ca$_{ex}$ was more important in drier regions, alkaline and less weathered soils. The other fixed effects usually did not explain much of the SOC variation.

### Regression tree and random forest

The root mean squared error (RMSE) for the topsoil regression tree was 1.47 wt-% (range: 0.80–3.11 wt-%) and for the subsoil regression tree was 0.67 wt-% (range: 0.44–2.26 wt-%); the relative RMSEs were 0.65% and 0.48%, respectively. In the topsoil regression tree (Figure A5a) Fe$_{ox}$, MAT and PET/MAP were the most important predictors to split and explain variation

in SOC concentration. About 23% of the SOC data could be explained by $Fe_{ox}$ and MAT alone. In general, higher $Fe_{ox}$, $Al_{ox}$ and $Ca_{ex}$ values resulted in higher SOC content. This was equally true for the subsoil tree (Figure A5b). While much of the SOC variation was explained by climate parameters in topsoils, the subsoil regression tree was more dominated by geochemical variables, namely $Fe_{ox}$ and $Al_{ox}$. About 40% of the subsoil SOC variation could be explained by $Fe_{ox}$ only. In both trees, clay + fine silt content and land cover poorly predicted SOC.

In summary, topsoil and subsoil regression trees contained the same predictors, yet with climate variables playing a larger role in the topsoil regression tree and geochemistry having a larger influence in the subsoil regression tree. Overall, the results showed that the explanatory variables did not differ much between the depth intervals (topsoil vs subsoil), while their magnitude did.

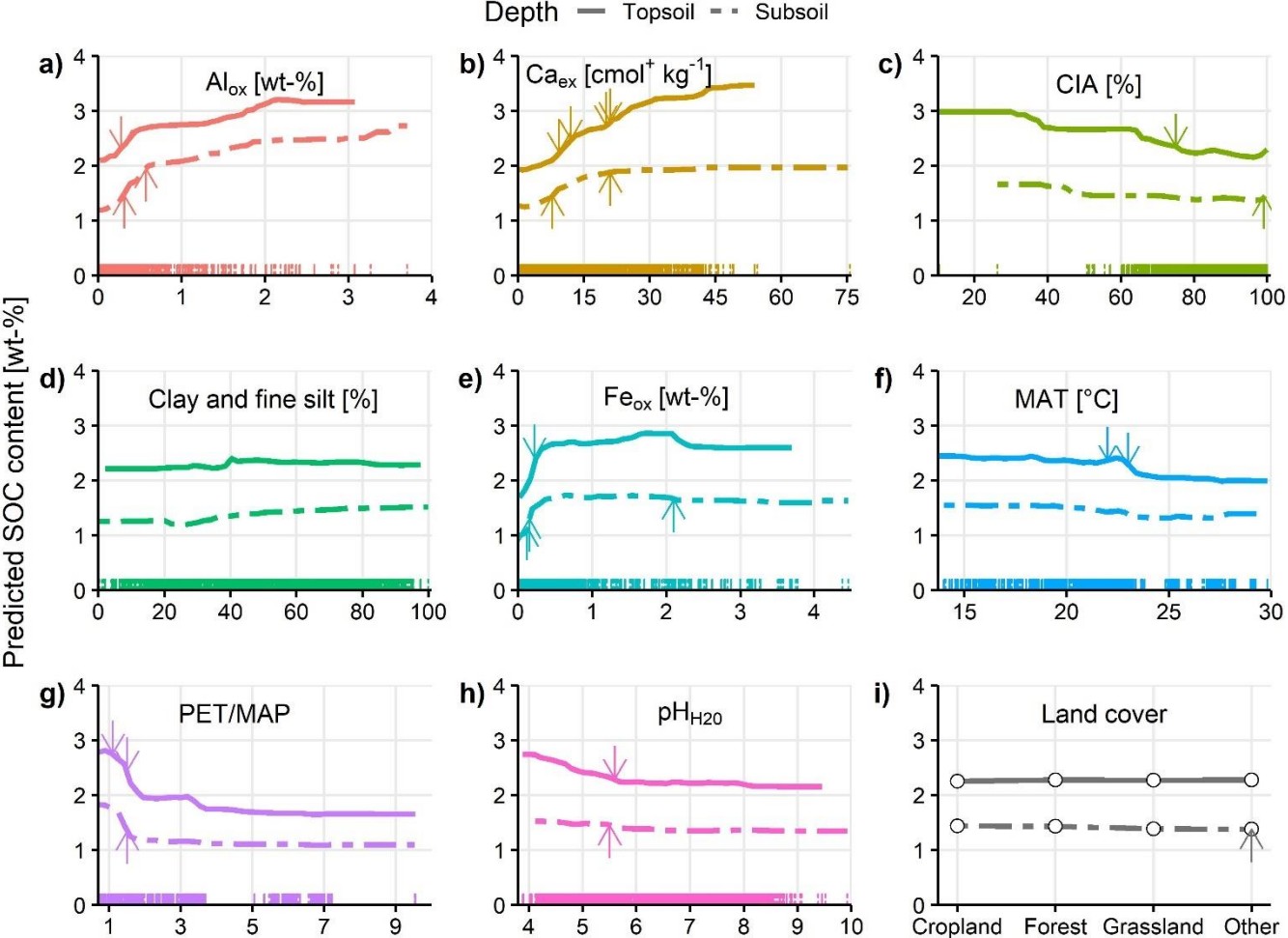

**Figure 6: Partial dependence plot for each explanatory variable of the random forest models (topsoil and subsoil). X-axes always correspond to the range of the explanatory variable. Arrows indicate splitting points in the regression tree (Figure A5). Each colored tick mark along the x-axes represents one sample.**

The random forest models had a RMSE of 1.31 wt-% and a R² of 0.70 for the topsoil samples, and for the subsoil samples a RMSE of 0.87 wt-% and a R² of 0.72. Based on the partial dependence plots (Figure 6), $Al_{ox}$ and $Ca_{ex}$ were important in

predicting SOC over the entire range of each variable (Figure 6a and b). However, in subsoils, the predictive power of $Ca_{ex}$ was reduced (Figure 6b). We observed a decrease in predicted SOC with increasing soil weathering status (CIA). However, due to the low number of samples with CIA values below 60%, the relationship should be interpreted with caution in this range (Figure 6c). Clay + fine silt content had almost no effect on SOC, with only a weak positive trend in subsoil samples (Figure 6d). The relationship between $Fe_{ox}$ concentration and predicted SOC content varied with $Fe_{ox}$ concentration. At low

concentrations (< 0.25 wt-%), there was a strong positive relationship between predicted SOC content and $Fe_{ox}$. For higher concentrations, the predicted SOC content was relatively constant (Figure 6e). MAT correlated negatively over the entire range with predicted SOC concentration (Figure 6f). For PET/MAP, the predicted SOC content declined sharply as PET/MAP increased from 1 to 2 (transition from wet to dry water regimes; Figure 6g). The relationship between $pH_{H2O}$ and predicted SOC content was not strong (Figure 6h). For land cover, there was almost no difference between the classes within the same

depth layer; however, topsoils had higher SOC content (2.2 wt-%) compared with the subsoil samples across all land covers (1.5 wt-%; Figure 6i).

## 4. Discussion

Climate and geochemical variables are similarly important in explaining SOC variations across sub-Saharan Africa (Figure 3); in line with findings from a global study (Luo et al., 2021). However, the explanatory power of climate and geochemical

variables is not independent of each other, reflecting the overall strong interaction between climate and geochemistry (Doetterl et al., 2015). Since it is likely that, in the long term, climate variables have predominantly indirect effects on SOC dynamics through their influence on soil geochemistry, we focus our discussion on those geochemical variables ($Ca_{ex}$, $Al_{ox}$ and $Fe_{ox}$) that showed the highest explanatory power with respect to SOC content across all models. In addition, we discuss the role of depth, clay + fine silt content, and land cover in explaining SOC variations on a continental scale, since these were identified by other

studies to play an important role in SOC dynamics.

### *Exchangeable Calcium*

Strong and positive relationships emerged between $Ca_{ex}$ and SOC concentration across all models, even though $Ca_{ex}$ concentration showed strong $pH_{H20}$ and precipitation dependence (Figure 5). Typical $Ca^{2+}$ sources in soils are from a) weathering of bedrock or surface rock formations, b) decomposition of $Ca^{2+}$-rich organic materials, c) lateral movement of

$Ca^{2+}$-rich water, d) atmospheric dust and rain deposition or e) anthropogenic inputs (Likens et al., 1998; Rowley et al., 2018). Characteristically, $Ca^{2+}$ is weathered easily from both primary and secondary minerals (Likens et al., 1998). This usually leads to its accumulation in semi-arid to arid environments that are characterized by low rates of water flow through the soil profile that drives slow weathering rates and high $pH_{H2O}$ values (Figure 4b-d). In such environments, $Ca^{2+}$ plays an important role as

a cation bridge that facilitates aggregate formation (Rimmer and Greenland, 1976; Tisdall and Oades, 1982) and bonding of
clay minerals to organic matter functional groups because of their divalent charge, relative abundance and modest hydration radius (Likens et al., 1998; Muneer and Oades, 1989). However, we found that $Ca_{ex}$ was not only important in alkaline and less-weathered soils in dry regions, but also in acidic and more-weathered soils under wetter conditions (Figure 5). It is likely that the main $Ca^{2+}$ source in those regions derives from atmospheric deposition (Albani et al., 2015; Goudie and Middleton, 2001) and/or biological cycling by plants (Likens et al., 1998). This is supported by the fact that $Ca_{ex}$ showed a stronger
relationship with SOC in topsoil than subsoil layers (Figure 4a and 6b). Since land cover, which is a major driver of C inputs into the soil, did not show a strong relationship with SOC in the models, we speculate that biological cycling of $Ca^{2+}$ does not play a major role in explaining the observed differences in SOC content. Yet, further analysis with better proxies for biological $Ca^{2+}$ inputs is needed to test this hypothesis. High $Ca^{2+}$ concentrations in acidic soils can also be derived from the development of those soils from $Ca^{2+}$-rich parent material which are out-of-equilibrium with modern climate conditions (Slessarev et al.,
2016).

     In conclusion, the important role of $Ca_{ex}$ in our data set was most pronounced in dry regions, dominated by alkaline and less weathered soils. However, it also played a role in explaining the SOC variation in wetter regions and more acidic soils, which is supporting the overall importance of $Ca_{ex}$ in stabilizing SOC.

### *Oxalate extractable Al and Fe*

Similar to $Ca_{ex}$, short range-order minerals ($M_{ox}$, $Al_{ox}$ and $Fe_{ox}$) showed a positive and strong correlation with SOC content across all models. The relationship was strongest in wet regimes, acidic and highly weathered soils (Figure 4b-d and 5b). Hydrous oxides of Al and Fe are usually highly reactive because of their large specific areas with a high proportion of reactive sites (Parfitt and Childs, 1988). This results in the adsorption of organic matter to Fe and Al oxides and the formation of stable soil aggregates (Tisdall and Oades, 1982). In humid regions, high rates of mineral weathering may release Fe, Al and Si faster
than crystalline minerals can precipitate (Rasmussen et al., 2018). Therefore, $Fe_{ox}$ and $Al_{ox}$ are usually found to be important in SOC stabilization in humid and acidic soils (Eusterhues et al., 2003; Kramer and Chadwick, 2018).

     In our study, short range-order minerals were also identified to play an important role for SOC stabilization in soils of sub-Saharan Africa. However, even though $Al_{ox}$ and $Fe_{ox}$ showed similar trends in their concentrations (Figure 5b), we observed diverging behavior in their predictive power of SOC in the regression trees (Figure A5) and the random forests (Figure 6a and
6e). For example, $Fe_{ox}$ was one of the most important explanatory variables in the regression tree and partial dependence plots, although only within a very narrow range and at low $Fe_{ox}$ concentrations (Figure 6e), whereas $Al_{ox}$ was important over the entire range (Figure 6a). Inagaki et al. (2020) showed that higher amounts of soil organic matter were co-localized with Fe in drier regions compared to sites with higher rainfall, whereas the content of $Al_{ox}$ co-localized with organic matter was not affected by precipitation changes. This may be linked to the different oxidation levels of Fe. At higher precipitation levels, Fe
oxides can be reduced, resulting in a release of associated SOC to the aqueous phase (Berhe et al., 2012; Chen et al., 2020; Thompson et al., 2011). This mechanism is probably responsible for the low correlation between SOC and high $Fe_{ox}$

concentrations in our data (Figure 6e), pointing to the fact that $Fe_{ox}$ can act as pedogenic threshold, depending on its oxidation level in the soil system.

In summary, short range order minerals also play an important role in SOC stabilization across sub-Saharan Africa, similar to other regions. However, $Al_{ox}$ and $Fe_{ox}$ do behave differently in explaining SOC content, even though they showed covariance in terms of their concentrations. Since we only have data for acid-oxalate extraction, we cannot speculate further about their diverging behavior in the models.

*Depth*

For the depth models, predictor differences were small between topsoil (0–20 cm) and subsoil (20–50 cm) samples (Figure 4a and 6). This may reflect the large depth increments for each of the two sampling depths, which may also explain the overall small explanatory power of depth in the linear-mixed effects model (Figure 3a). Since the identified SOC-controlling factors were similar for both depth layers (Figure 4a), differences in SOC content were likely driven by the fact that subsoil samples usually contain less SOC due to lower C inputs at greater depth (Jobbágy and Jackson, 2000). Soil erosion at some sites (data not shown) might also dilute differences between the two depth layers, since water and wind can permanently remove surface soil.

*Clay + fine silt content*

Clay + fine silt content (<8 µm) did not emerge as an important predictor of SOC concentration within our different models (Figure 3, 4 and 5e). This is in contrast to some earlier studies that indicated that total clay content explains a large proportion of SOC storage and stabilization due to the sorption of soil organic matter to surfaces of clay minerals and building of aggregates (Amelung et al., 1998; Kahle et al., 2002). The relationship between SOC and total clay content is used in various models to describe turnover and storage of SOC. However, this simplified correlation may not account for the different stabilization mechanisms related to various clay minerals, e.g. 1:1 vs 2:1 clay minerals (Oades, 1988). Past research has yielded contradictory results on whether clay content explains SOC variation in subtropical and tropical soils or not. For example, Bruun et al. (2010) showed for various tropical soils that clay mineralogy, $Fe_{ox}$ and $Al_{ox}$ are better explanatory variables for SOC content than clay content alone (<2 µm). In contrast, Quesada et al. (2020) found a strong relationship between clay and SOC content for highly weathered soils in the Amazon Basin that are dominated by 1:1 clay minerals such as kaolinite, whereas soils in the same system, dominated by 2:1 clay minerals, showed stronger relationship between SOC and Al species. In a comparison between tropical and temperate soils, Six et al. (2002b) found that less C was associated with the clay and silt fraction (<20 µm) in tropical soils than in temperate soils. Even though these studies used various cut-offs to define the clay (<2 µm), clay and fine silt (<8 µm), and clay and silt fraction (<20 µm), they all illustrate that the relationship with SOC can be complex in subtropical and tropical soils.

Due to the broad spatial scale, soils in the AfSIS data set contain different clay minerals (Butler et al., 2020). No clear relationship between clay + fine silt content (<8 µm) and SOC concentration was observed in the models, although the raw

data indicate an overall positive trend between clay + fine silt content (<8 µm) and SOC concentration (Figure 2b). This positive relationship does not hold across all sites (Figure 2c and A4). Variable relationships with SOC (Table B2) may explain the low predictive power of clay + fine silt content in this data set. Instead, variables that better capture the different behavior of clay-sized minerals, e.g. $Ca_{ex}$, $Fe_{ox}$ and $Al_{ox}$, are likely more suitable soil parameters to explain the variation of SOC content – even in highly weathered soils across sub-Saharan Africa. This is supported by the fact that a clay + fine silt-only model resulted in a very small $R^2$ (linear mixed-effects model: 0.01; random forest: 0.12; Table B3).

*Land cover*

The effect of land cover on SOC content was generally small in our models, even in topsoils (Figure 6i). Similar findings were recently encountered in a global study (Luo et al., 2021). One possibility may be that the relatively large 0–20 cm depth interval might dilute differences that could be more marked in the top few centimeters. However, we did observe differences in SOC content across land cover classes, with forests containing the highest amount of SOC – especially in topsoils (Figure 2a). Croplands had higher SOC content than grasslands, opposite of what is commonly observed in temperate regions (Prout et al., 2020).

Another possible explanation for the absence of land cover as an important predictor in our models, is that we lacked the detailed data necessary to disentangle impacts of different practices and land-use history. The land cover class cropland contained a wide variety of cultivated plots while more detailed information about land management practices were missing. This is particularly important since prior research in other regions showed that SOC stock changes in tropical cropland soils may be driven by C inputs (Fujisaki et al., 2018b). Additionally, historical land use may even play a more important role in explaining current stocks compared to recent land use (Vågen et al., 2006).

Furthermore, land cover may covary with other parameters (temperature, precipitation, geochemistry) to such a degree that it is not an explanatory variable. This might be the reason why the sub-models grouped by land cover did not show a clear pattern (Figure 4e). However, the land cover-only models resulted in small $R^2$ (linear mixed-effects models: 0.01; random forest: 0.10–0.16) which suggests that land cover is a poor predictor for our SOC data at this large spatial scale (Table B3). This may be due to the high variation of SOC content within the different land cover classes (Figure 2a). Land use changes and their impact on soil physico-chemical properties are scale-dependent and likely to be more distinct at smaller scales (Holmes et al., 2004,2005). For example, land management and land degradation (i.e. erosion) are known to impact SOC stocks in regionals scales in sub-Saharan Africa (Winowiecki et al., 2016a).

Future studies are needed to better understand the impacts of land management and carbon storage potential in soils across sub-Saharan Africa at different scales (Fujisaki et al., 2018a; Vanlauwe et al., 2015). Overall, our data for sub-Saharan Africa suggests that SOC content on a continental scale is better explained by stabilization potential in soils (climate, geochemistry) than by different aboveground C inputs (vegetation).

## 5. Conclusions

We used a continental-scale data set from sub-Saharan Africa to test relationships between SOC content, various soil properties and climate variables in order to address our core research questions:

*Which soil properties and climate parameters best explain SOC content variation across sub-Saharan Africa?*

Parameters similar to temperate regions are important to explain SOC variation for tropical and subtropical soils under various climate conditions across sub-Saharan Africa; namely $Ca_{ex}$, $M_{ox}$ ($Al_{ox}$ and $Fe_{ox}$), and PET/MAP. At this large spatial scale, climate and geochemical parameters are equally important and share some of the explained SOC variation. However, land cover and clay + fine silt content did not explain much of the variation in SOC content, in contrast to some findings from other regions and studies.

The selected climatic and geochemical parameters, which can be seen as proxies for most of the soil forming factors, explain about two thirds of SOC variation across sub-Saharan Africa. The remaining third likely reflects those soil forming factors that were not or only poorly represented within our selected variables, namely organisms, relief and time. Given the large spatial scale targeted, it appears unlikely to be able to explain all of the SOC variation measured.

*How do geochemical SOC-controlling factors vary between environmentally-distinct sub-regions?*

In dry regions with alkaline and less-weathered soils, $Ca_{ex}$ explained most of the SOC concentration variation, whereas $M_{ox}$ was more important in wetter regions with acidic and highly weathered soils. Still, $Ca_{ex}$ remained important in acidic and more weathered soils and in wetter regions. $Fe_{ox}$ as predictor of SOC content was only important at low concentrations in moderately weathered and wet soils. This observed trend leads to the assumption that $Fe_{ox}$ can play an important role in pedogenic thresholds in various soils across sub-Saharan Africa.

Overall, a combination of PET/MAP, $Ca_{ex}$ and $M_{ox}$ seems to be an appropriate set of variables to explain SOC-content variation on a continental scale across sub-Saharan Africa. This does not imply that other variables, such as clay + fine silt content and land cover are no good predictors on a regional scale as shown by previous studies. However, the variables identified by this study showed a consistent predictive power of SOC content across various climate regions.

Future studies on large-scale SOC stabilization should consider measuring those soil properties to include them in models.
This would likely improve the predictive capacity of these models and contribute to closing the gap between our theoretical understanding of SOC concentration across large scales and our ability to improve terrestrial biogeochemical projections that rely on existing models.

## Code availability

As a R markdown file (pdf) in the supplement materials.

## Data set availability

The soil properties data set used in this study is available from the authors upon reasonable request and under the following DOI: https://doi.org/10.34725/DVN/66BFOB (Vågen et al., 2021). Field data (i.e. land cover) for the sampling locations can be received from Vågen et al. (2013b). The climate data used (MAT, MAP and PET) can be downloaded from the sources cited: WorldClim:

Fick and Hijmans (2017) and Trabucco and Zomer (2019). Land-cover data used for gap-filling can be retrieved from http://2016africalandcover20m.esrin.esa.int/.

## Author contribution

Conceptualization of the study for this manuscript was done by SvF, AH, AAB, ST, and SD, with input from EA, SH, SMG, KS, JS, TGV and LW. Data curation, investigation and resources were done and provided by GA, EA, SH, SMG, KS, AS, ET, TGV, EW

and LW. The formal analysis, methodology, and visualization for the manuscript was performed by SvF with substantial input from AH, ML, SD and ST as well as feedback from all authors. SvF wrote the initial draft and all authors were involved in the review and editing of the manuscript.

## Competing interest

SD and AAB are liaison editors of the special issue *Tropical biogeochemistry of soils in the Congo Basin and the African Great*

*Lakes region* and JS is executive editor of the SOIL journal. However, none of them was involved in the review process of this manuscript. All other authors declare that they have no conflict of interest.

## Acknowledgement

SvF receives funding from the International Max-Planck Research School for Global Biogeochemical Cycles. ST and AH acknowledge support from the European Research Council (Horizon 2020 Research and Innovation Program, grant agreement 695101; 14Constraint).

SD receives supportive funds through DFG Emmy Noether Group "TropSOC" (project number: 387472333). The analytical data used in the study was produced by the "Chemical and Biological Assessment of AfSIS soils" project, funded by Biotechnology and Biological Sciences Research Council (BBSRC)/Global Challenges Research Fund (GCRF) (BBS/OS/GC/000014B). SPM and SH are partly funded by the Institute Strategic Program (ISP) grant "Soils to Nutrition" (S2N; grant number BBS/E/C/000I0310). Original field surveys and sample analysis costs at ICRAF were covered by the AfSIS Phase I project funded by the Bill and Melinda Gates Foundation Grant

Number 51353. SvF thanks Jörg Matschullat for proofreading earlier versions of the manuscript.

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

# Annex: Electronic supplement

## Appendix A – Figures

## Appendix B – Tables

**Appendix A – Figures**

The figures and tables on the next two pages do all belong to the same topic. They show the results for the different cut-offs we used to identify the best cut-off to be used for soil texture. We looked at and tested for <2 µm, <8 µm, and <20 µm. In the end we decided to use <8 µm because we wanted to stay as close as possible to <2 µm. However, we could not use <2 µm due to some reproducibility issued for duplicates. The differences between <8 µm and <20 µm are neglectable.

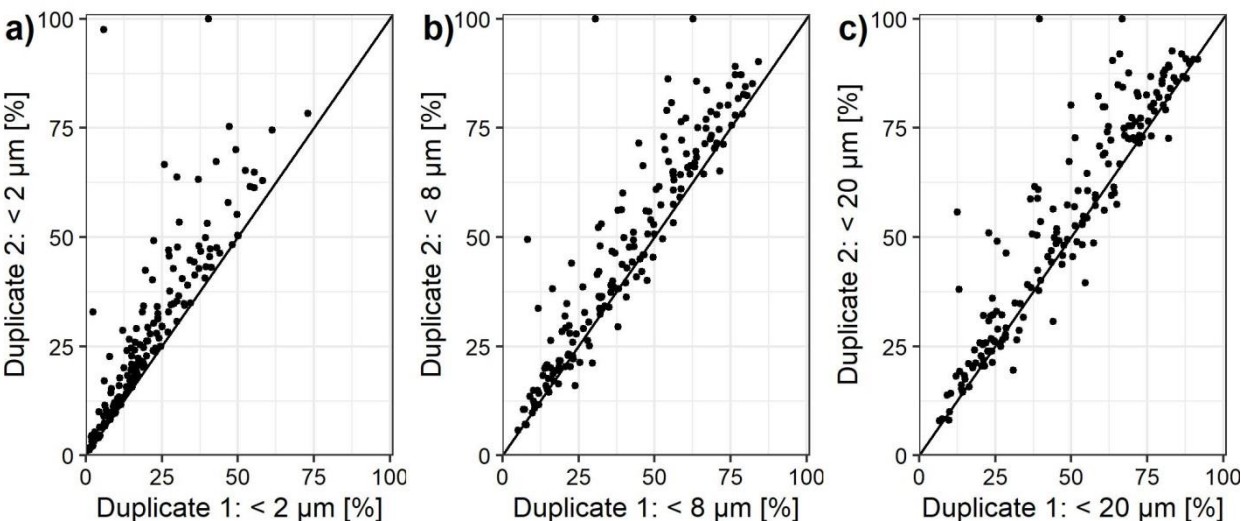

**Figure A1: Scatterplot of duplicate measurements for the particle size distribution data. a) Duplicate 1 and 2 <2µm; b) Duplicate 1 and 2 <8 µm; c) Duplicate 1 and 2 <20 µm**

**Table A1.1: Correlation coefficient between SOC and particle size data <8 µm and <20 µm for all samples (n = 1,601), topsoil (0–20 cm; n = 791), and subsoil (20–50 cm; n = 810)**

| Samples | <8 µm | <20 µm |
| --- | --- | --- |
| **All** | 0.32 | 0.41 |
| **Topsoil** | 0.37 | 0.46 |
| **Subsoil** | 0.43 | 0.49 |

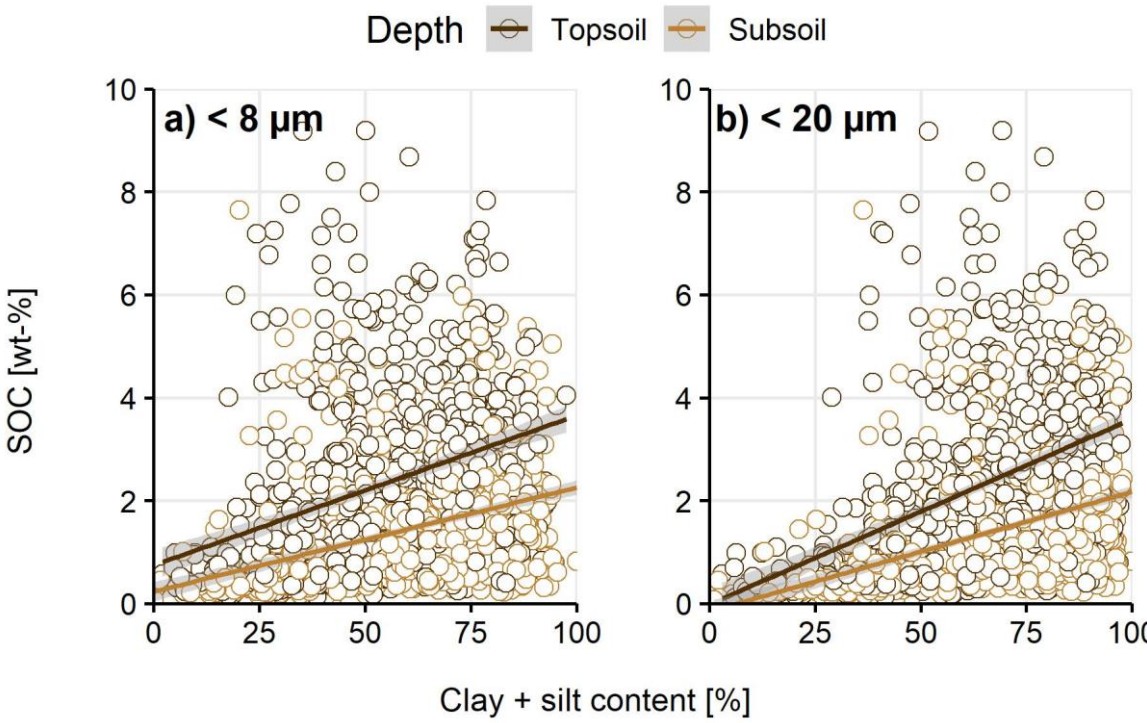

**Figure A1.2: a) Soil organic carbon (SOC) content [wt-%] and clay + fine silt content <8 μm [%] by depth; b) SOC content [wt-%]**
**clay + fine silt content <20 μm [%] by depth.**

**Table A1.3: Summary table of R² for the different models (linear mixed-effects model and random forest) for the two different**
**explanatory variables (<8 μm and <20 μm) for all samples (n = 1,601), topsoil (0–20 cm; n = 791), and subsoil (20–50 cm; n = 810)**

| Model | Linear mixed-effects model | Random forest (topsoil) | Random forest (subsoil) |
|---|---|---|---|
| **Clay + fine silt <8 μm** | 0.01 | 0.12 | 0.12 |
| **Clay + silt <20 μm** | 0.03 | 0.17 | 0.19 |

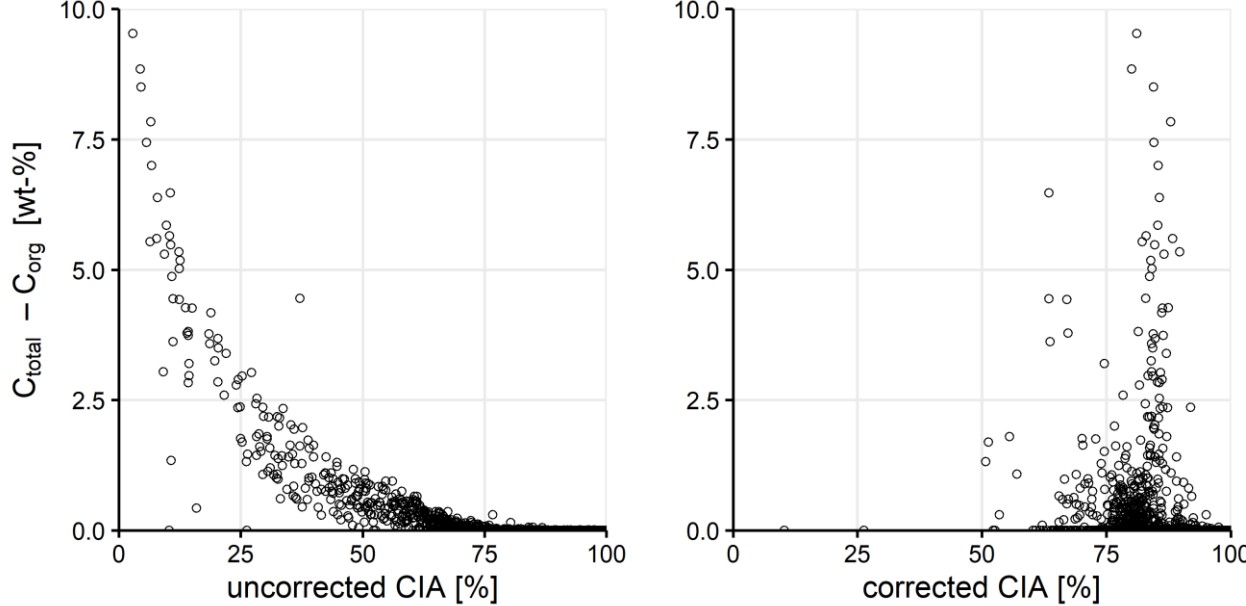

**Figure A2: Scatterplot of inorganic carbon ($C_{total} - C_{org}$ [wt-%]), the uncorrected chemical index of alteration (CIA [%]; left) and the CIA [%] correct for carbonates and apatite after Nesbit and Young (1982) (right). See *methods* for more details.**

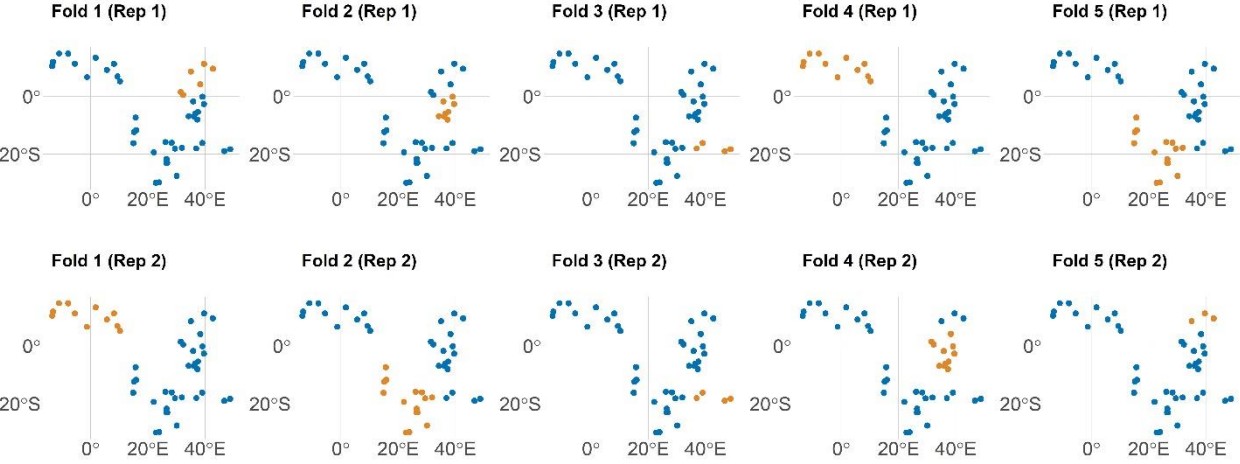

 **Figure A3: Spatial visualization of selected training (blue) and test (orange) observations for spatial cross-validation of two repetitions from the topsoil samples. Note: Each dot may represent multiple samples.**

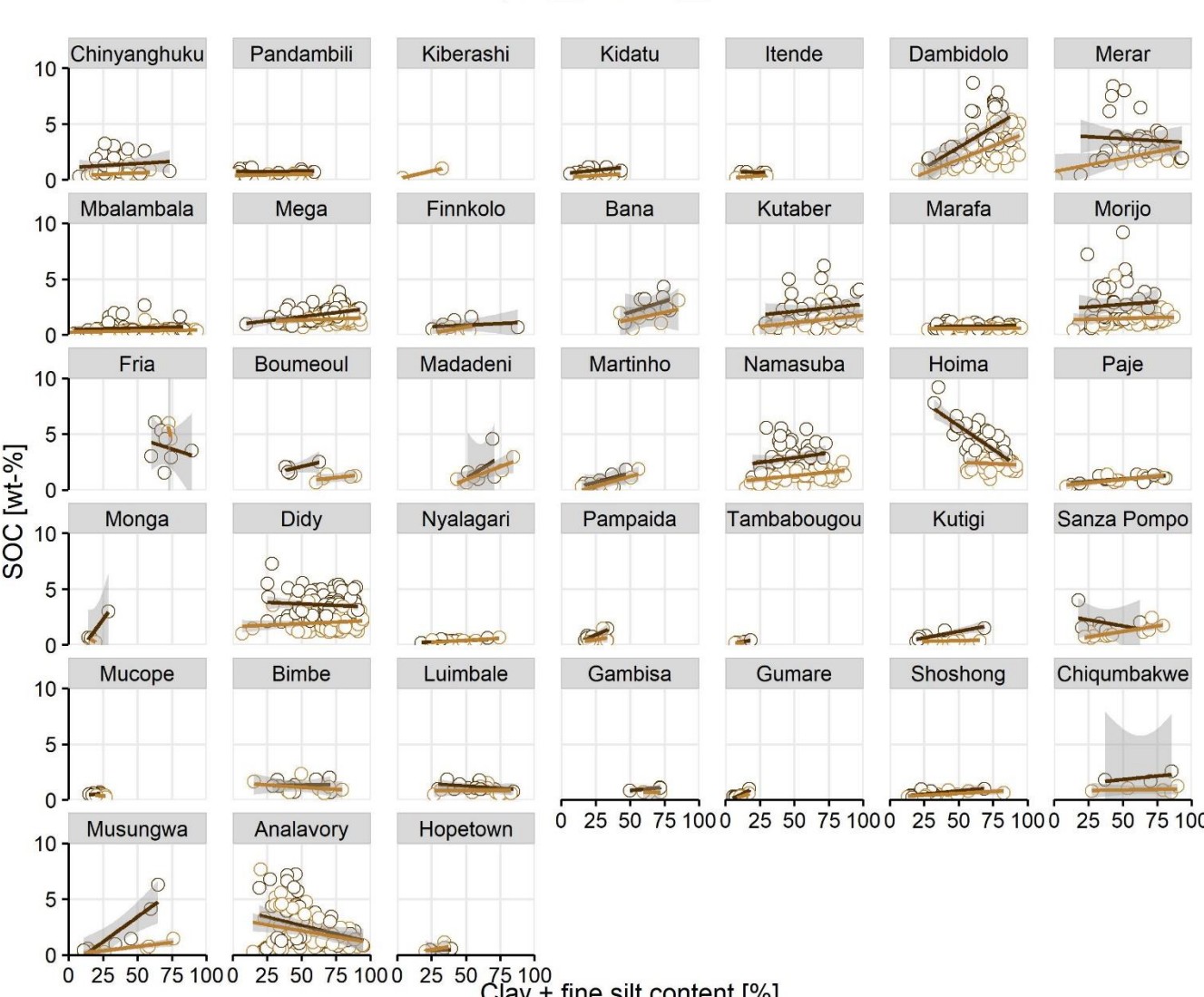

**Figure A4: Soil organic carbon (SOC) [wt-%] and clay + fine silt content [%] by depth for each sampling site that contained more than one sample per depth layer (topsoil: 0-20 cm, subsoil: 20-50 cm). Gray area around fitted linear regressions represent the 95% confidence interval.**

## a) Topsoil

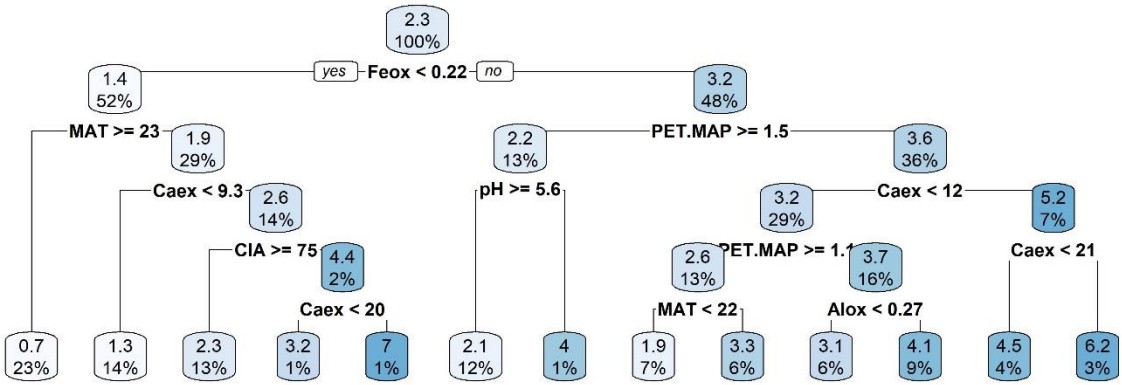

## b) Subsoil

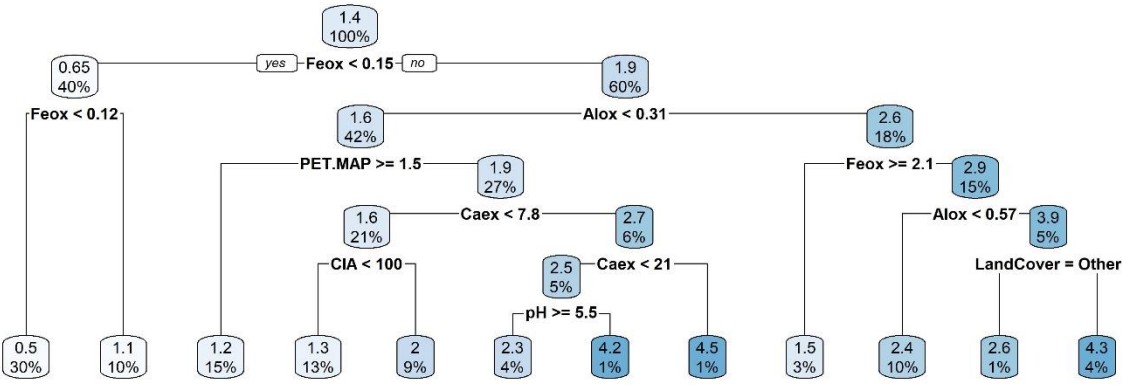

**Figure A5: Regression tree for a) Topsoil (0-20 cm) and b) Subsoil (20-50 cm). Splitting values are always in the units of the parameter used for the split (for units see Table 1). Absolute values in the boxes indicate the predicted soil organic carbon (SOC) content [wt-%]. The percentage corresponds to the relative number of samples.**

# Appendix B − Tables

**Table B1: Overview of sample distribution used in this study across geographical region, countries, sites, depths and land cover**

| Region | Country | Site | Depth | | Land cover | | | |
|--------|---------|------|---------|---------|--------|----------|-----------|-------|
| | | | Topsoil | Subsoil | Forest | Cropland | Grassland | Other |
| East | TZA | 5 | 61 | 54 | 6 | 16 | 13 | 80 |
| | ETH | 4 | 179 | 165 | 3 | 153 | 56 | 132 |
| | KEN | 3 | 131 | 153 | 5 | 4 | 55 | 220 |
| | UGA | 2 | 99 | 101 | 0 | 90 | 29 | 81 |
| | MDG | 2 | 161 | 175 | 206 | 86 | 20 | 24 |
| West | NGA | 5 | 16 | 19 | 1 | 15 | 5 | 14 |
| | MLI | 3 | 11 | 14 | 1 | 9 | 6 | 9 |
| | CMR | 1 | 8 | 6 | 2 | 10 | 2 | 0 |
| | GIN | 2 | 12 | 8 | 1 | 9 | 1 | 9 |
| | NER | 1 | 13 | 11 | 0 | 12 | 0 | 12 |
| | GHA | 1 | 1 | 0 | 1 | 0 | 0 | 0 |
| South | ZAF | 3 | 11 | 11 | 0 | 0 | 7 | 15 |
| | MOZ | 2 | 7 | 6 | 0 | 4 | 3 | 6 |
| | BWA | 3 | 29 | 26 | 0 | 2 | 11 | 42 |
| | ZMB | 2 | 10 | 9 | 1 | 2 | 13 | 3 |
| | AGO | 4 | 36 | 44 | 1 | 14 | 17 | 48 |
| | ZWE | 2 | 6 | 8 | 0 | 3 | 4 | 7 |

TZA: Tanzania; ETH: Ethiopia; KEN: Kenya; UGA: Uganda; MDG: Madagascar; NGA: Nigeria; MLI: Mali; CMR: Cameroon; GIN: Guinea; NER: Niger; GHA: Ghana; ZAF: South Africa; MOZ: Mozambique; BWA: Botswana; ZMB: Zambia; AGO: Angola; ZWE: Zimbabwe

**Table B2: Marginal R² for each fixed effect based on sequential fitting of the linear mixed-effects models for the different sub-models (depth, pH classes, number of wet months, weathering, land cover). Sign in brackets refers to the correlation between the fixed effect and soil organic carbon, respectively. Bold values have a p-value < 0.0001 based on likelihood-ratio test**

| Sub-model | | Clay + fine silt | $pH_{H2O}$ | CIA | $M_{ox}$ | $Ca_{ex}$ | $pH_{H2O}*M_{ox}$ |
|---|---|---|---|---|---|---|---|
| Depth | Topsoil | **0.02 (-)** | 0.04 (-) | **0.01 (-)** | **0.29 (+)** | **0.09 (+)** | **0.05 (-)** |
| | Subsoil | **0.08 (+)** | 0.01 (-) | 0.00 (-) | **0.27 (+)** | **0.03 (+)** | **0.05 (-)** |
| pH classes | Strongly acid | 0.00 (-) | – | 0.00 (-) | **0.54 (+)** | **0.02 (+)** | – |
| | Moderately acid | 0.04 (-) | – | **0.04 (-)** | **0.37 (+)** | **0.06 (+)** | – |
| | Neutral | 0.05 (+) | – | **0.16 (-)** | **0.19 (+)** | **0.13 (+)** | – |
| | Alkaline | 0.02 (-) | – | 0.00 (-) | **0.07 (+)** | **0.10 (+)** | – |
| Number of wet months | 0 | **0.04 (-)** | 0.00 (-) | 0.02 (-) | **0.10 (+)** | **0.18 (+)** | 0.01 (-) |
| | 1-3 | **0.07 (-)** | 0.01 (-) | **0.25 (-)** | **0.18 (+)** | **0.14 (+)** | **0.00 (-)** |
| | 4-7 | 0.00 (-) | 0.00 (-) | 0.00 (-) | **0.32 (+)** | **0.14 (+)** | **0.01 (-)** |
| Weathering | Moderate | **0.05 (-)** | 0.01 (-) | – | **0.19 (+)** | **0.17 (+)** | **0.02 (-)** |
| | High | 0.00 (-) | 0.01 (-) | – | **0.40 (+)** | **0.05 (+)** | **0.01 (+)** |
| Land cover | Cropland | 0.00 (-) | 0.00 (-) | **0.09 (-)** | **0.31 (+)** | **0.11 (+)** | **0.03 (-)** |
| | Forest | 0.01 (-) | 0.06 (-) | 0.00 (-) | **0.19 (+)** | **0.01 (+)** | 0.07 (-) |
| | Grassland | 0.05 (-) | **0.09 (-)** | **0.06 (-)** | **0.22 (+)** | **0.11 (+)** | **0.04 (-)** |
| | Other | **0.03 (-)** | 0.00 (-) | **0.02 (-)** | **0.20 (+)** | **0.09 (+)** | 0.03 (-) |

CIA: Chemical Index of Alteration, $M_{ox}$: Oxalate-extractable metals ($Al_{ox} + \frac{1}{2}Fe_{ox}$)

**Table B3: Summary table of R² for the different models (linear mixed-effects model and random forest) with different explanatory variables (clay and fine silt, land-cover, clay and fine silt + land-cover, full) included for the entire data set. The R² in brackets for the linear-mixed-effects models refer to the conditional R² which include the variation explained by the random effects (siteID/clusterID/plotID).**

| Model | Linear-mixed model | Random forest (topsoil) | Random forest (subsoil) |
|---|---|---|---|
| Clay + fine silt | 0.01 (0.72) | 0.12 | 0.12 |
| Land cover | 0.01 (0.75) | 0.10 | 0.16 |
| Clay + fine silt and land cover | 0.02 (0.72) | 0.22 | 0.26 |
| full | 0.71 (0.94) | 0.70 | 0.72 |

**Table B4:** Anova summary for linear mixed-effects analyses with the entire data set (n = 1,601) including all predictors, geochemistry-only and climate-only predictors. Fixed effects were step-wise added. The first entry (~1) refers to the constant null model, respectively.

| | df | AIC | BIC | logLik | Test | L.Ratio | p-value |
|---|---|---|---|---|---|---|---|
| **All predictors** | | | | | | | |
| ~1 | 5 | 2,993.22 | 3,020.11 | -1,491.61 | NA | NA | NA |
| MAT | 6 | 2,969.00 | 3,001.27 | -1,478.50 | 1 vs 2 | 26.23 | <0.0001 |
| …+ PET/MAP | 7 | 2,932.50 | 2,970.15 | -1,459.25 | 2 vs 3 | 38.50 | <0.0001 |
| … + Depth | 8 | 2,414.21 | 2,457.24 | -1,199.11 | 3 vs 4 | 520.29 | <0.0001 |
| … + Land cover | 11 | 2,416.06 | 2,475.22 | -1,197.03 | 4 vs 5 | 4.15 | 0.2454 |
| … + Clay + fine silt | 12 | 2,340.40 | 2,404.94 | -1,158.20 | 5 vs 6 | 77.65 | <0.0001 |
| … + $pH_{H2O}$ | 13 | 2,342.00 | 2,411.92 | -1,158.00 | 6 vs 7 | 0.40 | 0.5281 |
| … + CIA | 14 | 2,248.88 | 2,324.18 | -1,110.44 | 7 vs 8 | 95.13 | <0.0001 |
| … + $M_{ox}$ | 15 | 1,915.32 | 1,995.99 | -942.66 | 8 vs 9 | 335.56 | <0.0001 |
| … + $Ca_{ex}$ | 16 | 1,678.09 | 1,764.14 | -823.04 | 9 vs 10 | 239.23 | <0.0001 |
| … + $pH_{H2O}*M_{ox}$ | 17 | 1,599.15 | 1,690.59 | -782.58 | 10 vs 11 | 80.93 | <0.0001 |
| **Geochemistry-only** | | | | | | | |
| ~1 | 5 | 2,993.22 | 3,020.11 | -1,491.61 | NA | NA | NA |
| Clay + fine silt | 6 | 2,979.20 | 3,011.47 | -1,483.60 | 1 vs 2 | 16.03 | 0.0001 |
| … + $pH_{H2O}$ | 7 | 2,980.12 | 3,017.77 | -1,483.06 | 2 vs 3 | 1.07 | 0.3000 |
| … + CIA | 8 | 2,882.13 | 2,925.16 | -1,433.07 | 3 vs 4 | 99.99 | <0.0001 |
| … + $M_{ox}$ | 9 | 2,515.81 | 2,564.22 | -1,248.91 | 4 vs 5 | 368.32 | <0.0001 |
| … + $Ca_{ex}$ | 10 | 2,249.95 | 2,303.73 | -1,114.97 | 5 vs 6 | 267.86 | <0.0001 |
| … + $pH_{H2O}*M_{ox}$ | 11 | 2,170.66 | 2,229.82 | -1,074.33 | 6 vs 7 | 81.29 | <0.0001 |
| **Climate-only** | | | | | | | |
| ~1 | 5 | 2,993.22 | 3,020.11 | -1,491.61 | NA | NA | NA |
| MAT | 6 | 2,969.00 | 3,001.27 | -1,478.50 | 1 vs 2 | 26.23 | <0.0001 |
| … + PET/MAP | 7 | 2,932.50 | 2,970.15 | -1,459.25 | 2 vs 3 | 38.50 | <0.0001 |

MAT: Mean annual temperature; PET: Potential evapotranspiration; MAP: Mean annual precipitation; CIA: Chemical index of alteration; $M_{ox}$: Oxalate-extractable metals ($Al_{ox}$ + ½$Fe_{ox}$); Caex: Exchangeable calcium.

 **Table B5: Anova summary for linear mixed-effects grouped by depth ($n_{Topsoil} = 791$, $n_{Subsoil} = 810$). Fixed effects were step-wise added. The first entry (~1) refers to the constant null model, respectively.**

| | df | AIC | BIC | logLik | Test | L.Ratio | p-value |
|---|---|---|---|---|---|---|---|
| **Topsoil** | | | | | | | |
| ~1 | 4 | 1,440.72 | 1,459.42 | -716.36 | NA | NA | NA |
| Clay + fine silt | 5 | 1,418.88 | 1,442.25 | -704.44 | 1 vs 2 | 23.84 | <0.0001 |
| … + $pH_{H2O}$ | 6 | 1,408.74 | 1,436.78 | -698.37 | 2 vs 3 | 12.14 | 0.0005 |
| … + CIA | 7 | 1,350.41 | 1,383.12 | -668.20 | 3 vs 4 | 60.33 | <0.0001 |
| … + $M_{ox}$ | 8 | 1,148.87 | 1,186.26 | -566.44 | 4 vs 5 | 203.54 | <0.0001 |
| … + $Ca_{ex}$ | 9 | 1,016.14 | 1,058.20 | -499.07 | 5 vs 6 | 134.73 | <0.0001 |
| … + $pH_{H2O}*M_{ox}$ | 10 | 967.11 | 1,013.84 | -473.55 | 6 vs 7 | 51.03 | <0.0001 |
| **Subsoil** | | | | | | | |
| ~1 | 4 | 1,460.72 | 1,479.51 | -726.36 | NA | NA | NA |
| Clay + fine silt | 5 | 1,373.42 | 1,396.91 | -681.71 | 1 vs 2 | 89.30 | <0.0001 |
| … + $pH_{H2O}$ | 6 | 1,372.98 | 1,401.16 | -680.49 | 2 vs 3 | 2.44 | 0.1180 |
| … + CIA | 7 | 1,373.42 | 1,406.30 | -679.71 | 3 vs 4 | 1.56 | 0.2123 |
| … + $M_{ox}$ | 8 | 1,188.60 | 1,226.18 | -586.30 | 4 vs 5 | 186.82 | <0.0001 |
| … + $Ca_{ex}$ | 9 | 1,135.71 | 1,177.99 | -558.86 | 5 vs 6 | 54.89 | <0.0001 |
| … + $pH_{H2O}*M_{ox}$ | 10 | 1,106.11 | 1,153.09 | -543.06 | 6 vs 7 | 31.60 | <0.0001 |

MAT: Mean annual temperature; PET: Potential evapotranspiration; MAP: Mean annual precipitation; CIA: Chemical index of alteration; $M_{ox}$: Oxalate-extractable metals ($Al_{ox} + \frac{1}{2}Fe_{ox}$); Caex: Exchangeable calcium.

**Table B6: Anova summary for linear mixed-effects grouped by $pH_{H2O}$ ($n_{strongly\ acidic}$ = 404, $n_{moderatly\ acidic}$ = 399, $n_{neutral}$ = 398, $n_{alkaline}$ = 400). Fixed effects were step-wise added. The first entry (~1) refers to the constant null model, respectively.**

| | df | AIC | BIC | logLik | Test | L.Ratio | p-value |
|---|---|---|---|---|---|---|---|
| **Strongly acidic (3.9-5.2 pH)** | | | | | | | |
| ~1 | 5 | 909.23 | 929.23 | -449.61 | NA | NA | NA |
| Clay + fine silt | 6 | 909.32 | 933.33 | -448.66 | 1 vs 2 | 1.91 | 0.1673 |
| … + CIA | 7 | 911.31 | 939.32 | -448.65 | 2 vs 3 | 0.01 | 0.9293 |
| … + $M_{ox}$ | 8 | 712.18 | 744.19 | -348.09 | 3 vs 4 | 201.13 | <0.0001 |
| … + $Ca_{ex}$ | 9 | 690.68 | 726.69 | -336.34 | 4 vs 5 | 23.51 | <0.0001 |
| **Moderately acidic (5.2-6.1 pH)** | | | | | | | |
| ~1 | 5 | 876.39 | 896.34 | -433.20 | NA | NA | NA |
| Clay + fine silt | 6 | 864.42 | 888.36 | -426.21 | 1 vs 2 | 13.97 | 0.0002 |
| … + CIA | 7 | 849.82 | 877.74 | -417.91 | 2 vs 3 | 16.60 | <0.0001 |
| … + $M_{ox}$ | 8 | 734.60 | 766.51 | -359.30 | 3 vs 4 | 117.22 | <0.0001 |
| … + $Ca_{ex}$ | 9 | 679.03 | 714.93 | -330.51 | 4 vs 5 | 57.57 | <0.0001 |
| **Neutral (6.1-7.5 pH)** | | | | | | | |
| ~1 | 5 | 785.87 | 805.80 | -387.93 | NA | NA | NA |
| Clay + fine silt | 6 | 772.22 | 796.14 | -380.11 | 1 vs 2 | 15.65 | 0.0001 |
| … + CIA | 7 | 686.06 | 713.97 | -336.03 | 2 vs 3 | 88.16 | <0.0001 |
| … + $M_{ox}$ | 8 | 620.16 | 652.06 | -302.08 | 3 vs 4 | 67.90 | <0.0001 |
| … + $Ca_{ex}$ | 9 | 581.03 | 616.91 | -281.52 | 4 vs 5 | 41.13 | <0.0001 |
| **Alkaline (7.5-9.9 pH)** | | | | | | | |
| ~1 | 5 | 688.71 | 708.67 | -339.36 | NA | NA | NA |
| Clay + fine silt | 6 | 679.07 | 703.02 | -333.53 | 1 vs 2 | 11.64 | 0.0006 |
| … + CIA | 7 | 681.04 | 708.98 | -333.52 | 2 vs 3 | 0.02 | 0.8765 |
| … + $M_{ox}$ | 8 | 651.45 | 683.38 | -317.72 | 3 vs 4 | 31.59 | <0.0001 |
| … + $Ca_{ex}$ | 9 | 592.58 | 628.51 | -287.29 | 4 vs 5 | 60.87 | <0.0001 |

MAT: Mean annual temperature; PET: Potential evapotranspiration; MAP: Mean annual precipitation; CIA: Chemical index of alteration; $M_{ox}$: Oxalate-extractable metals ($Al_{ox} + \frac{1}{2}Fe_{ox}$); Caex: Exchangeable calcium.

**Table B7: Anova summary for linear mixed-effects grouped by number of wet months (P/PET > 1; $n_0 = 572$, $n_{1-3} = 367$, $n_{4-7} = 662$). Fixed effects were step-wise added. The first entry (~1) refers to the constant null model, respectively.**

| | df | AIC | BIC | logLik | Test | L.Ratio | p-value |
|---|---|---|---|---|---|---|---|
| **0 number of wet months** | | | | | | | |
| ~1 | 5 | 1,016.28 | 1,038.03 | -503.14 | NA | NA | NA |
| Clay + fine silt | 6 | 989.89 | 1,015.98 | -488.94 | 1 vs 2 | 28.40 | <0.0001 |
| … + pH$_{H2O}$ | 7 | 990.41 | 1,020.85 | -488.20 | 2 vs 3 | 1.48 | 0.2245 |
| … + CIA | 8 | 980.65 | 1,015.44 | -482.32 | 3 vs 4 | 11.76 | 0.0006 |
| … + M$_{ox}$ | 9 | 934.82 | 973.96 | -458.41 | 4 vs 5 | 47.82 | <0.0001 |
| … + Ca$_{ex}$ | 10 | 840.40 | 883.89 | -410.20 | 5 vs 6 | 96.42 | <0.0001 |
| … + pH$_{H2O}$*M$_{ox}$ | 11 | 840.08 | 887.92 | -409.04 | 6 vs 7 | 2.33 | 0.1272 |
| **1-3 number of wet months** | | | | | | | |
| ~1 | 5 | 933.01 | 952.53 | -461.50 | NA | NA | NA |
| Clay + fine silt | 6 | 912.86 | 936.29 | -450.43 | 1 vs 2 | 22.15 | <0.0001 |
| … + pH$_{H2O}$ | 7 | 910.07 | 937.41 | -448.04 | 2 vs 3 | 4.79 | 0.0287 |
| … + CIA | 8 | 811.91 | 843.16 | -397.96 | 3 vs 4 | 100.16 | <0.0001 |
| … + M$_{ox}$ | 9 | 708.70 | 743.85 | -345.35 | 4 vs 5 | 105.21 | <0.0001 |
| … + Ca$_{ex}$ | 10 | 618.44 | 657.49 | -299.22 | 5 vs 6 | 92.26 | <0.0001 |
| … + pH$_{H2O}$*M$_{ox}$ | 11 | 599.70 | 642.66 | -288.85 | 6 vs 7 | 20.74 | <0.0001 |
| **4-7 number of wet months** | | | | | | | |
| ~1 | 5 | 1,489.12 | 1,511.60 | -739.56 | NA | NA | NA |
| Clay + fine silt | 6 | 1,487.46 | 1,514.44 | -737.73 | 1 vs 2 | 3.66 | 0.0558 |
| … + pH$_{H2O}$ | 7 | 1,488.86 | 1,520.32 | -737.43 | 2 vs 3 | 0.61 | 0.4355 |
| … + CIA | 8 | 1,486.23 | 1,522.19 | -735.11 | 3 vs 4 | 4.63 | 0.0315 |
| … + M$_{ox}$ | 9 | 1,339.02 | 1,379.48 | -660.51 | 4 vs 5 | 149.21 | <0.0001 |
| … + Ca$_{ex}$ | 10 | 1,256.20 | 1,301.15 | -618.10 | 5 vs 6 | 84.82 | <0.0001 |
| … + pH$_{H2O}$*M$_{ox}$ | 11 | 1,237.14 | 1,286.58 | -607.57 | 6 vs 7 | 21.06 | <0.0001 |

MAT: Mean annual temperature; PET: Potential evapotranspiration; MAP: Mean annual precipitation; CIA: Chemical index of alteration; M$_{ox}$: Oxalate-extractable metals (Al$_{ox}$ + ½Fe$_{ox}$); Caex: Exchangeable calcium.

**Table B8: Anova summary for linear mixed-effects grouped by weathering ($n_{moderate} = 801$, $n_{high} = 800$). Fixed effects were step-wise added. The first entry (~1) refers to the constant null model, respectively.**

| | df | AIC | BIC | logLik | Test | L.Ratio | p-value |
|---|---|---|---|---|---|---|---|
| **Moderate weathering (10-88% CIA)** | | | | | | | |
| ~1 | 5 | 1,535.35 | 1,558.78 | -762.67 | NA | NA | NA |
| Clay + fine silt | 6 | 1,495.43 | 1,523.54 | -741.71 | 1 vs 2 | 41.92 | <0.0001 |
| … + $pH_{H2O}$ | 7 | 1,487.13 | 1,519.93 | -736.56 | 2 vs 3 | 10.30 | 0.0013 |
| … + $M_{ox}$ | 8 | 1,352.69 | 1,390.18 | -668.35 | 3 vs 4 | 136.44 | <0.0001 |
| … + $Ca_{ex}$ | 9 | 1,169.17 | 1,211.35 | -575.59 | 4 vs 5 | 185.52 | <0.0001 |
| … + $pH_{H2O}*M_{ox}$ | 10 | 1,151.67 | 1,198.53 | -565.84 | 5 vs 6 | 19.50 | <0.0001 |
| **High weathering (88-100% CIA)** | | | | | | | |
| ~1 | 5 | 1,536.25 | 1,559.67 | -763.13 | NA | NA | NA |
| Clay + fine silt | 6 | 1,538.15 | 1,566.26 | -763.07 | 1 vs 2 | 0.10 | 0.7483 |
| … + $pH_{H2O}$ | 7 | 1,535.93 | 1,568.72 | -760.96 | 2 vs 3 | 4.22 | 0.0400 |
| … + $M_{ox}$ | 8 | 1,343.70 | 1,381.17 | -663.85 | 3 vs 4 | 194.23 | <0.0001 |
| … + $Ca_{ex}$ | 9 | 1,248.82 | 1,290.99 | -615.41 | 4 vs 5 | 96.87 | <0.0001 |
| … + $pH_{H2O}*M_{ox}$ | 10 | 1,215.27 | 1,262.12 | -597.64 | 5 vs 6 | 35.55 | <0.0001 |

MAT: Mean annual temperature; PET: Potential evapotranspiration; MAP: Mean annual precipitation; CIA: Chemical index of alteration; $M_{ox}$: Oxalate-extractable metals ($Al_{ox} + \frac{1}{2}Fe_{ox}$); Caex: Exchangeable calcium.

**Table B9: Anova summary for linear mixed-effects grouped by land cover ($n_{Cropland}$ = 429, $n_{Forest}$ = 228, $n_{Grassland}$ = 242, $n_{Other}$ = 702). Fixed effects were step-wise added. The first entry (~1) refers to the constant null model, respectively.**

| | df | AIC | BIC | logLik | Test | L.Ratio | p-value |
|---|---|---|---|---|---|---|---|
| **Cropland** | | | | | | | |
| ~1 | 5 | 942.57 | 962.88 | -466.28 | NA | NA | NA |
| Clay + fine silt | 6 | 942.77 | 967.13 | -465.38 | 1 vs 2 | 1.80 | 0.1794 |
| … + pH$_{H2O}$ | 7 | 943.73 | 972.16 | -464.86 | 2 vs 3 | 1.04 | 0.3085 |
| … + CIA | 8 | 911.72 | 944.21 | -447.86 | 3 vs 4 | 34.01 | <0.0001 |
| … + M$_{ox}$ | 9 | 817.96 | 854.51 | -399.98 | 4 vs 5 | 95.77 | <0.0001 |
| … + Ca$_{ex}$ | 10 | 755.49 | 796.11 | -367.75 | 5 vs 6 | 64.46 | <0.0001 |
| … + pH$_{H2O}$*M$_{ox}$ | 11 | 736.80 | 781.48 | -357.40 | 6 vs 7 | 20.69 | <0.0001 |
| **Forest** | | | | | | | |
| ~1 | 5 | 627.98 | 645.13 | -308.99 | NA | NA | NA |
| Clay + fine silt | 6 | 626.06 | 646.64 | -307.03 | 1 vs 2 | 3.92 | 0.0477 |
| … + pH$_{H2O}$ | 7 | 615.79 | 639.79 | -300.89 | 2 vs 3 | 12.27 | 0.0005 |
| … + CIA | 8 | 614.94 | 642.38 | -299.47 | 3 vs 4 | 2.85 | 0.0915 |
| … + M$_{ox}$ | 9 | 556.77 | 587.64 | -269.39 | 4 vs 5 | 60.17 | <0.0001 |
| … + Ca$_{ex}$ | 10 | 538.35 | 572.64 | -259.17 | 5 vs 6 | 20.42 | <0.0001 |
| … + pH$_{H2O}$*M$_{ox}$ | 11 | 532.33 | 570.05 | -255.16 | 6 vs 7 | 8.02 | 0.0046 |
| **Grassland** | | | | | | | |
| ~1 | 5 | 570.23 | 587.68 | -280.12 | NA | NA | NA |
| Clay + fine silt | 6 | 561.06 | 581.99 | -274.53 | 1 vs 2 | 11.18 | 0.0008 |
| … + pH$_{H2O}$ | 7 | 542.45 | 566.88 | -264.23 | 2 vs 3 | 20.60 | <0.0001 |
| … + CIA | 8 | 484.66 | 512.57 | -234.33 | 3 vs 4 | 59.79 | <0.0001 |
| … + M$_{ox}$ | 9 | 430.95 | 462.35 | -206.47 | 4 vs 5 | 55.71 | <0.0001 |
| … + Ca$_{ex}$ | 10 | 381.49 | 416.38 | -180.75 | 5 vs 6 | 51.45 | <0.0001 |
| … + pH$_{H2O}$*M$_{ox}$ | 11 | 352.66 | 391.04 | -165.33 | 6 vs 7 | 30.83 | <0.0001 |
| **Other** | | | | | | | |
| ~1 | 5 | 1,313.24 | 1,336.01 | -651.62 | NA | NA | NA |
| Clay + fine silt | 6 | 1,291.22 | 1,318.54 | -639.61 | 1 vs 2 | 24.02 | <0.0001 |
| … + pH$_{H2O}$ | 7 | 1,293.10 | 1,324.98 | -639.55 | 2 vs 3 | 0.12 | 0.7294 |
| … + CIA | 8 | 1,277.31 | 1,313.75 | -630.66 | 3 vs 4 | 17.79 | <0.0001 |
| … + M$_{ox}$ | 9 | 1,146.62 | 1,187.60 | -564.31 | 4 vs 5 | 132.70 | <0.0001 |
| … + Ca$_{ex}$ | 10 | 1,020.27 | 1,065.81 | -500.13 | 5 vs 6 | 128.35 | <0.0001 |
| … + pH$_{H2O}$*M$_{ox}$ | 11 | 1,011.66 | 1,061.75 | -494.83 | 6 vs 7 | 10.61 | 0.0011 |

MAT: Mean annual temperature; PET: Potential evapotranspiration; MAP: Mean annual precipitation; CIA: Chemical index of alteration; M$_{ox}$: Oxalate-extractable metals (Al$_{ox}$ + ½Fe$_{ox}$); Caex: Exchangeable calcium.