# Peer review of "Continental-scale controls on soil organic carbon across sub-Saharan Africa"

_SOIL, 2020_

## Referee Comment (RC1) · Anonymous Referee #1 · 12 Nov 2020

General comments

This very interesting paper tries to explain the heterogeneity of soil organic contents in tropical soils. Pedogenic, climatic and land cover properties were related to SOC contents to identify the main factors that control SOC. It mobilized numerous original data from a network of soil measurements through Africa. Two soil depths (0-20 cm and 20-50 cm) were considered. The paper is really worth a reading as the results were rich and surprising.

However, the work needs to be more explicit to be really compelling on some points (i.e. please give the main soil types studied: is there any andosols or young volcanic soils which could reinforce the role of Al, Fe extractable with oxalate in the SOC stabilization? It should be easy to dismiss that hypothesis).

[Figure]

Discussion on the limit of your study and on the differences with the previous studies could deepen the significance of your results. The clay and fine silt contents seem to have poor impact on the soil organic content on the contrary of previous results. OK but previous results have related SOC to the clay+silt size fraction at 0-20 $\mu$m, you have related SOC to the clay+silt size fraction at 0-8 $\mu$m. Please discuss that point? The separation of the two-studied soil depths is not always clear in the Ms. Can it affect the results?

One of your main result is that land cover do not explain SOC content at continent scale, so in the world wide discussion on C sequestration to mitigate CO2, what do you suggest? The final sentence on soil erosion is very OK but could be strengthen. Your remark on line 436 (". . .. on a regional scale") is also OK. Please insist a little more on that point. You have also to be clear at the end of the Ms between predict SOC content (for instance to map SOC content or stock) and predict SOC dynamics (you have not studied SOC dynamic in your study): Ln. 437 to the end are not very clear. After your result on the lack of significance between soil clay and SOC contents, what do you think about using the soil carbon saturation deficit to quantitatively assess the soil carbon storage potential in tropical soils? Could you add one or two sentences on that point.

Specific comments

Abstract: The conclusion of the abstract is not clear/strong enough.

Ln. 62: "broader variables such as clay content. . . " Ok but clay content implies SOC protection and association with minerals (L. 61). Please rephrase.

Ln. 64 "variety of processes" Please be more explicit (aggregation, organo mineral association. . .) Ln. 64 "differ" or their relative importance differ? (different hierarchisation). Please specify

End of the introduction: Please specify your hypothesis, you have not searched factors

randomly. Please be more explicit. Your questions are too vague. Which soil properties you will focus on? What findings do you except? You can base hypothesis on soil properties, climate properties and land cover. You have everything to make a more compelling introduction. Please explicitly justify the soil properties and the soil depth you will focus on in your study.

Ln. 105 "mid-infrared spectroscopy data" data or model? How have chosen the representative spectral data? Have you separate the two soil depth? Why have you finally chosen to study the lab measurements and not the predictions?

Ln. 120 why have you chosen this limit at 8 $\mu$m? What arguments?

Ln. 133 It seemed that soil texture was performed without organic matter destruction? Is that really the case? Please specify.

Ln. 136-140 it was not really clear to me.

Ln. 145 "1,601 soil samples" is that on the 2,002 measured soil samples?

Statistical analysis: Note for the editor: A deep analysis of the statistics has to be done by a reviewer more competent than I am in statistics.

Table 1: Please specify in the title that it is a summary for the two soil depths (0-20, n= ? ,and 20-50 cm, n= ?)

Figure 2: Please prefer unit (g kg-1 soil) than

Ln. 250 could you give some essential data from the Table B2 in the Ms. It could be nice for the ones who do not go and see the supplement data.

Table 2: "Depth (Subsoil)" is not clear. It is clear when reading ln 246-247, but it is not when reading the table alone. Table should explicit by itself.

Figure 3: I did not get the sentence "Note that the x-axis is truncated. . .." At what Caex content value it should end? 76? Please specify.

Ln. 286-287 and 301-302 Please make it clearer.

Ln. 304 yes we could already notice that point on Figure 3c according to the pH class (20 or 30 cmol+/kg)

Ln. 321-322 it is not very compelling. You have shown previously that there are slight differences between 0-20 and 20-50 cm (Ln 294, 253. . ...) Please be clearer.

Ln. 334 Please specify the size of the clay and fine silt fraction of the previously studies and yours. Discuss with these differences in mind.

Ln. 360-361 Yes it is OK, but please end the §by your results and not by the results of other scientists. Your result seemed to show that this mechanism seem to be stronger to explain SOC stabilization than straight organo-mineral association in quite different tropical environment. There is no words on particulate organic matter in your discussion. Have you any data to inform the role or the proportion of this organic matter pool in the total SOC content in your soil samples.

Here is some bibliography you could read in relation with your study. Perhaps that could help in the discussion : Barthès et al. (2008) in Geoderma (to discuss about Al, Fe oxides in tropical soils) ; Beare et al. (2014) in Biogeochemistry (to discuss about oxides and clay content) ; Fujisaki et al. (2018) in Geoderma and in AEE (to discuss about clay content, aridity index and SOC content and SOC storage in tropical soils specially) ; Wang et al. (2016) in Biogeoscience (to discuss about aridity index) ; Chenu and Plante (2006) in European Journal of Soil Science (to discuss about clay and fine silt content at 0-8 $\mu$m and 0-20 $\mu$m). Bibliography on the carbon saturation deficit.

---

## Referee Comment (RC2) · Anonymous Referee #2 · 23 Nov 2020

General comments:

The manuscript "Continental-scale controls on soil organic carbon across sub-Saharan Africa" describes a continental-scale analysis of associations between soil organic carbon and soil physico-chemical properties across Africa. The manuscript outlines a novel soil dataset collected at the Afsis "sentinel sites", and then steps through several statistical analyses that tease apart associations between carbon, extractable metals, and soil exchange pools across different domains of climate, soil pH, and soil weathering status. The authors conclude that short-range order (oxalate extractable Al) and to an extent Fe explain much of the variation in carbon stocks in wet/acid soils, whereas exchangeable calcium explains much of the variation in dry/alkaline soils. soil texture and land use appear largely irrelevant at this scale.

I think this manuscript is excellent and will be a very useful contribution to the study of soil geography. While the primary result has been identified in earlier studies (particularly Rasmussen et al.'s 2018 study), this manuscript applies to a different geographic domain (tropical and subtropical Africa) and with a more systematic data collection effort. It also considers soil weathering status using total elemental inventories and chemical weathering indices, which adds novelty. The results provide clear confirmation of the patterns hinted at in the Rasmussen study, and also point to some new complexities (particularly in relation to Fe). Furthermore, this study applies to data that were collected in a systematic sampling effort–hence these results should be considered more conclusive than those in earlier studies. The manuscript does a good job of balancing different statistical approaches, and stands as an example of how data-driven modelling tools (i.e. random forests) can be used responsibly in a process-oriented way to compliment more traditional statistical approaches. While at points the interpretation slides into a more descriptive "data-mining" posture, it is also punctuated with insightful process-based insights. In short: overall this is a strong manuscript!

My main criticisms apply to the way the methods are presente–I think some details are left out or insufficiently documented. I also think that the methods and discussion sections could use more of a "road map" at the start–particularly the discussion, which dives into a description of the correlations between different variables where it could start with some pithy statements summarizing the high level process-based interpretation.

I also would appreciate a bit more discussion of the underlying geographic patterns in the context of African geology (perhaps just a paragraph). I realize that the existing geospatial products don't allow for a thorough quantitative analysis of geologic state factors, but some limited qualitative might be good. More specifically the authors might address how parent material, soil age, and erosion rates vary (or do not vary) across the sampling locations, and how these might exert some influence on the results independent of climate.

Specific comments:

Lines 39-40: The phrase "complex analytical approaches with a large number of parameters" is somewhat opaque. Perhaps substitute something more specific?

Lines 62-63: To be fair here: there is an implicit representation of competition between microbes and minerals in Earth System models via clay content. There are two issues in this case: (1) competition between minerals and microbes is not represented in an explicit, mechanistic way; and (2) clay content doesn't capture the relevant aspects of soil mineralogy or chemistry. I think this manuscript addresses the latter issue more than the former.

Lines 129-131: Was this digestion quantitative? I believe some silicates are resistant to aqua regia. Perhaps clarify whether these should be considered total elemental pools or simply aqua-regia-digestible pools, as this may influence the interpretation of the CIA (though probably not much I imagine).

Line 160: It would be good to include a short overview paragraph at the start of the statistical analysis section explaining the overall strategy. It seems that several approaches were applied to the same data: linear mixed effects models, regression trees, and random forests. I can see how the approaches complement each other (the mixed effects models seem more conservative and permit statistical hypothesis testing while accounting for non-independence of the data, but the CART based approaches can handle non-linearity). This is explained later, but the readers will benefit from a quick signpost at the start. Similarly, the discussion section is hard to follow at the start. I strongly recommend adding a concise paragraph at the beginning of the discussion that identifies the major results. As it stands now the discussion dives right into the details and I can only identify an emergent narrative at the end.

Lines 167-171: I understand that the transformation is necessary for comparing different predictors on the same scale. However, what does the transformation mean with respect to the functional relationships in the data? Are the models linear with respect

to the original scale? I suspect not: a linear model fit to transformed data is not necessarily a linear model with respect to the original data. This is worth noting, even if the analysis stays the way it is.

Line 183: How was the hierarchical clustering done?

Line 204: The spatial partitioning is really laudable. It is surprising how infrequently this is done, and it really should be a community standard. Thank you for being rigorous!

Line 242: Please introduce the marginal/conditional R-squared values before mentioning here. To many readers this distinction might not be obvious.

Figure 2: The univariate linear regression fits in this figure are purely for illustration? Perhaps mention them briefly in the statistical analysis section.

Figure 3 (and throughout): How were confidence intervals obtained? They are reported throughout the paper, but unless I missed something the method used to obtain them is not reported.

Line 289: How was the % variation explained obtained here? Is this an R-squared value for a reduced model? Or is it some sort of variable importance metric? Perhaps something is missing from the methods description?

Line 446: I hope that the data presented in this study are eventually made available in some easy-to-access way. A database of this size and completeness could be extremely valuable to other researchers and would be best archived on some sort of data repository rather than only available on request from the author.

---

## Author Comment (AC1) · 5 Jan 2021

Note: The line numbers in our answers are referring to the updated manuscript.

**Anonymous Referee #1**

Received and published: 12 November 2020

**General comments:**

REVIEWER\_01: This very interesting paper tries to explain the heterogeneity of soil organic contents in tropical soils. Pedogenic, climatic and land cover properties were related to SOC contents to identify the main factors that control SOC. It mobilized numerous original data from a network of soil measurements through Africa. Two soil depths (0-20 cm and 20-50 cm) were considered. The paper is really worth a reading as the results were rich and surprising.

ANSWER: Thank you very much for taking the time to review our manuscript and your most constructive feedback. Our detailed answers can be found below.

R\_01: However, the work needs to be more explicit to be really compelling on some points (i.e. please give the main soil types studied: is there any andosols or young volcanic soils which could reinforce the role of AI, Fe extractable with oxalate in the SOC stabilization? It should be easy to dismiss that hypothesis).

A: This is a really good point. Unfortunately, we do not have the information about the main soil types. When extracting data from global and African soil-type products, none of the soil profiles was classified as Andosols. However, those products are usually not that precise and it is difficult to derive detailed conclusions. Based on the position of some soil profiles (e.g. Rift Valley in East Africa) it might be possible that some of them are Andosols or influenced by volcanic activities in the past.

Based on this comment and also on one comment from Reviewer #2 we added the following paragraph to the discussion section in the manuscript.

Line 465: "

**Geographic patterns**

All soils result globally from the same soil-forming factors (climate, organisms, topography, parent material and time) and are formed by similar processes (e.g. oxidation, reduction, leaching, transport and accumulation). This might explain to some extent, why similar soil and climate parameters are important to explain SOC content variation in sub-Saharan Africa as compared to other regions. However, significant differences are still visible in subtropical and tropical soils, especially in terms of mineralogy, weathering and soil formation, which are related to important differences in climate, soil age, parent material and vegetation (Buringh 1970). Such differences do occur between soils from sub-Saharan Africa, which do not only differ greatly in their soil properties and climate (Table 1), but also in vegetation, parent material, soil age and their vulnerability to degradation (Jones et al. 2013). However, due to the lack of precise geospatial products for those parameters for sub-Saharan Africa, we can only discuss them here qualitatively.

The AfSIS sites (Figure 1) are mainly derived from two parent material types: i) metamorphic rocks and ii) volcanic rocks (Hartmann and Moosdorf 2012; Jones et al. 2013; Schlüter 2008). Metamorphic rocks are most commonly found in West Africa, Southern Africa and Madagascar. These regions are characterized by old cratons, except for Madagascar, which is influenced by Mesozoic volcanism (Schlüter 2008). Most of these soils are classified as Ferralsols according to the WRB soil classification system (Jones et al. 2013). This partly explains, why the AfSIS soils from those regions are usually highly weathered with low pHH2O values. In contrast, soils derived from volcanic rocks are mainly found in East Africa in the Great Rift

Valley. These soils are usually younger and less weathered (Buringh 1970), which is also mirrored in soil properties within the AfSIS data set. These soils are characterized by lower CIA values and higher Alox and Feox concentrations compared to other soils in the AfSIS data set. Outside of the influence of volcanic rocks, Ca2+ rich soils are frequent in East Africa and are dominated by a high concentration of Caex and high pHH2O values. Since Alox, Feox, and Caex were important predictors of SOC in our analyses, the SOC content is usually also higher at AfSIS sites in East Africa compared to sites in West Africa and Southern Africa.

Although certain soil properties, in combination with climate variables, are important to explain SOC concentration variation, different soil forming factors, such as parent material and soil age, are important to understand, which soil properties will dominate – at least at this large-scale approach. To link those two aspects more quantitatively on continental-scales might be a direction for future studies."

R\_01: Discussion on the limit of your study and on the differences with the previous studies could deepen the significance of your results. The clay and fine silt contents seem to have poor impact on the soil organic content on the contrary of previous results. OK but previous results have related SOC to the clay+silt size fraction at 0-20 µm, you have related SOC to the clay+silt size fraction at 0-8 µm. Please discuss that point?

A: Thank you for pointing out this important aspect. We realized that we have to be more explicit on why we used the rather unusual cut-off of

Figure A1: Duplicate measurements of the particle size distribution data [%]. Panel a) Duplicate 1 and 2

---

## Author Comment (AC2) · 5 Jan 2021

*Note: The line numbers in our answers are referring to the updated manuscript.*

**Anonymous Referee #2**

**General comments:**

REVIEWER_02: The manuscript "Continental-scale controls on soil organic carbon across sub-Saharan Africa" describes a continental-scale analysis of associations between soil organic carbon and soil physico-chemical properties across Africa. The manuscript outlines a novel soil dataset collected at the Afsis "sentinel sites", and then steps through several statistical analyses that tease apart associations between carbon, extractable metals, and soil exchange pools across different domains of climate, soil pH, and soil weathering status. The authors conclude that short-range order (oxalate extractable Al) and to an extent Fe explain much of the variation in carbon stocks in wet/acid soils, whereas exchangeable calcium explains much of the variation in dry/alkaline soils. soil texture and land use appear largely irrelevant at this scale.

I think this manuscript is excellent and will be a very useful contribution to the study of soil geography. While the primary result has been identified in earlier studies (particularly Rasmussen et al.'s 2018 study), this manuscript applies to a different geographic domain (tropical and subtropical Africa) and with a more systematic data collection effort. It also considers soil weathering status using total elemental inventories and chemical weathering indices, which adds novelty. The results provide clear confirmation of the patterns hinted at in the Rasmussen study, and also point to some new complexities (particularly in relation to Fe). Furthermore, this study applies to data that were collected in a systematic sampling effort–hence these results should be considered more conclusive than those in earlier studies. The manuscript does a good job of balancing different statistical approaches, and stands as an example of how data-driven modelling tools (i.e. random forests) can be used responsibly in a processoriented way to compliment more traditional statistical approaches. While at points the interpretation slides into a more descriptive "data-mining" posture, it is also punctuated with insightful process-based insights. In short: overall this is a strong manuscript!

ANSWER: We highly appreciate this very thoughtful and appreciative review. Thank you for taking the time to carefully comment on our manuscript. We will address the suggestions in detail in the following response.

R_02: My main criticisms apply to the way the methods are presente–I think some details are left out or insufficiently documented. I also think that the methods and discussion sections could use more of a "road map" at the start–particularly the discussion, which dives into a description of the correlations between different variables where it could start with some pithy statements summarizing the high level process-based interpretation.

A: We really appreciate these comments. We will address them in detail under the corresponding *Specific comments*.

R_02: I also would appreciate a bit more discussion of the underlying geographic patterns in the context of African geology (perhaps just a paragraph). I realize that the existing geospatial products don't allow for a thorough quantitative analysis of geologic state factors, but some limited qualitative might be good. More specifically the authors might address how parent material, soil age, and erosion rates vary (or do not vary)

across the sampling locations, and how these might exert some influence on the results independent of climate.

A: Thank you for this comment. We agree that there seems to be no appropriate geospatial product for lithology that allows for a thorough quantitative analysis. We added a paragraph called *Geographic patterns* in the discussion section where we did a qualitative interpretation of some geographical patterns, such as lithology and soil age. In terms of erosion we added some details in the paragraph *Land cover* in the discussion section (those details are also based on a comment and some references from the Reviewer#1).

Line 465: "

**Geographic patterns**

All soils result globally from the same soil-forming factors (climate, organisms, topography, parent material and time) and are formed by similar processes (e.g. oxidation, reduction, leaching, transport and accumulation). This might explain to some extent, why similar soil and climate parameters are important to explain SOC content variation in sub-Saharan Africa as compared to other regions. However, significant differences are still visible in subtropical and tropical soils, especially in terms of mineralogy, weathering and soil formation, which are related to important differences in climate, soil age, parent material and vegetation (Buringh 1970). Such differences do occur between soils from sub-Saharan Africa, which do not only differ greatly in their soil properties and climate (Table 1), but also in vegetation, parent material, soil age and their vulnerability to degradation (Jones et al. 2013). However, due to the lack of precise geospatial products for those parameters for sub-Saharan Africa, we can only discuss them here qualitatively.

The AfSIS sites (Figure 1) are mainly derived from two parent material types: i) metamorphic rocks and ii) volcanic rocks (Hartmann and Moosdorf 2012; Jones et al. 2013; Schlüter 2008). Metamorphic rocks are most commonly found in West Africa, Southern Africa and Madagascar. These regions are characterized by old cratons, except for Madagascar, which is influenced by Mesozoic volcanism (Schlüter 2008). Most of these soils are classified as Ferralsols according to the WRB soil classification system (Jones et al. 2013). This partly explains, why the AfSIS soils from those regions are usually highly weathered with low $pH_{H2O}$ values. In contrast, soils derived from volcanic rocks are mainly found in East in the Great Rift Valley. These soils are usually younger and less weathered (Buringh 1970), which is also mirrored in soil properties within the AfSIS data set. These soils are characterized by lower CIA values and higher $Al_{ox}$ and $Fe_{ox}$ concentrations compared to other soils in the AfSIS data set. Outside of the influence of volcanic rocks, $Ca^{2+}$ rich soils are frequent in East Africa and are dominated by a high concentration of $Ca_{ex}$ and high $pH_{H2O}$ values. Since $Al_{ox}$, $Fe_{ox}$, and $Ca_{ex}$ were important predictors of SOC in our analyses, the SOC content is usually also higher at AfSIS sites in East Africa compared to sites in West Africa and Southern Africa.

Although certain soil properties, in combination with climate variables, are important to explain SOC concentration variation, different soil forming factors, such as parent material and soil age, are important to understand, which soil properties will dominate – at least at this large-scale approach. To link those two aspects more quantitatively on continental-scales might be a direction for future studies."

Line 451: "This might be due to the high variation of SOC content within the different land cover classes at this large spatial scale ("Figure 1a). For example, the class cropland contains a wide variety of cultivated plots – we did not have more detailed information about land management practices. Fujisaki et al. (2018b) showed that SOC stock changes in tropical cropland soils are mainly driven by C inputs.

On the other hand, it is also known that land use changes and their impact on soil physico-chemical properties are scale-dependent and are likely to be more distinct at smaller scales (Holmes et al. 2005; Holmes et al. 2004). For example, land management and land degradation (i.e. erosion) are known to impact SOC stocks on regional scales in sub-Saharan Africa (Winowiecki et al. 2016a). However, we lacked the detailed data necessary to disentangle the impacts of different practices. Additionally, since the focus

of our work was on natural soil physico-chemical and climate properties, we did not further investigate those anthropogenic factors at this large spatial scale. Future studies are needed to better understand the impacts of land management and carbon storage potential in soils across sub-Saharan Africa at different scales (Fujisaki et al. 2018a; Vanlauwe et al. 2015). Overall, our data for sub-Saharan Africa suggests that SOC content on a continental scale is better explained by stabilization potential in soils (climate, geochemistry) than by different aboveground C inputs (vegetation)."

**Specific comments:**

R_02: Lines 39-40: The phrase "complex analytical approaches with a large number of parameters" is somewhat opaque. Perhaps substitute something more specific?

A: Line 40: "Assessing the state of soils and their potential response to climate and land-use change requires carefully designed sampling strategies, combined with systematic analytical and statistical analyses across locations and scale (IPCC 2019)."

R_02: Lines 62-63: To be fair here: there is an implicit representation of competition between microbes and minerals in Earth System models via clay content. There are two issues in this case: (1) competition between minerals and microbes is not represented in an explicit, mechanistic way; and (2) clay content doesn't capture the relevant aspects of soil mineralogy or chemistry. I think this manuscript addresses the latter issue more than the former.

A: A similar comment was brought up by the Reviewer #1. We agree that the paragraph about model approaches was not that accurate and have now updated it. Since this is not the main focus of our manuscript we would like to keep this paragraph rather short. However, we think it is an important aspect worth mentioning in the introduction. We have revised the paragraph as follows:

Line 61: "SOC stabilization is commonly conceptualized as competition between accessibility for microorganisms versus chemical associations with minerals (Oades 1988; Schmidt et al. 2011). These processes are often only considered implicitly by models (Blankinship et al. 2018; Schmidt et al. 2011). Instead, models commonly rely on broader variables such as clay content, which is used as a proxy for sorption and other organo-mineral interactions (Rasmussen et al. 2018; Schmidt et al. 2011). These more generic variables integrate a variety of stabilization processes which can be difficult to disentangle. They might even differ in their relative importance and may not adequately capture soil mineralogy and chemistry across different ecosystems and climate zones. Hence, improving the predictive capacity of such models requires not only a better understanding of the factors that control SOC dynamics, but also verification (or falsification) of those new findings in regions that are underrepresented in field studies and models."

R_02: Lines 129-131: Was this digestion quantitative? I believe some silicates are resistant to aqua regia. Perhaps clarify whether these should be considered total elemental pools or simply aqua-regia-digestible pools, as this may influence the interpretation of the CIA (though probably not much I imagine).

A: Line 142: "Aqua regia acid digestion was applied for major and trace elements, including Al, Ca, K and Na. Although this method does not give absolute total contents, it does give results sufficiently close to accepted values for different soils (McGrath and Cunliffe 1985)."

R_02: Line 160: It would be good to include a short overview paragraph at the start of the statistical analysis section explaining the overall strategy. It seems that several approaches were applied to the same data: linear mixed effects models, regression trees, and random forests. I can see how the approaches complement each other (the mixed effects models seem more conservative and permit statistical hypothesis testing while accounting for non-independence of the data, but the CART based approaches can handle non-linearity). This is explained later, but the readers will benefit from a quick signpost at the start. Similarly, the discussion section is hard to follow at the start. I strongly recommend adding a concise

paragraph at the beginning of the discussion that identifies the major results. As it stands now the discussion dives right into the details and I can only identify an emergent narrative at the end.

A: Statistical analyses section: Line 179: "We used several statistical approaches to analyze our data set, including linear mixed effects models, regression trees and random forests. We compared the results of these three different methods to confirm key findings and derive complementary insights. In brief, we used linear mixed effects model to handle the clustered and dependent sampling design of the AfSIS data set, whereas regression trees and random forests enabled us to account for non-linearities within the data. More precisely, we used regression trees as a qualitative tool to explore and understand the structure of the data, whereas random forests offered more generalizable models. All statistical analyses were performed within the R computing environment (Version 4.0.0, R Core Team 2020). The R Markdown file in the SI provides the code to reproduce all our analyses."

A: Discussion section: Line 349 (We moved the last paragraph to the beginning of this section and summarized our main findings at the beginning of the discussion): "Here, we focus on those variables that showed the most explanatory power in terms of SOC content across all models. We then compare their explanatory power with those reported in other studies for different regions. Short-range order minerals ($Al_{ox}$) and to some extent $Fe_{ox}$ explained much of the variation in SOC concentration in wet regions with acidic and highly weathered soils. In contrast, $Ca_{ex}$ explained much of the variation in dry regions, dominated by alkaline and less weathered soils. In addition, we discuss the role of clay and fine silt content, and of land cover, since they were important in other studies. However, in our study, the latter did not explain much of the variation in SOC content, which may be due to the large spatial scale. At the end of this section, we discuss the underlying geographic patterns that emerged in the data.

Some common predictors of SOC and dependencies between predictors (MAP/PET, $pH_{H2O}$, CIA) emerged across all modeling approaches. Number of wet months, soil $pH_{H2O}$ and weathering status (Figures 3 and 4) occurred as key parameters in the linear mixed effects models that influence how other parameters, such as $Ca_{ex}$, $Al_{ox}$ and $Fe_{ox}$, explain SOC content variation across sub-Saharan Africa. In contrast, predictor differences were much smaller between topsoil (0–20 cm) and subsoil (20–50 cm) samples. This may partly be due to the large depth increments for each of the two sampling depths. However, since the identified SOC-controlling factors were similar for both depth layers, any differences were probably mostly driven by the fact that subsoil samples usually contain less SOC due to lower C inputs at depth (Jobbágy and Jackson 2000). Soil erosion at some sites might also dilute differences between the two depth layers, since water and wind can permanently remove surface soil.

These findings were supported by the regression trees (Figure A6) and partial dependence plots (Figure 5), where $Ca_{ex}$, $Al_{ox}$ and $Fe_{ox}$ seemed to be more important in explaining the variation of SOC concentration compared to $pH_{H2O}$, PET/MAP and CIA. For example, soil $pH_{H2O}$ was important in the full linear mixed effects model, yet it mainly influenced $Ca_{ex}$, $Al_{ox}$, and $Fe_{ox}$ concentrations in correlation with MAP (Figure 3d); the same was true for weathering (Figure 4b). Similar relationships have been found for temperate regions, where the importance of $Ca_{ex}$ increased with increasing $pH_{H2O}$ and decreasing precipitation, whereas the opposite was true for $Al_{ox}$ (Oades 1988; Rasmussen et al. 2018). However, Rasmussen et al. (2018) did not identify $Fe_{ox}$ as an important predictor of SOC content."

R_02: Lines 167-171: I understand that the transformation is necessary for comparing different predictors on the same scale. However, what does the transformation mean with respect to the functional relationships in the data? Are the models linear with respect to the original scale? I suspect not: a linear model fit to transformed data is not necessarily a linear model with respect to the original data. This is worth noting, even if the analysis stays the way it is.

A: It is correct that transformation and standardization of the data prior to linear mixed effects modelling does not mean that the original relationship between SOC and the predictors is always linear. We have

clarified this in the text. This is one of the reasons why we also used regression trees and random forests. $Fe_{ox}$, for example, nicely demonstrates this point – it is not important in the linear mixed effects models, yet, it is really important in regression trees and random forests. This is because it does not have a linear relationship with SOC across its entire range – it only shows a strong correlation with SOC at low concentrations (see Figure 5e in the manuscript).

Line 198: "This only holds for the transformed and standardized data and the relationship between SOC and the predictors of the original data may not be linear."

R_02: Line 183: How was the hierarchical clustering done?

A: Thank you for the critical question. We realized that 'hierarchal clustering' is not the appropriate term here and apologize for any confusion that was caused by that. We used a built-in function called *cut_number* from the *ggplot2* package in *R* which allows to control for the number of groups, whereas the cut-offs are determined by the function internally to approximately equalize the number of samples in each group. We tried different numbers of groups to match common pH and CIA classes while trying to maximize the number of samples in each group (i.e. keeping the numbers of groups as small as possible) at the same time. The exact approach can be found in the R Markdown file in the SI (p 11-12). We have clarified this in the text.

Line 210: "Soil $pH_{H2O}$ and CIA data were grouped using hierarchical clustering, with the number of classes chosen to maximize and equalize the number of samples in each class and to correspond with common $pH_{H2O}$ and weathering categories (Nesbit and Young 1982)."

R_02: Line 204: The spatial partitioning is really laudable. It is surprising how infrequently this is done, and it really should be a community standard. Thank you for being rigorous!

A: Thank you very much for this really positive feedback. We agree that this should become a standard when working with geospatial data.

R_02: Line 242: Please introduce the marginal/conditional R-squared values before mentioning here. To many readers this distinction might not be obvious.

A: Since this is the only time we use the term marginal, we decided to remove it and added in brackets what the $R^2$ in this case means.

Line 272: "The final linear mixed effects model for the entire data set (n = 1,601) had a marginal $R^2$ of 0.71 (excluding the proportion variance explained by the fixed effects *Site/Cluster/Profile*)."

R_02: Figure 2: The univariate linear regression fits in this figure are purely for illustration? Perhaps mention them briefly in the statistical analysis section.

A: It is correct that we added those regression lines in Figure 2 for illustrative reasons. The regression line follows the simple linear equation y ~ x. Since the linear regression line in the figure is not used in further analysis and not important for the discussion, we have now included the formula and clarifying information in the caption of Figure 2, rather than in the methods section:

Line 264: "Figure 1: a) Soil organic carbon (SOC) content [wt-%] for the different land-covers (cropland, forest, grassland, other (bushland, shrubland, woodland) by depth (topsoil: 0–20 cm, subsoil: 20–50 cm); b) SOC [wt-%] and clay and fine silt content [%] by depth; c) SOC [wt-%] and clay and fine silt content [%] by depth for three example sites that show contrasting trends. Gray area around fitted linear regressions (y ~ x, for illustration only) in b) and c) show the 95% confidence interval. For the relationship between SOC and clay and fine silt content for all individual sites, see Figure A4."

R_02: Figure 3 (and throughout): How were confidence intervals obtained? They are reported throughout the paper, but unless I missed something the method used to obtain them is not reported.

A: It is correct that we did not specify the method how we obtained the confidence intervals – this has now been corrected. To visualize and report the linear mixed effects models (including the confidence intervals) in Figure 3a,b, 4a, and A5 and in Table B2 to Table B7 we used the *sjPlot* package in R (Lüdecke 2020). Within the package we used the Wald-test approximation to calculate the 95% confidence interval (https://easystats.github.io/parameters/reference/p_value_wald.html). Based on this method the confidence interval is calculated the follow for each explanatory variable:

*Confidence interval (95%) = Coefficient ± 1.96 * SE*

Where *Coefficient* is the coefficient of each explanatory variable from the linear mixed effects model and *SE* is the standard error of the maximum likelihood of the same explanatory variable.

Line 192: "The 95% confidence intervals were obtained by using the Wald-test approximation within the sjPlot and parameters R packages (Lüdecke 2020; Lüdecke et al. 2020)."

R_02: Line 289: How was the % variation explained obtained here? Is this an R-squared value for a reduced model? Or is it some sort of variable importance metric? Perhaps something is missing from the methods description?

A: We replaced the word *variation* with *data* and added an explanation in the method section. The percentage is referring to the relative number of observations in this particular node of the regression tree. In this particular case, the SOC content of 23% of the samples was predicted by using $Fe_{ox}$ and MAT only.

Line 236: "Absolute values at the bottom of each node indicate the predicted SOC content [wt-%] and the percentage corresponds to the relative number of samples in this node (Figure A6)."

Line 320: "About 23% of the SOC data could be explained by $Fe_{ox}$ and MAT alone."

R_02: Line 446: I hope that the data presented in this study are eventually made available in some easy-to-access way. A database of this size and completeness could be extremely valuable to other researchers and would be best archived on some sort of data repository rather than only available on request from the author.

A: Thank you very much for bringing up this important aspect of *Open Science.* We fully agree that the analyzed data set should be open and easily accessible to everyone. In parallel to this review process we are already working on this. We are planning to archive the dataset on the following repository were already other data from the same project has been archived: https://data.worldagroforestry.org/dataverse/icraf_soils. However, we still need to solve some legal issues.

Line 501: "The soil properties data set used in this study is available from the corresponding author upon reasonable request and will be available on https://data.worldagroforestry.org/dataverse/icraf_soils in mid-2021."

**References**

Buringh P (1970) *Introduction to the study of soils in tropical and subtropical regions.* Centre for Agricultural Publishing and Documentation: Wageningen, Netherlands, pp.°99.

Hartmann J, Moosdorf N (2012) The new global lithological map database GLiM: A representation of rock properties at the Earth surface. *Geochemistry, Geophysics, Geosystems,* 13(12): pp., doi: 10.1029/2012gc004370.

IPCC (2019) *Climate Change and Land, an IPCC special report on climate change, desertification, land degradation, sustainable land management, food security, and greenhouse gas fluxes in terrestrial ecosystems.* IPCC: Geneva, Switzerland.

Jones A, Breuning-Madsen H, Brossard M, Dampha A, Deckers J, Dewitte O, Gallali T, Hallett S, Jones R, Kilasara M, Le Roux P, Michéli E, Montanarella L, Spaargaren O, Thiombiano L, van Ranst E, Yemefack M, Zougmore R (eds., 2013) *Soil Atlas of Africa.* Publications Office of the European Union: Luxembourg, pp. 176.

Lüdecke D (2020) *sjPlot: Data Visualization for Staistics in Social Science.* https://CRAN.R-project.org/package=sjPlot.

Lüdecke D, Ben-Shachar MS, Patil I, Makowski D (2020) parameters: Extracting, Computing and Exploring the Parameters of Statistical Models using R. *Journal of Open Source Software,* 5(53): pp. 2445, doi: 10.21105/joss.02445.

McGrath SP, Cunliffe CH (1985) A simplified method for the extraction of the metals Fe, Zn, Cu, Ni, Cd, Pb, Cr, Co and Mn from soils and sewage sludges. *Journal of the Science of Food and Agriculture,* 36(9): pp. 794-798, doi: 10.1002/jsfa.2740360906.

Nesbit HW, Young GM (1982) Early Proterozoic climates and plate motions inferred from major element chemistry of lutites. *Nature,* 299: pp. 715-717, doi: 10.1038/299715a0.

Oades JM (1988) The retention of organic matter in soils. *Biogeochemistry,* 5(1): pp. 35-70, doi: 10.1007/BF02180317.

Schlüter T (2008) *Geological Atlas of Africa.* Springer: Heidelberg, Germany, pp.°307.

Schmidt MWI, Torn MS, Abiven S, Dittmar T, Guggenberger G, Janssens IA, Kleber M, Kögel-Knabner I, Lehmann J, Manning DAC, Nannipieri P, Rasse DP, Weiner S, Trumbore SE (2011) Persistence of soil organic matter as an ecosystem property. *Nature,* 478: pp. 49, doi: 10.1038/nature10386.

---

## Author Response (AR1)

Dear Dr. Bauters,

Thank you very much for your time to evaluate our manuscript ("Continental-scale controls on soil organic carbon across sub-Saharan Africa", soil-2020-69). We are very pleased that the two reviewers positively assessed our work and that you are further considering the publication of our work in the special issue "Tropical biogeochemistry of soils in the Congo Basin and the African Great Lakes region" in SOIL. The issues raised by you and the reviewers helped greatly to improve our manuscript and we sincerely thank you all for the constructive and valuable insights.

We have addressed all comments and suggestions to the best of our ability. In particular, we carefully addressed your comments on the statistics we used for the linear mixed-effects models and the related comments by the reviewers about the role of land cover and clay + fine silt content in our data set. After revising and changing our statistical approach for the linear mixed-effects models, the main findings and conclusions of our manuscript did not change. Overall, we think that the changes we made improved the manuscript and that the results are presented in a more straightforward way.

Please find below a point-by-point response to all the concerns raised, and the changes we made to address them. We have submitted the revised document, a track-changes version and the updated supplement information to ease the review of the implemented changes. Page and line references given in our answers refer to the revised manuscript.

Thank you for your consideration and we look forward to hearing from you.

Sincerely, on the behalf of all authors,
Sophie von Fromm

**Editor: Point-by-Point response**

*EDITOR1:*

*Dear Authors,*

*Two reviewers have now submitted their evaluation of the paper. Both saw great potential and were generally positive about the manuscript. However, both reviewers raise excellent points where the paper could be improved. I suggest to rework the paper using these constructive comments, along the lines of your response. I especially would encourage clarifying some parts of the methods section as referee 2 points out.*

*However, in addition to the reviewer comments, I have some additional thoughts/concerns that deserve the authors' attention while reworking the manuscript. Based on referee 1's call for an in-depth check of the paper's statistics, I have checked the methods section of the paper myself...*

**ANSWER1:** Thank you very much for taking the time to review our manuscript and to read the answers we gave to the two reviewers. We really appreciate your very constructive feedback that has led us to improve the manuscript. Our detailed answers can be found below.

*E2: I nowhere read that the model assumptions (residual normality and homoscedasticity to start with) were met.*

**A2:** Thank you for making this important point. We have carefully ensured that the model assumptions are met for each model. We made the following changes in the manuscript:

Line 196: "To meet linear mixed-effects model assumptions and to standardize variation among variables, all continuous parameters were transformed to a normal distribution using Box-Cox transformation, followed by standardization to a mean of 0 and standard deviation of 1 by using the R package bestNormalize (Peterson and Cavanaugh 2019)."

In addition, the diagnostic plots for all linear-mixed effects models can be found in the R markdown file in the supplementary materials now. For each model, we checked if model assumptions are met by plotting i) residuals vs fitted values to verify homogeneity, ii) a qq-plot of the residuals for normality, and iii) residuals versus each explanatory variable to check for independence. Below is an example for the linear-mixed effects model with the entire dataset (n = 1,601):

[Figure]

[Figure]

**E3:** *What approximation method was used to estimate the mixed effects models P-values? Kenward Rogers? Satterthwaite?*

**A3:** In the new version of the manuscript, we carefully revised the statistics of the linear-mixed effects models, so this comment no longer applies. A more detailed answer and description of the method applied in the revised manuscript can be found below.

**E4:** *The R2 table at the end (Table B8) is currently confusing. It reports only 4 model fits, where there is little reference to these in other parts of the paper. Inline you state that land cover (L413)/ clay+fine silt (L404) explain 10% of the variation (conditional R2), but the table currently says 0.01 for both.*

**A4:** We apologize for the mismatch between the numbers in the text and the table. The number reported in the table (marginal $R^2 = 0.01$) is correct. The numbering of the table changed from B8 to B3 (line: 772). We changed the text accordingly (line 444).

**E5:** *A major thing to consider: fixed effects selection in regression analyses is debated, and it has been shown to lead to spurious claims on the data, depending on what method you use for the selection. It is simply not the best practice, although I acknowledge that it is commonly used. I suggest you address this by either explaining more in-depth why you did a predictor selection this way, or use a more robust method (splitting up data in model building - model validation, or using shrinkage via a penalizing function, or others).*

**A5:** We agree that fixed effects selection is not trivial and not always the best choice. Therefore, we have now updated our methods, and no longer reduce the models. Since we selected the SOC predictors (n = 10) based on a-priori knowledge and given the number of samples within each model (n = 228–1,601), there should be no danger of over-fitting the model. Instead, we started with a constant model that only included siteID/clusterID/plotID as random effects. We then extended the model step-wise by fitting the fixed effects in a pre-defined order – starting with large-scale climate variables and ending with fine scale physiochemical soil properties. We used the maximum likelihood method and the likelihood ratio test (L.ratio) to evaluate model performance and the significance of each added fixed effect; rather than relying on the coefficient estimates.

In the end, each predictor is included in the full model, independent of whether the contribution of the predictor to the model was significant or not. We used the marginal $R^2$ (excluding the variation explained by the random effects) to

obtain the variation explained by each added predictor. We added summary tables for each model in the SI of the manuscript (Table B4-B9). In the revised version, the section on the linear-mixed reads:

Line 200: "To answer our 1st research question, which soil properties and climate parameters best explain SOC content, we started from a constant null model with siteID/clusterID/plotID as random effects and then extended the model step-wise by fitting the following sequence of fixed effects: MAT, PET/MAP, depth, land cover, clay and fine silt, $pH_{H20}$, CIA, $M_{ox}$ ($Al_{ox}$ + ½$Fe_{ox}$), $Ca_{ex}$, $pH_{H2O}$*$M_{ox}$. The order and selection of fixed effects was pre-defined based on a-priori knowledge out of a larger set of variables (Burnham and Anderson 2002), starting with large-scale climate variables and ending with fine scale physiochemical soil properties. The oxalate-extractable metals $Al_{ox}$ and $Fe_{ox}$ were summed to $M_{ox}$ ($Al_{ox}$ + ½$Fe_{ox}$) to normalize the atomic mass difference between Al and Fe (Wagai et al. 2020) and to account for their similar behavior over their concentration range (Figure 5b). The maximum likelihood method and likelihood ratio tests (L.ratio) were applied to evaluate model performance and the statistical significance of the added fixed effects (Table B4-B9). The variation explained by each fixed effect was obtained by calculating the marginal $R^2$ (excluding the variation explained by the random effects siteID/clusterID/plotID) for each model and subtracting the $R^2$ from the previous fitted model using the function r.squaredGLMM from the MuMIn R package (Barton 2020; Nakagawa and Schielzeth 2013)."

**E6:** *In this specific case, I think it is actually very important: as the authors note themselves in the manuscript, but also as both referees noted: the land cover not being retained in the final model will lead the readership to conclude that land cover is not important in predicting SOC%. This is a potentially important finding and one of the main conclusions of the manuscript. I think that you could consider stressing that this conclusion is based on a regional-scale analysis where some other predictors simply range in about two orders of magnitude (for some predictors).*

**A6:** Land cover is now included in each model. However, the results (based on L.ratio tests and marginal $R^2$ estimates) of the revised models still suggest that land cover is not an important predictor of SOC content. Nevertheless, we agree that the role of land cover is not always that easy to disentangle and may be obscured by other predictors in such a continental-scale analysis. Based on the comment from the Editor and additional comments made by the two reviewers, we made the following changes in the manuscript:

Line 445: "The effect of land cover on SOC content was generally small in our models, even in topsoils (Figure 6i). Similar findings were recently encountered in a global study (Luo et al. 2021). One possibility may be that the relatively large 0–20 cm depth interval might dilute differences that would be more marked in the top few centimeters. However, we did observe differences in SOC content across land cover classes, with forests containing the highest amount of SOC – especially in topsoils (Figure 2a). Croplands had higher SOC content than grasslands, opposite of what is commonly observed in temperate regions (Prout et al. 2020).

Another possible explanation for the absence of land cover as an important predictor in our models, is that we lacked the detailed data necessary to disentangle impacts of different practices and land-use history. The land cover class cropland contained a wide variety of cultivated plots while more detailed information about land management practices were missing. This is particularly important since prior research in other regions showed that SOC stock changes in tropical cropland soils may be driven by C inputs (Fujisaki et al. 2018b). Additionally, historical land use may even play a more important role in explaining current stocks compared to recent land use (Vågen et al. 2006).

Furthermore, land cover may covary with other parameters (temperature, precipitation, geochemistry) to such a degree that it is not an explanatory variable. This might be the reason why the sub-models grouped by land cover did not show a clear pattern (Figure 4e). However, the land cover-only models resulted in small $R^2$ (linear mixed-effects models: 0.01; random forest: 0.10–0.16) which suggests that land cover is a poor predictor for our SOC data at this large spatial scale (Table B3). This may be due to the high variation of SOC content within the different land cover classes (Figure 2a). Land use changes and their impact on soil physico-chemical properties are scale-dependent and likely to be more distinct at smaller scales (Holmes et al. 2004, 2005). For example, land management and land degradation (i.e. erosion) are known to impact SOC stocks in regionals scales in sub-Saharan Africa (Winowiecki et al. 2016a).

Future studies are needed to better understand the impacts of land management and carbon storage potential in soils across sub-Saharan Africa at different scales (Fujisaki et al. 2018a; Vanlauwe et al. 2015). Overall, our data for sub-Saharan Africa suggests that SOC content on a continental scale is better explained by stabilization potential in soils (climate, geochemistry) than by different aboveground C inputs (vegetation)."

**E7:** *Secondly, you fully rely on the variable selection procedure for that claim. Your full model seems to show that forest as a land use significantly increases SOC by 0.26 units across the dataset (Table B2), which is about half of what is predicted for a unit increase in Caex (0.54). For Caex, as a continuous standardized variable, 1 unit increase means increasing Caex by 1 SD (derived from the Caex distribution from the whole dataset), which is about 11 cmol kg-1 (while the mean is 10 cmol kg-1!). This is almost comparing two different things. To me, that forest effect size would even seem interesting enough to run parallel models for the different land-use classes, much like you do for the pH and MAP classes.*

**A7:** That is a great point and we added the sub-models (grouped by the land-cover classes) to our analysis. The results can be found in Figure 4e (line: 299 Figure 4 replaces the previous figures that showed the coefficient estimates for the different models). The results for the land cover models suggest that not all predictors are equally important among the land cover models, yet there is no clear pattern. The results of the sub-models (as shown in Figure 4) are described in line 295–326.

[Figure]

Figure 1: Explained variation (based on marginal R²) for each fixed effect, based on sequential fitting of the linear mixed-effects models grouped by a) Depth (topsoil: 0–20 cm; subsoil: 20–50 cm); b) pH classes (strongly acidic: 3.9–5.2 pH, moderately acidic: 5.2–6.1, neutral: 6.1–7.5, alkaline: 7.5–9.9); c) Wetness (number of wet months (P/PET > 0); 0, 1–3, 4-7); d) Weathering (CIA: Chemical Index of alteration: moderate: 10–88% CIA, high: 88–100%) d) land cover.

The changes we made in the method section reads:

Line 217: "For the 2$^{nd}$ research question, how geochemical controls on SOC vary between environmentally distinct sub-regions, we grouped the data based on a) $pH_{H2O}$, b) wetness, c) weathering, and d) land cover (Table 1). Soil $pH_{H2O}$ and weathering data were grouped with the number of categories chosen to maximize and equalize the number of samples in each category and to correspond with common $pH_{H2O}$ and weathering groups (Nesbit and Young 1982). In order to take seasonality of the sites into account separately, the data were divided into three categories based on the number of wet months (i.e. months with P/PET > 1). Land cover was grouped based on the four pre-defined categories. For each category within each sub-group, we built a linear mixed-effects model as previously described, yet only included the geochemical properties (clay + fine silt, $pH_{H2O}$, CIA, $M_{ox}$, $Ca_{ex}$, $pH_{H2O}*M_{ox}$) as fixed effects, since we intended to test if the importance of these predictors changed between environmentally distinct sub-regions

(Table 1). When CIA or $pH_{H2O}$ were used to create the categories, they were not included as a fixed effect in the corresponding sub-models.

Table 1: Grouping variables, sub-groups, number of samples and fixed effects used for the linear mixed-effects models

| Groups | Categories | n | Fixed effects |
|---|---|---|---|
| All samples | None | 1,601 | All, Climate, Geochemistry |
| Depth | Topsoil (0–20 cm) | 791 | Geochemistry |
| | Subsoil (30–50 cm) | 810 | |
| $pH_{H2O}$ | Strongly acidic (3.9–5.2 $pH_{H2O}$) | 404 | Geochemistry |
| | Moderately acidic (5.2–6.1 $pH_{H2O}$) | 399 | |
| | Neutral (6.1–7.5 $pH_{H2O}$) | 398 | |
| | Alkaline (7.5–9.9 $pH_{H2O}$) | 400 | |
| Wetness (Number of wet months (P/PET > 1)) | 0 wet months | 572 | Geochemistry |
| | 1–3 wet months | 367 | |
| | 4–7 wet months | 662 | |
| Weathering (CIA) | Moderate (10–88% CIA) | 801 | Geochemistry |
| | High (88–100% CIA) | 800 | |
| Land cover | Cropland | 429 | Geochemistry |
| | Forest | 228 | |
| | Grassland | 242 | |
| | Other | 702 | |

P: Monthly precipitation [mm]; PET: Monthly potential evapotranspiration [mm]; CIA: Chemical Index of Alteration [%]; Fixed effects: All (Mean annual precipitation (MAT), Aridity index (PET/MAP), depth, land cover, clay and fine silt, $pH_{H2O}$, CIA, oxalate-extractable metals ($M_{ox}$), exchangeable Ca ($Ca_{ex}$), $pH_{H2O}*M_{ox}$), Climate (MAT, PET/MAP), Geochemistry (clay and fine silt, $pH_{H2O}$, CIA, $M_{ox}$, $Ca_{ex}$, $pH_{H2O}*M_{ox}$)"

*E8: Add to that that land use is a categorical predictor (the only one, between all your continuous predictors), and that you lose more degrees of freedom for the inclusion of land use as a categorical predictor in the model, compared to any continuous predictor. Obviously, this would be penalized by the AIC. If you would have used direct likelihood testing (using ML, and not REML estimation) of your models, land use might even have been retained in the model, and than the message would have been completely different? Bottom line: I think variable selection in mixed effects is complicated, add to that that you mix one categorical with multiple continuous predictors. I'm not necessarily saying that you need to change the methods, but these things need to be better founded, evaluated and discussed in the paper. You now 'let the data speak' through your analyses, which is great, but you need to be aware of the potential pitfalls of your choices as an analyst. In this case, this could even affect the main message you give to the readership.*

**A8:** We have updated our analysis to address these considerations. We now use the maximum likelihood (ML) estimation, which is required to apply the L.ratio test and to evaluate the importance of the added fixed effects. The L.ratio test, in contrast to the AIC, is not biased towards continuous predictors. We hope that the changes we made regarding the regression analysis fully address these helpful comments made by the editor. In addition to the above-mentioned changes, we also ran a geochemistry-only and a climate-only model. We used variation partitioning to identify how much of the explained SOC variation is shared between those two models. Those results are now presented in Figure 3 (line 289):

| Model | MAT | PET/MAP | Depth | Land cover | Clay + fine silt | $pH_{H2O}$ | CIA | $M_{ox}$ | $Ca_{ex}$ | $pH*M_{ox}$ |
|---|---|---|---|---|---|---|---|---|---|---|
| Full ($R^2 = 0.72$) | 0.17 (-) | 0.30 (-) | 0.05 (-) | 0.01 (+) | 0.04 (+) | 0.00 (-) | 0.00 (-) | 0.08 (+) | 0.05 (+) | 0.01 (-) |
| Geochemistry ($R^2 = 0.46$) | – | – | – | – | 0.01 (-) | 0.00 (-) | 0.04 (-) | 0.26 (+) | 0.11 (+) | 0.04 (-) |
| Climate ($R^2 = 0.48$) | 0.17 (-) | 0.30 (-) | – | – | – | – | – | – | – | – |

[Figure]

Geochemistry: Clay + fine silt, $pH_{H2O}$, CIA, $M_{ox}$, $Ca_{ex}$
Climate: MAT, PET/MAP

19%    27%    21%

Geochemistry + Climate: 67%

Figure 2: a) Marginal $R^2$ for each predictor based on sequential fitting of the linear mixed-effects models of all samples ($n_{Total}$ = 1,601) for the full, geochemistry-only and climate-only model. Sign in brackets refers to the correlation between the predictors and soil organic carbon, respectively. Bold values have a p-value < 0.05 based on likelihood-ratio test; b) Venn-Diagram illustrating the independent explained and shared explained variations by the geochemistry-only and the climate-only linear mixed-effects models.

Line 290: "The marginal $R^2$ for the geochemistry model was 0.46; almost the same as for the climate model ($R^2$ = 0.48). For the geochemistry model, the contribution of $M_{ox}$ and $Ca_{ex}$ to explain SOC content was much higher than in the full model (Figure 3a). Based on variation partitioning, 27% of the explained variation is shared between the geochemistry model and the climate model, whereas the variation explained by the geochemical or climate variables alone is 19% and 21%, respectively (Figure 3b)."

*E9: in line 163 you state that the slope of the regression varies across the sites, but if I understood your methods section well, than you fitted random intercept-only models? Needs clarification.*

**A9:** Thank you very much for recognizing this inconsistency. It is correct that we only fitted random intercept models. We removed the word *slope* in line 193.

**Reviewer #1: Point-by-Point response**

**General comments**

*REVIEWER10: This very interesting paper tries to explain the heterogeneity of soil organic contents in tropical soils. Pedogenic, climatic and land cover properties were related to SOC contents to identify the main factors that control SOC. It mobilized numerous original data from a network of soil measurements through Africa. Two soil depths (0-20 cm and 20-50 cm) were considered. The paper is really worth a reading as the results were rich and surprising.*

**ANSWER10:** Thank you very much for taking the time to review our manuscript and your most constructive feedback. Our detailed answers can be found below.

**R11:** *However, the work needs to be more explicit to be really compelling on some points (i.e. please give the main soil types studied: is there any andosols or young volcanic soils which could reinforce the role of Al, Fe extractable with oxalate in the SOC stabilization? It should be easy to dismiss that hypothesis).*

**A11:** This is a great point. Unfortunately, we do not have field data on the main soil types for the location where these AfSIS and LDSF samples were taken. When extracting data from global and African soil-type products, none of the soil profiles was classified as Andosols. However, those products are usually not that precise and it is difficult to derive detailed conclusions. Based on the position of some soil profiles (e.g. Rift Valley in East Africa) it might be possible that some of them are Andosols or influenced by volcanic activities in the past.

Based on this comment and also on one comment from Reviewer_02 (see comment R38) we added the following paragraph to the method section in the manuscript.

Line 175: "Due to the lack of precise data products for lithology and soil types in sub-Saharan Africa, we did not include these variables in our analyses. Soils at AfSIS sites (Figure 1) developed mainly from two parent material types: i) metamorphic and ii) volcanic rocks (Hartmann and Moosdorf 2012; Jones et al. 2013; Schlüter 2008), likely modified throughout the Quaternary. i) Metamorphic rocks are most commonly found in West Africa, Southern Africa and Madagascar. These regions are characterized by old cratons, except for Madagascar, which is influenced by Mesozoic volcanism (Schlüter 2008). Most of these soils are classified as Ferralsols (WRB soil classification system; Jones et al. 2013). Related AfSIS soils from those regions are usually highly weathered with low $pH_{H2O}$ values. In contrast, soils derived from ii) volcanic rocks are mainly found in the East African Rift System. They are usually younger and less weathered (Buringh 1970). Beyond the influence of volcanic rocks, $Ca^{2+}$ rich soils are frequent in East Africa."

**R12:** *Discussion on the limit of your study and on the differences with the previous studies could deepen the significance of your results. The clay and fine silt contents seem to have poor impact on the soil organic content on the contrary of previous results. OK but previous results have related SOC to the clay+silt size fraction at 0-20 µm, you have related SOC to the clay+silt size fraction at 0-8 µm. Please discuss that point?*

**A12:** Thank you for pointing out this important aspect. We realized that we have to be more explicit on why we used the rather unusual cut-off of <8 µm (clay + fine silt), instead of the more standard <2 µm (clay) or <20 µm (clay + silt). Our initial idea was to use clay content only. This particle size fraction is commonly used in soil and biogeochemistry studies to link SOC stabilization with clay and clay minerals (e.g. Rasmussen et al. 2018; Quesada et al. 2020; Bruun et al. 2010). However, the reproducibility for duplicates in our data set was sometimes poor for particles <2 µm, but much more reliable for particles <8 µm (Figure A1). This is the main reason why we decided to use the cut-off <8 µm. As this is closer to the pure clay fraction (< 2 µm), it allows us to better link our results to the stabilization capacity of clay content/clay minerals. A larger cut-off would include more silt-sized aggregates, which might result in additional and different C stabilization mechanisms, as discussed in more detail in Six et al. (2002).

[Figure]

Figure A1: Duplicate measurements of the particle size distribution data [%]. Panel a) Duplicate 1 and 2 <2 µm; b) Duplicate 1 and 2 <8 µm; c) Duplicate 1 and 2 <20 µm

However, based on the comment we also re-ran our analysis with the clay + silt size fraction (0–20 µm) and compared it to the results with the clay + silt size fraction (0–8 µm). We will display and discuss the findings briefly in the following section and in the supplement (Figure A1).

When comparing the raw data, there is a slight increase in the correlation coefficients between SOC and particle size data <8 µm and <20 µm, respectively (Table A1.1 and Figure A1.2). Yet, the differences between the respective correlation coefficients are always smaller than 0.1 (Table A1.1).

Table A1.1: Correlation coefficient between SOC and particle size data <8 µm and <20 µm for all samples (n = 1,601), topsoil (0–20 cm; n = 791), and subsoil (20–50 cm; n = 810)

| Samples | <8 µm | <20 µm |
|---|---|---|
| All | 0.32 | 0.41 |
| Topsoil | 0.37 | 0.46 |
| Subsoil | 0.43 | 0.49 |

[Figure]

Figure A1.2: a) Soil organic carbon (SOC) content [wt-%] and clay and fine silt content <8 µm [%] by depth; b) SOC content [wt-%] and clay and fine silt content <20 µm [%] by depth.

In addition, we also re-ran all our models that only contained clay + fine silt (<8 µm) as an explanatory variable with clay + silt <20 µm. Overall, the differences between the two particle size groups (<8 µm and <20 µm) became even smaller (Table A1.3) compared to the raw data. The differences between the $R^2$ for the different models was always less than or equal to 0.07. Furthermore, we also substituted the variable clay + fine silt <8 µm with clay + silt <20 µm within the entire random forest models (one model for each depth layer with all explanatory variables). For the topsoil samples, the $R^2$ of the <20 µm model decreased by 0.01 (from 0.71 to 0.70) and stayed the same for the subsoil samples (0.72).

Table A1.3: Summary table of $R^2$ for the different models (linear mixed effects model and random forest) for the two different explanatory variables (<8 µm and <20 µm) for all samples (n = 1,601), topsoil (0–20 cm; n = 791), and subsoil (20–50 cm; n = 810)

| Model | Linear mixed model | Random forest (topsoil) | Random forest (subsoil) |
|---|---|---|---|
| Clay + fine silt <8 µm | 0.01 | 0.12 | 0.12 |
| Clay + silt <20 µm | 0.03 | 0.17 | 0.19 |

Based on the results of the re-analysis with the size fraction <20 µm, we argue that both groups show more or less similar results and that the overall picture does not change. Therefore, we suggest keeping the size fraction group <8 µm, which only includes clay and fine silt and is more closely related to the pure clay content (<2 µm).

Based on the previous discussion, we made the following changes in the manuscript to better clarify the selected particle size data for the readers. In addition, we added the figures and tables shown here to the supplement of the manuscript – they are all displayed together in Figure A1.

Line 145: "We used 8 µm as cut-off to capture all clay and fine silt particles. Results were comparable to <20 µm (see SI material Figure A1), but <8 µm was selected because it is more relevant to our interest in studying the influence of smaller particles with large surface area on SOC concentration. In addition, particles <8 µm resulted in a reproducible fraction across soil types, unlike using only clay particles <2 µm (Figure A1)."

*R13: The separation of the two-studied soil depths is not always clear in the Ms. Can it affect the results?*

**A13:** We revised the manuscript carefully and tried to be clearer regarding the two sampling depths, and where the results vary between topsoil and subsoil:

Line 238: "Regression tree analysis was applied to obtain an easily interpretable and non-linear model for the entire dataset and for both depth layers (topsoil vs subsoil) that best describes the existing data (Breiman et al. 1984)."

Line 295: "Differences between the predictors were negligible for the two depth models (topsoil vs subsoil).

Line 337: "Overall, the results showed that the explanatory variables did not differ much between the depth intervals (topsoil vs subsoil), while their magnitude did."

A: In addition, it is really likely that the selected sampling depth intervals have an effect on the results. We added a paragraph in the discussion section.

Line 413: "

*Depth*

For the depth models, predictor differences were small between topsoil (0–20 cm) and subsoil (20–50 cm) samples (Figure 4a and 6). This may reflect the large depth increments for each of the two sampling depths, which may also explain the overall small explanatory power of depth in the linear-mixed effects model (Figure 3a). Since the identified SOC-controlling factors were similar for both depth layers (Figure 4a), differences in SOC content were likely driven by the fact that subsoil samples usually contain less SOC due to lower C inputs at greater depth (Jobbágy and Jackson 2000). Soil erosion at some sites (data not shown) might also dilute differences between the two depth layers, since water and wind can permanently remove surface soil."

*R14: One of your main result is that land cover do not explain SOC content at continent scale, so in the world wide discussion on C sequestration to mitigate CO2, what do you suggest?*

**A14:** When we first did the analysis, we were also surprised that land cover appeared irrelevant to explain the variation in SOC within the data. However, even after revising the statistics of the linear-mixed effects models (based on the comments made by the editor), land cover was still not an important predictor of SOC. This can be several reasons. It has been shown by previous studies that the effect of land cover changes is more important at smaller scales, whereas climate and soil development properties are more dominant at larger scales (Holmes et al. 2004, 2005). This is likely also true for $CO_2$ mitigation potential/strategies, since they depend on site and/or regionally specific aspects. The importance of site-scale management practices makes it difficult to draw any conclusions on C sequestration to mitigate $CO_2$, based on our analysis. In addition, the information we have about land use at the profile level is not detailed enough. We do not know what kind of land management practice is specifically applied in a given region, nor do we know much about the land use history at those sites. Additionally, other studies in Africa have shown that historic land use can be a more important determinant than current land use in determining current soil organic carbon stocks (Vågen et al. 2006).

Although our ability to draw conclusions from this dataset is limited, given the importance of the issue, we made the following changes in the manuscript:

Line 446: "The effect of land cover on SOC content was generally small in our models, even in topsoils (Figure 6i). Similar findings were recently encountered in a global study (Luo et al. 2021). One possibility may be that the relatively large 0–20 cm depth interval might dilute differences that would be more marked in the top few centimeters. However, we did observe differences in SOC content across land cover classes, with forests containing the highest amount of SOC – especially in topsoils (Figure 2a). Croplands had higher SOC content than grasslands, opposite of what is commonly observed in temperate regions (Prout et al. 2020).

Another possible explanation for the absence of land cover as an important predictor in our models, is that we lacked the detailed data necessary to disentangle impacts of different practices and land-use history. The land cover class cropland contained a wide variety of cultivated plots while more detailed information about land management practices were missing. This is particularly important since prior research in other regions showed that SOC stock changes in tropical cropland soils may be driven by C inputs (Fujisaki et al. 2018b). Additionally, historical land use may even play a more important role in explaining current stocks compared to recent land use (Vågen et al. 2006).

Furthermore, land cover may covary with other parameters (temperature, precipitation, geochemistry) to such a degree that it is not an explanatory variable. This might be the reason why the sub-models grouped by land cover did not show a clear pattern (Figure 4e). However, the land cover-only models resulted in small $R^2$ (linear mixed-effects models: 0.01; random forest: 0.10–0.16) which suggests that land cover is a poor predictor for our SOC data at this large spatial scale (Table B3). This may be due to the high variation of SOC content within the different land cover classes (Figure 2a). Land use changes and their impact on soil physico-chemical properties are scale-dependent and likely to be more distinct at smaller scales (Holmes et al. 2004, 2005). For example, land management and land degradation (i.e. erosion) are known to impact SOC stocks in regionals scales in sub-Saharan Africa (Winowiecki et al. 2016a).

Future studies are needed to better understand the impacts of land management and carbon storage potential in soils across sub-Saharan Africa at different scales (Fujisaki et al. 2018a; Vanlauwe et al. 2015). Overall, our data for sub-Saharan Africa suggests that SOC content on a continental scale is better explained by stabilization potential in soils (climate, geochemistry) than by different aboveground C inputs (vegetation)."

**R15:** *The final sentence on soil erosion is very OK but could be strengthen.*

**A15:** We understand that soil erosion is an important aspect of SOC stocks – especially in subtropical and tropical soils – and we mention this briefly in the introduction. During the field campaign, visible signs of soil erosion by water were reported: *none*, *sheet* (uniform removal of soils in thin layers; sign of sheet erosion include bare areas, water puddling, visible grass roots, exposed tree roots, and exposed subsoil or stony soils), *rill* (shallow drainage lines less than 30 cm deep), and *gully* (channels that are deeper than 30 cm). Most of the studied profiles showed no signs of erosion (~300) or sheet erosion (~375). The other profiles showed either rill erosion (~75) or gully erosion (~25). However, we did not include soil erosion in our main analysis for several reasons. First, the focus of our work was on natural processes. However, land cover, which can be seen as a broad proxy for potential soil erosion, did not explain much of the variance in SOC. Furthermore, we think that some of the other soil properties we included in our analysis are capturing the relevant indicators for soil erosion as well, such as pH, $Ca_{ex}$ and MAP. And lastly, similar to land cover, it is likely that the importance of soil erosion is more important at a regional scale.

To address this comment, we have added additional text on erosion within the manuscript (line 413 and 446, see text reproduced in the previous comments (A13 and A14)).

**R16:** *Your remark on line 436 ("… on a regional scale") is also OK. Please insist a little more on that point.*

*A16:* We hope that the changes we made in line 446 (see previous comment) now help to make this statement clearer and therefore only added a few words in line 490:

Line 490: "Overall, a combination of PET/MAP, $Ca_{ex}$ and $M_{ox}$ seems to be an appropriate set of variables to explain variation of SOC content on a continental scale across sub-Saharan Africa. This does not imply that other variables, such as clay and fine silt content and land cover are not good predictors on a regional scale as shown by previous studies."

**R17:** *You have also to be clear at the end of the Ms between predict SOC content (for instance to map SOC content or stock) and predict SOC dynamics (you have not studied SOC dynamic in your study): Ln. 437 to the end are not very clear.*

Line 494: "Future studies on large-scale SOC stabilization should consider measuring those soil properties to include them in models. This would likely improve the predictive capacity of these models and contribute to closing the gap

between our theoretical understanding of SOC concentration across large scales and our ability to improve terrestrial biogeochemical projections that rely on existing models."

**R18:** *After your result on the lack of significance between soil clay and SOC contents, what do you think about using the soil carbon saturation deficit to quantitatively assess the soil carbon storage potential in tropical soils? Could you add one or two sentences on that point.*

**A18:** The reviewer brings up a really interesting and relevant point. This topic is especially relevant for future strategies to increase the C amount in soils as proposed in the *4 per 1000* initiative. Following a literature review (e.g., Hassink 1997; Feng et al. 2013) on this topic, we have some concerns that our data and analysis will allow us to sufficiently address this topic in the presented manuscript. However, it would be a really interesting study to address this topic in sub-Saharan Africa based on a more systematic approach.

One concern is related to the fact that we do not always have a degraded site next to an undisturbed site – which is needed as reference site. In addition, the information we have about the degraded sites is not always sufficiently detailed. We do not have information, e.g., about exact land management practices, nor do we know much about land management history at those sites. Such information is crucial to put the results in a broader context since it is likely that the soil carbon saturation deficit is influenced by the type of management (e.g. till vs non-till, compost additions, enhanced residue return, …) and the time since the land was converted. Furthermore, we do not know the amount of C that is associated with the clay + silt fraction due to the fact that we only have bulk data available. However, the clay + silt fraction is of most interest regarding carbon storage potential in soils (Matus 2021). For this, it would also be good to have more information about the specific clay minerals and their specific surface area. Last but not least, soil carbon saturation deficit assessments are likely be more relevant on a regional scale, similar to $CO_2$ mitigation strategies.

Based on the previous discussion we made some changes (line 446; see answer above) which are related to the changes we made regarding the $CO_2$ mitigation comment made earlier by the reviewer).

**Specific comments**

**R19:** *Abstract: The conclusion of the abstract is not clear/strong enough.*

**A19:** Based on the changes we made throughout the manuscript, we revised the abstract.

Line 15: "Soil organic carbon (SOC) stabilization and destabilization has been studied intensively, yet the factors which control SOC content across scales and regions remain unclear. Earlier studies demonstrated that soil texture and geochemistry strongly affect soil organic carbon (SOC) content. However, those findings primarily rely on data from temperate regions with soil mineralogy, weathering status and climatic conditions that generally differ from tropical and sub-tropical regions. We investigated soil properties and climate variables influencing SOC concentrations across sub-Saharan Africa. A total of 1,601 samples were analyzed, collected from two depths (0–20 cm and 20–50 cm) from 17 countries as part of the Africa Soil Information Service project. The data set spans arid to humid climates and includes soils with a wide range of $pH_{H20}$ values, weathering status, soil texture, exchangeable cations, extractable metals and land cover types. The most important SOC predictors were identified by linear mixed-effects models, regression trees and random forest models. Our results indicate that geochemical properties, mainly oxalate-extractable metals (Al and Fe) and exchangeable Ca, are equally important compared to climatic variables (mean annual temperature and aridity index). Together, they explain approximately two thirds of SOC variation across sub-Saharan Africa. Oxalate-extractable metals were most important in wet regions with acidic and highly weathered soils, whereas exchangeable Ca was more important in alkaline and less weathered soils in drier regions. In contrast, land cover and soil texture were not significant SOC predictors on this large scale. Our findings indicate that key factors controlling SOC across sub-Saharan Africa are similar to those reported for temperate regions – except for soil texture and land cover. The similarities between the two regions suggest that SOC content in highly weathered soils is not primarily related to long-term soil development, but to common geochemical and climatic properties."

*R20: Ln. 62: "broader variables such as clay content: : : " Ok but clay content implies SOC protection and association with minerals (L. 61). Please rephrase. Ln. 64 "variety of processes" Please be more explicit (aggregation, organo mineral association: : :) Ln. 64 "differ" or their relative importance differ? (different hierarchisation). Please specify*

**A20:** A similar comment was brought up by Reviewer_02. We agree that the paragraph was not really well-written. Since modeling is not the main focus of our manuscript, we would like to keep this paragraph rather short. However, we think it is an important aspect worth mentioning in the introduction. We revised the paragraph:

Line 58: "SOC stabilization is commonly conceptualized as competition between accessibility for microorganisms versus chemical associations with minerals (Oades 1988; Schmidt et al. 2011). These processes are often only considered implicitly by models (Blankinship et al. 2018; Schmidt et al. 2011). Instead, models commonly rely on broader variables such as clay content, which is used as a proxy for sorption and other organo-mineral interactions (Rasmussen et al. 2018; Schmidt et al. 2011). These more generic variables integrate a variety of stabilization processes which can be difficult to disentangle. They can differ in their relative importance and may not adequately capture soil mineralogy and chemistry across different ecosystems and climate zones. Hence, improving the predictive capacity of such models requires not only a better understanding of the factors that control SOC dynamics, but also verification (or falsification) of those new findings in regions that are underrepresented in field studies and models."

*R21: End of the introduction: Please specify your hypothesis, you have not searched factors randomly. Please be more explicit. Your questions are too vague. Which soil properties you will focus on? What findings do you except? You can base hypothesis on soil properties, climate properties and land cover. You have everything to make a more compelling introduction. Please explicitly justify the soil properties and the soil depth you will focus on in your study.*

**A21:** We revised the end of the introduction carefully to make it more explicit and to give the reader a road-map for the following sections.

Line 86: "This data set covers a wide range of climatic and mineralogical conditions – from very arid to humid regions, with different $pH_{H2O}$ values, soil texture, weathering status, exchangeable cations and extractable metals – allowing us to test different parameters to explain the variation in SOC content in subtropical and tropical soils across sub-Saharan Africa for two distinctive depth layers (topsoil: 0–20 cm and subsoil: 20–50 cm). Here, we use this continental-scale data set to address the following research questions:

1. Which soil properties and climate parameters best explain SOC content variation across sub-Saharan Africa?

We explored the importance of soil texture, exchangeable Ca, oxalate-extractable Al and Fe, soil $pH_{H2O}$, mean annual temperature, aridity index (PET/MAP), land cover and weathering status to explain variation in SOC content on a continental scale. We expect that oxalate-extractable metals, soil texture and climate will be among the most important predictors of SOC concentration.

2. How do geochemical SOC-controlling factors vary between environmentally distinct sub-regions?

Due to the heterogeneity of climate and soil conditions across sub-Saharan Africa, we expect to see different geochemical controls explaining variations in SOC content between regions. For example, we expect exchangeable Ca will be most important in regions that are drier with less weathered and alkaline soils, while oxalate-extractable Al and Fe will mainly be important in humid regions with highly weathered and acidic soils."

*R22: Ln. 105 "mid-infrared spectroscopy data" data or model? How have chosen the representative spectral data? Have you separate the two soil depth? Why have you finally chosen to study the lab measurements and not the predictions?*

*A22:* Thank you for this critical comment. It should be "mid-infrared spectroscopy model" and we changed it in the manuscript accordingly (line 113). The reference data set for the mid-infrared spectroscopy model was chosen using the Kennard-Stone algorithm (Kennard and Stone 1969). First, the spectra is decomposed using a PCA and selecting a subset using Kennard-Stone that gives a uniform coverage across the spectral space of the data, including also boundary cases. Terhoeven-Urselmans et al. (2010) contains more information on this and also the adaptations of the approach: "The adaptations were (i) that no samples from the same soil profiles were allowed to be split between the

calibration and validation sets, and (ii) that the two extreme score values of each principal component were chosen for the calibration set and the next extreme score values chosen as starting samples for the validation set."

We decided to analyze the lab measurements only for this study since they contained more variables of interest in terms of SOC stabilization, such as oxalate-extractable Fe and Al.

Line 114: "The calibration subset was chosen to maximize the variation of the spectral data using the Kennard-Stone algorithm (Kennard and Stone 1969). More information about this approach can be found in Terhoeven-Urselmans et al. (2010). This selection strategy results in unequally distributed samples across 51 of the 60 sentinel sites, yet captures the variation of the original data set."

**R23:** *Ln. 120 why have you chosen this limit at 8 μm? What arguments?*

*A23:* We kindly refer here to the answer (A12) we gave above to the more general comment from the reviewer regarding clay and clay+silt content.

**R24:** *Ln. 133 It seemed that soil texture was performed without organic matter destruction? Is that really the case? Please specify.*

**A24:** The pretreatment Calgon and ultrasonic dispersion that was done prior to Laser Diffraction measurement were done to disperse aggregates, yet it does not destroy carbon.

Line 144: "Each sample was shaken for 4 min in a 1% sodium hexametaphosphate (calgon) solution with ultrasonic energy before measuring to disperse aggregates."

**R25:** *Ln. 136-140 it was not really clear to me.*

**A25:** We hope that the following changes made it easier to understand the correction we did for samples that contained inorganic carbon before calculating the Chemical Index of Alteration (CIA). According to Nesbit and Young (1982) the equation holds only for soil samples that do not contain carbonates or apatite. If so, an appropriate correction is needed.

Line 149: "Aluminum, Ca, K and Na concentrations were used to calculate the chemical index of alteration (CIA) after Nesbit and Young (1982), using the following equation:

$$CIA = Al_2O_3 / (Al_2O_3 + CaO + K_2O + Na_2O) * 100 \quad (2)$$

where CaO is the amount incorporated in the silicate fraction. Correction is necessary for samples that contain carbonates and apatite (Nesbit and Young 1982). We adopted an approach introduced by McLennan (1993): The correction assumes that Ca is typically lost more rapidly than Na during weathering. If a soil sample contained inorganic C ($C_{total} - C_{org}$; used as a proxy for carbonates and apatite) and the CaO content was greater than that of $Na_2O$ in the same sample (n = 476), then the CaO concentration was set to that of $Na_2O$ from the same sample (Malick and Ishiga 2016)."

**R26:** *Ln. 145 "1,601 soil samples" is that on the 2,002 measured soil samples?*

**A26:** Line 160: "This resulted in a total of 1,601 soil samples (out of the original 2,002 samples) at 45 sentinel sites across 17 countries."

**R27:** *Statistical analysis: Note for the editor: A deep analysis of the statistics has to be done by a reviewer more competent than I am in statistics.*

*Table 1: Please specify in the title that it is a summary for the two soil depths (0-20, n= ?, and 20-50 cm, n= ?)*

**A27:** Please note that the table numbering changed from 1 to 2.

Line 265: "Table 2: Summary statistics of all numerical soil and climate variables for the entire data set ($n_{total} = 1,601$; $n_{Topsoil} = 791$; $n_{Subsoil} = 810$)"

**R28:** *Figure 2: Please prefer unit (g kg-1 soil) than*

**A28:** With this manuscript we do not only target the soil science community, but also those of biogeochemistry and geochemistry. The two latter groups generally use wt-% instead of g kg$^{-1}$ soil, since the data are compositional. If

wt-% is accepted by SOIL, we prefer to keep the unit as it is. Otherwise we can change the units for SOC, $Al_{ox}$, and $Fe_{ox}$ from wt-% to g $kg^{-1}$ soil (wt-% multiple by 10) in the entire manuscript, the supplement and the corresponding R markdown file. We leave this decision to the handling editor.

**R29:** *Ln. 250 could you give some essential data from the Table B2 in the Ms. It could be nice for the ones who do not go and see the supplement data.*

**A29:** This table no longer exists because of to the changes we made to the linear-mixed effects models.

**R30:** *Table 2: "Depth (Subsoil)" is not clear. It is clear when reading ln 246-247, but it is not when reading the table alone. Table should explicit by itself.*

**A30:** This table also no longer exists since we are not using the coefficient estimates anymore to evaluate the linear-mixed effects models.

**R31:** *Figure 3: I did not get the sentence "Note that the x-axis is truncated…" At what Caex content value it should end? 76? Please specify.*

**A31:** Please not that this is now Figure 5.

Line 310: Figure 5 caption: "[…] Note that x-axis is truncated for improved visualization, which removes 3 data points ($Ca_{ex}$ = 53.91, 54.58, and 75.66 cmol$^+$ kg$^{-1}$) […]"

**R32:** *Ln. 286-287 and 301-302 Please make it clearer.*

**A32:** Some of the confusion was likely caused by the fact that we referred to the wrong figure: it should be Figure A5a instead of Figure 5b. We apologize for any inconvenience this might have caused.

Line 328: "The root mean squared error (RMSE) for the topsoil regression tree was 1.47 wt-% (range: 0.80–3.11 wt-%) and for the subsoil regression tree was 0.67 wt % (range: 0.44–2.26 wt-%); the relative RMSEs were 0.65% and 0.48%, respectively. In the topsoil regression tree (Figure A5a) $Fe_{ox}$, MAT and PET/MAP were the most important predictors to split and explain variation in SOC concentration."

Line 344: "The random forest models had a RMSE of 1.31 wt-% and a R² of 0.70 for the topsoil samples, and for the subsoil samples a RMSE of 0.87 wt-% and a R² of 0.72."

**R33:** *Ln. 304 yes we could already notice that point on Figure 3c according to the pH class (20 or 30 cmol+/kg)*

**A33:** It is true that this point could already be noticed on Figure 5a (previously Figure 3c) and we shortened the sentence. Nevertheless, Figure 5a shows raw data, while the sentence in line 346 and Figure 6b refer to the random forest model results that confirm the pattern observed within the raw data. The sentence now reads:

Line 346: "However, in subsoils, the predictive power of $Ca_{ex}$ was reduced (Figure 6b) $_{ex}$~~ concentrations of about 20 cmol~$^+$~kg~$^{+}$."

**R34:** *Ln. 321-322 it is not very compelling. You have shown previously that there are slight differences between 0-20 and 20-50 cm (Ln 294, 253..) Please be clearer.*

**A34:** Based also on comments made by Reviewer_02, we changed the structure of this section and re-wrote parts of it. This section no longer contains information about the role of depth in our models. In addition, we also added a section in the discussion where we discuss the role of depth in our models (see previous answer (A13) and line 414 in the manuscript).

**R35:** *Ln. 334 Please specify the size of the clay and fine silt fraction of the previously studies and yours. Discuss with these differences in mind.*

**A35:** We tried to be more explicit in regard to the cut-offs for the clay and clay+silt fraction that have been used by other studies and added the cut-off were possible. However, we could not always find the exact cut-off used in those studies (e.g. Quesada et al. 2020; Rasmussen et al. 2018). In addition, we added the following sentence to the section:

Line 434: "Even though these studies used various cut-offs to define the clay (<2 µm), clay and fine silt (<8 µm), and clay and silt fraction (<20 µm), they all illustrate that the relationship with SOC can be complex in subtropical and tropical soils."

**R36:** *Ln. 360-361 Yes it is OK, but please end the §by your results and not by the results of other scientists. Your result seemed to show that this mechanism seem to be stronger to explain SOC stabilization than straight organo-mineral association in quite different tropical environment. There is no words on particulate organic matter in your discussion. Have you any data to inform the role or the proportion of this organic matter pool in the total SOC content in your soil samples.*

**A36:** Unfortunately, we do not have any data about the particulate organic matter pool and thus, cannot discuss it in our manuscript. However, it is likely that some of those processes are integrated within the topsoil samples due to the relatively large sampling depth interval (0–20 cm). We changed the last two sentences to finish this section with our own results:

Line 386: "In conclusion, the important role of $Ca_{ex}$ in our data set was most pronounced in dry regions, dominated by alkaline and less weathered soils. However, it also played a role in explaining the SOC variation in wetter regions and more acidic soils, which is supporting the overall importance of $Ca_{ex}$ in stabilizing SOC."

**Reviewer #2: Point-by-Point response**

**General comments**

**REVIEWER37:** *The manuscript "Continental-scale controls on soil organic carbon across sub-Saharan Africa" describes a continental-scale analysis of associations between soil organic carbon and soil physico-chemical properties across Africa. The manuscript outlines a novel soil dataset collected at the Afsis "sentinel sites", and then steps through several statistical analyses that tease apart associations between carbon, extractable metals, and soil exchange pools across different domains of climate, soil pH, and soil weathering status. The authors conclude that short-range order (oxalate extractable Al) and to an extent Fe explain much of the variation in carbon stocks in wet/acid soils, whereas exchangeable calcium explains much of the variation in dry/alkaline soils. soil texture and land use appear largely irrelevant at this scale.*

*I think this manuscript is excellent and will be a very useful contribution to the study of soil geography. While the primary result has been identified in earlier studies (particularly Rasmussen et al.'s 2018 study), this manuscript applies to a different geographic domain (tropical and subtropical Africa) and with a more systematic data collection effort. It also considers soil weathering status using total elemental inventories and chemical weathering indices, which adds novelty. The results provide clear confirmation of the patterns hinted at in the Rasmussen study, and also point to some new complexities (particularly in relation to Fe). Furthermore, this study applies to data that were collected in a systematic sampling effort–hence these results should be considered more conclusive than those in earlier studies. The manuscript does a good job of balancing different statistical approaches, and stands as an example of how data-driven modelling tools (i.e. random forests) can be used responsibly in a processoriented way to compliment more traditional statistical approaches. While at points the interpretation slides into a more descriptive "data-mining" posture, it is also punctuated with insightful process-based insights. In short: overall this is a strong manuscript!*

*My main criticisms apply to the way the methods are presente–I think some details are left out or insufficiently documented. I also think that the methods and discussion sections could use more of a "road map" at the start– particularly the discussion, which dives into a description of the correlations between different variables where it could start with some pithy statements summarizing the high level process-based interpretation.*

**ANSWER37:** We highly appreciate this very thoughtful and appreciative review. Thank you for taking the time to carefully comment on our manuscript. We will address the suggestions in detail in the following response.

*R38: I also would appreciate a bit more discussion of the underlying geographic patterns in the context of African geology (perhaps just a paragraph). I realize that the existing geospatial products don't allow for a thorough quantitative analysis of geologic state factors, but some limited qualitative might be good. More specifically the authors might address how parent material, soil age, and erosion rates vary (or do not vary) across the sampling locations, and how these might exert some influence on the results independent of climate.*

**A38:** Thank you for this comment. We agree that there seems to be no appropriate geospatial product for geologic state factors that allows for a thorough quantitative analysis. However, the studied soils developed mainly from two parent material types: i) metamorphic and ii) volcanic rocks. Metamorphic rocks are most commonly found in West Africa, Southern Africa and Madagascar. Soil from those regions are usually highly weathered with low pH values. Volcanic rocks are usually younger and less weathered and are mainly found in the East African Rift System.

Erosion rates were reported in the field for each profile. Yet, most of the studied profiles showed no or minor signs of erosion (> 600 from ~ 775 profiles). However, we did not include soil erosion in our main analysis for several reasons. First, the focus of our work was on natural processes. However, land cover, which can be seen as a broad proxy for potential soil erosion, did not explain much of the variance in SOC. Furthermore, we think that some of the other soil properties we included in our analysis are capturing the relevant indicators for soil erosion as well, such as pH, $Ca_{ex}$ and MAP. And lastly, similar to land cover, it is likely that the importance of soil erosion is more important at a regional scale (see also A15).

We added a paragraph in the method section, were we address this comment (see line 175, and 414 and 446 in the manuscript and text reproduced in previous answers (A6, A11, A13) to Reviewer_01).

**Specific comments:**

*R39: Lines 39-40: The phrase "complex analytical approaches with a large number of parameters" is somewhat opaque. Perhaps substitute something more specific?*

**A39:** Line 36: "Assessing the state of soils and their potential response to climate and land-use change requires carefully designed sampling strategies, combined with systematic analytical and statistical analyses across locations and scale (IPCC 2019)."

*R40: Lines 62-63: To be fair here: there is an implicit representation of competition between microbes and minerals in Earth System models via clay content. There are two issues in this case: (1) competition between minerals and microbes is not represented in an explicit, mechanistic way; and (2) clay content doesn't capture the relevant aspects of soil mineralogy or chemistry. I think this manuscript addresses the latter issue more than the former.*

**A40:** A similar comment was brought up by the Reviewer_01. We agree that the paragraph about model approaches was not that accurate and have now updated it. Since this is not the main focus of our manuscript we would like to keep this paragraph rather short. However, we think it is an important aspect worth mentioning in the introduction (see line 60 in the manuscript and reproduced text in the answer to Reviwer_01 (A20)).

*R41: Lines 129-131: Was this digestion quantitative? I believe some silicates are resistant to aqua regia. Perhaps clarify whether these should be considered total elemental pools or simply aqua-regia-digestible pools, as this may influence the interpretation of the CIA (though probably not much I imagine).*

**A41:** Line 139: "Aqua regia acid digestion was applied for major and trace elements, including Al, Ca, K and Na. Although this method does not give absolute total contents, it does give results sufficiently close to accepted values for different soils (McGrath and Cunliffe 1985)."

*R42: Line 160: It would be good to include a short overview paragraph at the start of the statistical analysis section explaining the overall strategy. It seems that several approaches were applied to the same data: linear mixed effects models, regression trees, and random forests. I can see how the approaches complement each other (the mixed effects models seem more conservative and permit statistical hypothesis testing while accounting for non-independence of the data, but the CART based approaches can handle non-linearity). This is explained later, but the readers will benefit from a quick signpost at the start. Similarly, the discussion section is hard to follow at the start. I strongly*

*recommend adding a concise paragraph at the beginning of the discussion that identifies the major results. As it stands now the discussion dives right into the details and I can only identify an emergent narrative at the end.*

**A42:** Statistical analyses section: Line 185: "We used three different statistical approaches, including linear mixed-effects models, regression trees and random forests to determine geochemical and climatic parameters that best explain SOC variation across sub-Saharan Africa. In brief, we used linear mixed-effects models to handle the hierarchal sampling design of the AfSIS data set, whereas regression trees and random forests enabled us to account for non-linearities within the data. More precisely, we used regression trees as a qualitative tool to explore and understand the structure of the data, whereas random forests offered more generalizable models. All statistical analyses were performed within the R computing environment (Version 4.0.0, R Core Team 2020). The R Markdown file in the SI provides the code to reproduce all our analyses."

Discussion section: Line 359: "Climate and geochemical variables are similarly important in explaining SOC variations across sub-Saharan Africa (Figure 3), which is in line with findings from a global study (Luo et al. 2021). However, the explanatory power of climate and geochemical variables is not independent of each other, reflecting the overall strong interaction between climate and geochemistry (Doetterl et al. 2015). Since it is likely that, in the long term, climate variables have predominantly indirect effects on SOC dynamics through their influence on soil geochemistry, we focus our discussion on those geochemical variables ($Ca_{ex}$, $Al_{ox}$ and $Fe_{ox}$) that showed the highest explanatory power with respect to SOC content across all models. In addition, we discuss the role of depth, clay + fine silt content, and land cover in explaining SOC variations on a continental scale, since these were identified by other studies to play an important role in SOC dynamics."

*R43: Lines 167-171: I understand that the transformation is necessary for comparing different predictors on the same scale. However, what does the transformation mean with respect to the functional relationships in the data? Are the models linear with respect to the original scale? I suspect not: a linear model fit to transformed data is not necessarily a linear model with respect to the original data. This is worth noting, even if the analysis stays the way it is.*

**A43:** We no longer use the coefficient estimates to evaluate the importance of individual predictors. However, to meet model assumptions we still transform the data prior to regression analysis. It is correct that transformation and standardization of the data does not mean that the original relationship between SOC and the predictors is always linear. We have clarified this in the text. This is one of the reasons why we also used regression trees and random forests.

Line 198: "The relationship between SOC and the predictors of the original data may not be linear."

*R44: Line 183: How was the hierarchical clustering done?*

**A44:** Thank you for the critical question. We realized that 'hierarchal clustering' is not the appropriate term here and apologize for any confusion that was caused by that. We used a built-in function called *cut_number* from the *ggplot2* package in *R* which allows to control for the number of groups, whereas the cut-offs are determined by the function internally to approximately equalize the number of samples in each group. We tried different numbers of groups to match common pH and CIA classes while trying to maximize the number of samples in each group (i.e. keeping the numbers of groups as small as possible) at the same time. The exact approach can be found in the R Markdown file in the SI. We have clarified this in the text.

Line 218: "Soil $pH_{H2O}$ and weathering data were grouped  with the number of categories chosen to maximize and equalize the number of samples in each category and to correspond with common $pH_{H2O}$ and weathering groups (Nesbit and Young 1982)"

*R45: Line 204: The spatial partitioning is really laudable. It is surprising how infrequently this is done, and it really should be a community standard. Thank you for being rigorous!*

*Line 242: Please introduce the marginal/conditional R-squared values before mentioning here. To many readers this distinction might not be obvious.*

**A45:** We now explain the term marginal $R^2$ in the method section.

Line 208: "The variation explained by each fixed effect was obtained by calculating the marginal $R^2$ (excluding the variation explained by the random effects siteID/clusterID/plotID) for each model and subtracting the $R^2$ from the previous fitted model using the function *r.squaredGLMM* from the *MuMIn* R package (Barton 2020; Nakagawa and Schielzeth 2013)."

*R46: Figure 2: The univariate linear regression fits in this figure are purely for illustration? Perhaps mention them briefly in the statistical analysis section.*

**A46:** It is correct that we added those regression lines in Figure 2 for illustrative reasons. The regression line follows the linear equation y~x. Since the linear regression line in the figure is not used in further analysis and not important for the discussion, we have now included the formula and clarifying information in the caption of Figure 2, rather than in the methods section:

Line 279: "Figure 2: […] Gray area around fitted linear regressions (y~x, for illustration only) in b) and c) show the 95% confidence interval. […]"

*R47: Figure 3 (and throughout): How were confidence intervals obtained? They are reported throughout the paper, but unless I missed something the method used to obtain them is not reported.*

**A47:** Due to the changes we made to statistics of the linear-mixed effects models, we do not show any figures with confidence intervals for the models any longer.

*R48: Line 289: How was the % variation explained obtained here? Is this an R-squared value for a reduced model? Or is it some sort of variable importance metric? Perhaps something is missing from the methods description?*

**A48:** We replaced the word *variation* with *data* and added an explanation in the method section. The percentage is referring to the relative number of observations in this particular node of the regression tree. In this particular case, the SOC content of 23% of the samples was predicted by using $Fe_{ox}$ and MAT only.

Line 246: "Absolute values at the bottom of each node indicate the predicted SOC content [wt-%] and the percentage corresponds to the relative number of samples in this node (Figure A6)."

Line 331 "About 23% of the SOC data could be explained by $Fe_{ox}$ and MAT alone."

*R49: Line 446: I hope that the data presented in this study are eventually made available in some easy-to-access way. A database of this size and completeness could be extremely valuable to other researchers and would be best archived on some sort of data repository rather than only available on request from the author.*

**A49:** Thank you very much for bringing up this important aspect of *Open Science*. We fully agree that the analyzed data set should be open and easily accessible to everyone. In parallel to this review process, we have been working on this. We archived the data on a repository under the following DOI: https://doi.org/10.34725/DVN/66BFOB, yet with an embargo until January 31, 2022. Access before the end of the embargo can be requested from Vågen et al. (2021).

Line 502: "The soil properties data set used in this study is available from the authors upon reasonable request and under the following DOI: https://doi.org/10.34725/DVN/66BFOB (Vågen et al. 2021)."

[revised manuscript text omitted]

---

## Author Response (AR2)

Dear Dr. Bauters,

Thank you very much for evaluating the revised version of our manuscript ("Continental-scale controls on soil organic carbon across sub-Saharan Africa", soil-2020-69). We are pleased that you and the referee are overall satisfied with the changes we made.

We addressed the minor comment from the referee. Please find below the change we made to address this comment.

Sincerely, on the behalf of all authors,
Sophie von Fromm

**Point-by-Point response**

*Reviewer1:*

*Lines 29-30: On this read through the manuscript, I find the I strongly disagree with the following sentence: "The similarities between the two regions suggest that SOC content in highly weathered soils is not primarily related to long-term soil development, but to common geochemical and climatic properties". I would argue that long-term soil development determines (or at least strongly influences) common geochemical properties. Soil development is expressed in soil properties. The findings of this paper do not indicate that long term soil development has no influence on organic C: in fact, quite the opposite, since exchangeable ions and reactive metal phases accumulate over time as soils develop. Is the point rather that the same geochemical factors dominate in tropical and temperate regions? If so, better to say that directly.*

**Answer1:** Thank you for pointing this out in the abstract. We made the following changes to the abstract to address the issue raised.

Line 15 to 29: "Soil organic carbon (SOC) stabilization and destabilization has been studied intensively. Yet, the factors which control SOC content across scales remain unclear. Earlier studies demonstrated that soil texture and geochemistry strongly affect SOC content. However, those findings primarily rely on data from temperate regions where soil mineralogy, weathering status and climatic conditions generally differ from tropical and sub-tropical regions. We investigated soil properties and climate variables influencing SOC concentrations across sub-Saharan Africa. A total of 1,601 samples were analyzed, collected from two depths (0–20 cm and 20–50 cm) from 17 countries as part of the Africa Soil Information Service project (AfSIS). The data set spans from arid to humid climates and includes soils with a wide range of pH values, weathering status, soil texture, exchangeable cations, extractable metals and land cover types. The most important SOC predictors were identified by linear mixed-effects models, regression trees and random forest models. Our results indicate that geochemical properties, mainly oxalate-extractable metals (Al and Fe) and exchangeable Ca, are equally important compared to climatic variables (mean annual temperature and aridity index). Together, they explain approximately two thirds of SOC variation across sub-Saharan Africa. Oxalate-extractable metals were most important in wet regions with acidic and highly weathered soils, whereas exchangeable Ca was more important in alkaline and less weathered soils in drier regions. In contrast, land cover and soil texture were not significant SOC predictors on this large scale. Our findings indicate that key factors controlling SOC across sub-Saharan Africa are broadly similar to those in temperate regions, despite differences in soil development history."